# BIGFAM - variance components analysis from relatives without genotype

**Jaeeun Jerry Lee** [1] **& Buhm Han** [1,2,3] ✉

Estimating variance components of phenotypes provides a fundamental basis for understanding complex traits. However, most existing methods require genotype data, which is costly to obtain and often unavailable, limiting their scalability. To address this limitation, we developed BIGFAM, a genotype-free framework that estimates variance components by genetic, shared environmental, and X chromosome effects using only phenotype data from relative pairs. We analyze variance components in Generation Scotland and UK Biobank datasets and demonstrate that BIGFAM's estimates show high correlation with genotype-based methods ($r = 0.85$ for heritability and $0.64$ for X chromosome components). We identify strong nuclear-family-specific shared environmental effects in dietary-related phenotypes. These results establish a new approach for analyzing variance components across diverse populations without the need for genetic data.

Phenotypes are influenced by various variance components, including genetic and environmental factors. In genetic studies of unrelated individuals, phenotype is commonly modeled as the sum of two independent components: genetic and environmental effects. However, for related individuals, phenotype modeling becomes more challenging due to the complex interplay between shared genetics and environment. The most widely used framework for related individuals is the ACE model[1], which partitions phenotypic variance into additive genetic (A), common (shared) environmental (C), and unique environmental (E) effects. The presence of shared environment, however, can confound the estimation of other variance components, making analyses using relatives particularly challenging.

To address these challenges, several methodological approaches have been developed for estimating variance components using relatives, though each has limitations. Genotype-based methods, such as REML-based[2] and regression-based[3] approaches, can partition variance components while accounting for relatedness. However, these methods have two major limitations: they may miss the contribution of ungenotyped variants that are not in linkage disequilibrium (LD) with genotyped variants, and they require large-scale genotype data, which is often costly to obtain.

Other approaches that do not require genotype data have also been developed. Twin studies, for example, can estimate variance

components by comparing trait correlations between monozygotic and dizygotic twins, separating genetic from shared-environmental components[4]. However, the limited availability of twin data restricts the scalability of this approach. In contrast, parent-offspring regression (PO-reg) provides a more scalable alternative by analyzing phenotypes of parents and offspring[5,6]. PO-reg typically allows for larger sample sizes compared to twin studies, but faces two key methodological limitations. First, it is hard to distinguish between genetic and shared-environmental variance components[6]. Second, estimates can vary significantly depending on the familial relationships[7,8], potentially due to relation-specific shared environmental effects such as maternal effects. Structural equation modeling (SEM)[8] addresses these limitations by explicitly modeling shared environmental effects. However, SEM parameters can be biased when including too many variables in the model[9]. This issue is particularly problematic when analyzing multiple degrees of relatives, as the number of shared environmental variables in the model increases substantially.

Additionally, these genotype-free methods share a critical limitation: they focus solely on autosomal effects, unable to analyze chromosome-specific genetic contributions. This limitation is particularly noteworthy for the X chromosome, which plays a crucial role in biological processes[10] and has unique characteristics. Unlike

[1]Interdisciplinary Program in Bioengineering, Seoul National University, Seoul, Republic of Korea. [2]Department of Biomedical Sciences, BK21 Plus Biomedical Science Project, Seoul National University College of Medicine, Seoul, Republic of Korea. [3]Convergence Research Center for Dementia, Seoul National University Medical Research Center, Seoul, Republic of Korea. ✉e-mail: buhm.han@snu.ac.kr

autosomes, where both sexes have chromosome pairs, females have two X chromosomes while males have only one. Furthermore, X chromosome inactivation (XCI), where one X chromosome in females is randomly inactivated for dosage compensation between sexes, provides key insights into understanding the biological mechanisms of the traits. Despite these important features, current methods overlook X chromosome contributions, significantly limiting our understanding of the complete genetic architecture of complex traits.

To address these limitations, we developed a genotype-free framework called BIGFAM. By extending the traditional PO-reg, BIGFAM analyzes variance components between relatives. This framework leverages phenotype data from relative pairs of first to third degree relationships, pursuing two key objectives: (1) partitioning variance components into genetic and shared environmental effects with a shared-environmental decay assumption reflecting the gradual decrease in environmental sharing across generations and (2) inferring X chromosome heritability using relationship-specific inheritance patterns. We applied BIGFAM to two large datasets: the Generation Scotland (GS:SFHS) and UK Biobank (UKB). Using only phenotype data without any genotype information from first to third degree relative pairs, we demonstrate three key findings. First, our genetic variance estimates strongly correlate with those from genotype-based methods. Second, we identify strong shared environmental effects in dietary-related phenotypes. Third, X chromosome heritability estimates show strong concordance with genotype-based methods and detect patterns of X chromosome inactivation, demonstrating that chromosome-specific analyses are possible using phenotype data alone.

## Results

### Overview of BIGFAM

Using only phenotypic data of relatives without genotype information, our model addresses two objectives. First, we disentangle variance components by genetic ($V_G$) and by shared environmental effects ($V_S$) by leveraging the degree of relatedness ($d$) information. Second, we infer the variance component by the X chromosome ($V_X$) using the familial relationships ($l$) information.

The key idea of the first objective is that for each degree of relatedness pair, the genetic and shared environmental effects are expected to decay at different rates. Genetic correlation between relatives decreases by a factor of 2 as degree of relatedness ($d$) increases[5,11]. Based on this pattern, we model shared environmental correlation also decay with $d$, but potentially at a different rate (Fig. 1a).

The intuition is that closer relatives are more likely to have a higher correlation on shared environmental factors. By analyzing these distinct decay patterns, we can differentiate the variance components by genetic and shared environmental effects.

Specifically, we begin by conducting familial relationship regression (FR-reg), extending parent-offspring regression (PO-reg). For each degree of relatedness ($d$), the coefficient of FR-reg ($\lambda_d$) can be decomposed into two parts: a genetic ($G$) and a shared environment part ($S$). Each part can be further decomposed as the product of genetic and shared environmental correlation ($r_{G,d}$ and $r_{S,d}$) between relatives and variance component by genetic and shared environment ($V_G$ and $V_S$). By comparing FR-reg coefficients across different degrees of relatedness, we can examine how genetic and shared environmental correlation decay. While genetic correlation is known to decrease by a factor of 2, we model shared environmental correlation to decrease by a factor of $w_S$ as $d$ increases.

Based on these decompositions, we can express the expected FR-reg coefficient as follows.

$$E[\lambda_d] = 2^{-d} \cdot V_G + w_S^{-d+1} \cdot V_S \tag{1}$$

When the decaying rate is identical in both genetic and shared environmental parts ($w_S = 2$), the equation simplifies to $E[\lambda_d] = 2^{-d} \cdot (V_G + 2V_S)$. In this case, it is statistically impossible to distinguish between $V_G$ and $V_S$ as they become a single combined term. However, if the decaying rate differs ($w_S \neq 2$), we can decompose these components using the FR-reg coefficient by analyzing how FR-reg coefficients change across different degrees of relatedness (see Methods).

The second objective infers the variance component by the X chromosome ($V_X$) without genotype data. The key idea is that for each familial relationship ($l$), the X chromosome inheritance pattern leads to distinct genetic correlations, even when their degree of relatedness ($d$) is constant (Fig. 1b). However, shared environmental correlations may also vary by relationship, potentially affecting the estimation of $V_X$. To address this, we model shared environmental effects into two components: relationship-specific correlation ($r_{S,l}$), which varies by relationship and variance component ($V_S$), which is constant within each degree of relatedness.

Specifically, we begin by conducting familial relationship regression (FR-reg) for each familial relationship ($l$). The coefficient of FR-reg ($\lambda_l$) can be decomposed into autosomal ($A$), X-chromosomal ($X$), and shared environmental ($S$) parts. To estimate X chromosome effects, we first group these coefficients by degree of relatedness and compute

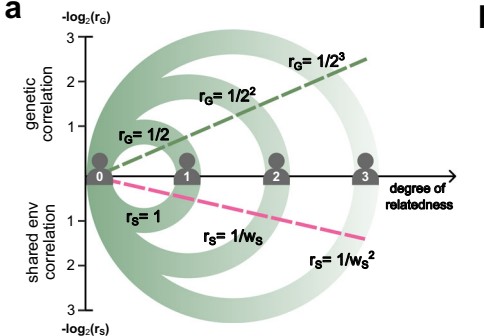
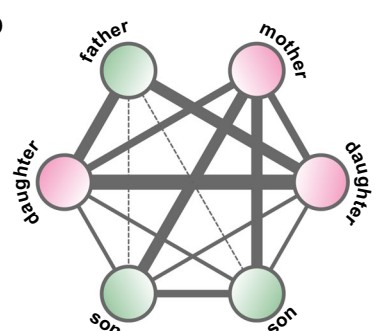

**Fig. 1 | Central idea of BIGFAM.** BIGFAM has two primary objectives: **a** partitioning variance components by genetic and shared environmental effects and **b** inference of variance component by X chromosome. **a** Schematic representation of the central idea for the first objective. It illustrates how genetic correlation ($r_G$) and shared environmental correlation ($r_S$) between relatives exhibit different rates of decay as degree of relatedness ($d$) increases. The genetic correlation ($r_G$, green dashed line) decreases by a factor of 2, while the shared environmental correlation ($r_S$, pink dashed line) follows a distinct decay pattern with a factor of $w_S$ as $d$

increases. **b** Visualization of the central idea for the second objective. It highlights that genetic correlations on the X chromosome ($r_{X,l}$) between relatives differ across various familial relationships ($l$), even within the same degree of relatedness (the first-degree in this case). Each node represents a family member (e.g., father, mother, daughter, son). The thickness of each line reflects the strength of genetic correlation on the X chromosome ($r_{X,l}$), and dashed lines denote relationships with $r_{X,l} = 0$.

their residuals ($\lambda_l^{res}$) by removing the group means. This step eliminates degree-specific effects while preserving relationship-specific patterns of both X chromosome and shared environmental effects. Then, since we know the X chromosome correlation for each relationship type but not the shared environmental patterns, we can treat the latter as a mean-zero error and estimate $V_X$ using the known X chromosome inheritance patterns.

Based on these decompositions, we can express the residual of FR-reg coefficient ($\lambda_l^{res}$) as:

$$E\left[\lambda_l^{res}\right] = t_l \cdot V_X + \epsilon_S \qquad (2)$$

where $t_l$ represents the deviation of relationship-specific X chromosome correlation from its degree-group mean and $\epsilon_S$ is the mean-zero shared environmental effect (see Methods). Using this expression, we can estimate X chromosome heritability ($V_X$) by regressing $\lambda_l^{res}$ on $t_l$. Since relationship-specific X chromosome correlations can be determined by the familial relationship information[11,12] this approach enables us to estimate X chromosome heritability ($V_X$) without requiring genetic information.

## Partitioning genetic and shared environmental effects in various shared environmental scenarios

Shared environmental effects manifest in various patterns across different degrees of relatedness. Some environmental factors might affect only nuclear families, others have maternal-specific effects or persist across multiple generations. To effectively partition genetic and shared environmental components, a method should be robust across these diverse scenarios. We evaluated BIGFAM's performance under various shared environmental patterns through comprehensive simulations.

BIGFAM employs a two-step approach to partition variance components without genotype: (1) a slope test to determine the decay pattern of shared environmental effects and (2) prediction of variance components based on the identified pattern. We assessed the performance of each step through simulations.

First, we evaluated the slope test, which determines whether genetic and shared environmental effects decay at different rates across degrees of relatedness. While genetic correlation decreases by a factor of 2 as degree of relatedness increases, shared environmental correlation may decay at a different rate, which we denote as $w_S$. To evaluate how effectively our slope test can identify different patterns of environmental decay, we simulated four scenarios where shared environmental effects exhibit distinct decay patterns: fast decay ($w_S > 2$), similar decay ($w_S \approx 2$), slow decay ($1 < w_S < 2$), and increase ($w_S < 1$). For each scenario, we generated 1,000 sets of simulated FR-reg coefficients (see Methods) by sampling genetic variance components ($V_G$) from 0.1 to 0.8 and shared environmental variance components ($V_S$) from 0 to 0.2.

The slope test classifies patterns based on whether the decay rate of shared environmental effects significantly differs from that of genetic effects. Specifically, using estimated slopes and 95% confidence intervals (CIs), the test first determines if the decay pattern is similar (interval includes 1) or different (interval excludes 1). When different, the pattern is further classified as fast decay if the lower bound exceeds 1, or slow decay if the upper bound is below 1. The reliability of this classification is confirmed by type 1 error and power analysis (Supplementary Fig. 1). While this classification focuses on the statistical difference in decay rates, our simulations revealed that cases where shared environmental effects increased with degree ($w_S < 1$) showed distinct negative slopes (Fig. 2a and Supplementary Fig. 1), allowing their additional identification. This classification of decay patterns guides the subsequent prediction step by determining the appropriate range for shared environmental decay rate, enabling more accurate estimation of variance components (see Methods).

Second, we assessed the prediction step. In this step, the genetic and shared environmental variance components are estimated with the shared environmental decay rate ($w_S$). To evaluate prediction performance, under different decay patterns (fast with $w_S = 5$, slow with $w_S = 1.3$, and similar with $w_S = 2$), we simulated four relationship-specific shared environmental scenarios: a reference scenario where shared environmental effects gradually decay with degree ($SC1$), a nuclear-family-specific scenario where shared environment only affects first-degree relatives ($SC2$), a maternal-effect scenario where mother-offspring pairs show stronger environmental sharing than other first-degree relationships ($SC3$), and a scenario where shared environmental effects are strongest in second-degree relatives ($SC4$). For all scenarios, we set genetic and shared environmental variance components as $(V_G, V_S) = (0.5, 0.1)$.

Results demonstrated robust prediction across scenarios, with each scenario offering unique insights into how different patterns of shared environmental effects are captured in our estimates. In the reference scenario ($SC1$), prediction showed robust performance across both fast and slow decay patterns but not in similar decay pattern (Fig. 2b and Supplementary Fig. 2), which is expected as genetic and shared environmental components become statistically indistinguishable when they decay at the same rate. In the nuclear-family-specific scenario ($SC2$), the $w_S$ showed the extremely large value ($w_S \approx 100$), indicating a sharp decline in shared environmental effects between first- and second-degree relatives, capturing the nuclear-family-specific nature of environmental sharing. In the maternal-effect scenario ($SC3$), genetic components remained accurately estimated ($V_G = 0.506$, SD = 0.010). Interestingly, the estimated shared environmental estimate was higher than the reference scenario ($V_S = 0.111$, SD = 0.005), indicating that our method captures the enhanced environmental sharing in mother-offspring relationships. Notably, this scenario showed varying performance across different decay patterns (Supplementary Fig. 3). While maternal effects consistently cause deviations in the observed decay rate between degrees of relatedness, the impact of these deviations varies by decay pattern. Under fast decay ($w_S > 2$), these deviations have minimal impact as they are small relative to the large baseline decay rate. Under similar decay ($w_S \approx 2$), these deviations actually improve our estimates by creating a slightly faster decay rate between first and second degrees, helping distinguish genetic from shared environmental components. However, under slow decay ($w_S < 2$), these deviations lead to substantially different decay rates between first-to-second and second-to-third degrees, violating our model's assumption of consistent decay patterns and resulting in less accurate estimates. Finally, in the second-degree-specific scenario ($SC4$), the prediction yielded large uncertainties in its estimates, demonstrated by wide confidence intervals (CIs including zero for all 1000 simulations). This suggests that reliable estimation may be challenging when shared environmental effects deviate significantly from the expected decay pattern.

## Partitioning variance components by genetic and shared environmental effects in real data

We applied BIGFAM to two datasets: Generation Scotland (GS:SFHS) and UK Biobank (UKB), analyzing relative pairs from first to third degrees (38,006 and 81,326 pairs from GS:SFHS and UKB). After adjusting phenotypes for sex, age, squared age, sex × age, and sex × squared age, we focused on 130 continuous phenotypes (24 and 106 from UKB and GS:SFHS) that showed significantly nonzero pairwise regression (FR-reg) coefficients ($p < 0.05$) across all degrees of relatedness.

To partition variance components, BIGFAM employs a two-step process: a slope test followed by a prediction step. In the slope test, we examine whether the variance components ($V_G$ and $V_S$) can be statistically distinguished or not. Specifically, if the slope in the slope test significantly deviates from 1, it indicates that the shared environmental

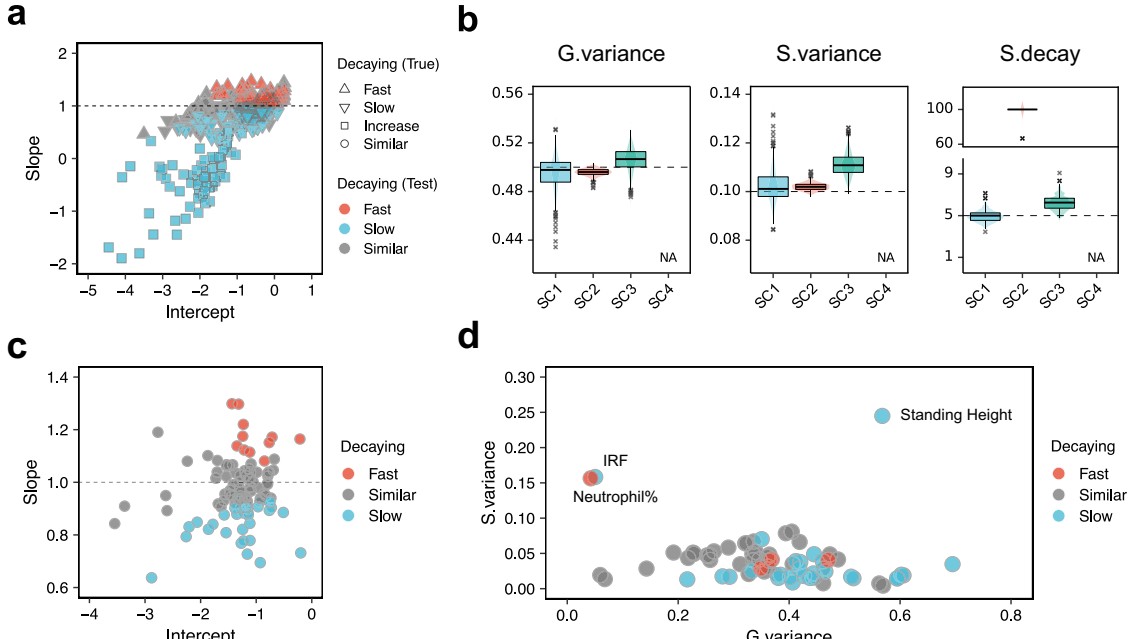

**Fig. 2 | Performance of BIGFAM in partitioning genetic and shared environmental effects. a** Results of the slope test in simulations. The scatter plot shows intercept and slope relationships. Shapes indicate true shared environmental decay patterns: fast (upper triangle), slow (lower triangle), similar (circle), and increase (square). Colors represent test results: fast (red), slow (blue), and similar (gray). The dashed line at slope = 1 indicates identical decay rates for genetic and environmental effects. **b** Prediction performance across four shared environmental scenarios: gradual decay ($SC1$), nuclear-family-specific ($SC2$), maternal-effect ($SC3$), and second-degree-specific ($SC4$). Box plots with overlaid violin plots show the distribution of the estimated genetic variance (*G.variance*), shared environmental variance (*S.variance*), and shared environmental decay rate (*S.decay*) across 1000 simulations. Simulations were performed assuming a standard error of 0.005 for FR-reg coefficients. Boxes represent interquartile ranges (IQR), center lines show medians, whiskers extend to $1.5 \times IQR$, and dots represent outliers. Dashed lines indicate true values ($V_G = 0.5$, $V_S = 0.1$, $w_S = 5$). "NA" indicates no significant estimates for $SC4$ (95% CIs included zero for all simulations). **c** Slope test results in UKB. The scatter plot shows the intercept and slope distribution. Colors indicate estimated shared environmental decay patterns: fast (red), slow (blue), and similar (gray). The dashed line at slope = 1 represents where genetic and shared environmental effects decay at the same rate. **d** Estimated variance components in UKB. The scatter plot shows the relationship between genetic variance (*G.variance*, x-axis) and shared environmental variance (*S.variance*, y-axis). Colors indicate decay pattern classification from panel (**c**). Labels highlight phenotypes with high shared environmental variance: standing height, immature reticulocyte fraction (IRF), and neutrophil percentage.

decaying parameter ($w_S$) is significantly different from the genetic decaying parameter ($w_G = 2$) (see Methods). In such cases, BIGFAM effectively distinguishes $V_G$ and $V_S$ in the subsequent prediction step.

The slope test demonstrated that 41% (53/130) of the phenotypes exhibited significant differences between their $w_S$ and $w_G$ (Fig. 2c for UKB and Supplementary Fig. 4 for GS:SFHS). To further analyze these phenotypes, we predict the variance components ($V_G$ and $V_S$) and shared environmental decaying factor ($w_S$) in the prediction step.

Among 130 phenotypes, we retained 81 phenotypes with significantly nonzero $V_G$ and $V_S$ (95% confidence intervals from resampling, excluding zero for both components). For these phenotypes, the average of predicted $V_G$ was 0.353(SD = 0.145), $V_S$ was 0.043(SD = 0.036), and shared environmental decaying factor ($w_S$) was 1.482(SD = 3.978). Note that $V_S$ estimate (0.043) from BIGFAM is consistent with previous findings that contribution of shared environment is ~0.08 within families[8] and 0.04 within twin data[4].

The results on shared environmental effects revealed several notable findings. First, we identified phenotypes with high $V_S$. The top three phenotypes were standing height (0.245, 95% CIs = [0.164, 0.398]), immature reticulocyte fraction (0.158, 95% CIs = [0.022, 0.164]), and neutrophil percentage (0.156, 95% CIs = [0.040, 0.161]). Notably, these traits are significantly influenced by nutritional status[13–15] which may explain their high shared environmental effect. In particular, the shared environmental effect for standing height (0.245) is consistent with previous twin studies, which report estimates around 0.3 and show age-dependent variation[16,17] ranging from 0.2 to 0.4.

Second, we identified phenotypes exhibiting nuclear-family-specific shared environmental effects. In the simulation study, we highlighted that $w_S$ was notably large ($w_S \approx 100$) in nuclear-family-specific shared environmental scenarios. Applying this insight to our real data analysis, we found that urate and whole body impedance showed $w_S \approx 100$. Previous studies have shown that both traits are influenced by various environmental factors, including dietary habits[18] and lifestyle patterns[19], and are closely associated with body fat composition[20,21]. While direct evidence for nuclear-family-specific environmental effects is limited, the complex nature of these multiple environmental factors and their tendency to be shared within nuclear families might explain our observation of nuclear-family-specific shared environmental patterns in these traits.

Third, we observed contrasting patterns in forced vital capacity (FVC) between cohorts: while genetic variance components remained consistent, shared environmental effects differed significantly. FVC showed significant (slow) decay of shared environmental effects in both cohorts, with similar genetic variance components ($V_G = 0.331$, 95% CIs = [0.327, 0.334] in GS:SFHS; $V_G = 0.333$, 95% CIs = [0.332, 0.334] in UKB) aligning with twin studies reporting heritability of 0.26 (0.03–0.49)[22]. While shared environmental effects varied substantially between cohorts ($V_S = 0.096$, 95% CIs = [0.0945, 0.098] in GS:SFHS and $V_S = 0.025$, 95% CIs = [0.025, 0.026] in UKB). Similar patterns occur in other traits as well, such as conduct disorder (CD), where studies have shown that while $V_G$ remains relatively stable across populations, $V_S$ tends to increase in recent cohorts[23]. These findings suggest that while genetic architecture might be preserved across populations, shared environmental effects can be cohort-specific, potentially due to demographic differences, even when the pattern of environmental decay remains consistent.

## Variance components in parent-offspring and sibling pairs

The variance components, especially shared environmental effects, may differ between parent-offspring and sibling pairs due to their distinct environmental sharing patterns. To investigate these patterns, we separated first-degree relative pairs into parent-offspring (PO) and sibling (SIB) pairs and computed variance components using BIGFAM. Specifically, we analyzed 9811 PO pairs (involving 12,762 individuals) and 8847 SIB pairs (involving 11,707 individuals) from the GS:SFHS dataset, as well as 4741 PO pairs (involving 8580 individuals) and 5663 SIB pairs (involving 11,017 individuals) from the UKB dataset. We retained 51 phenotypes with significantly nonzero $V_G$ and $V_S$ in both PO and SIB pairs (eight from GS:SFHS and 43 from UKB).

Among these 51 phenotypes, 35% (18/51) showed significantly different decay patterns between genetic ($w_G = 2$) and shared environmental effects ($w_S$) in both PO and SIB pairs. We classified these phenotypes into four distinct decay pattern groups based on slope test CIs. Three groups showed consistent patterns between PO and SIB pairs: the slow group of 17 phenotypes (five from GS:SFHS, 12 from UKB) with upper CI <1, the fast group of one phenotype from UKB with lower CI >1, and the similar group of 16 phenotypes (one from GS:SFHS, 15 from UKB) with CIs containing 1. The remaining different group comprised 17 phenotypes (two from GS:SFHS, 15 from UKB) showing distinct patterns between PO and *SIB* pairs. Notably, among possible combinations (fast-slow, fast-similar, slow-similar) between PO and SIB pairs, we observed none of the fast-slow cases. This suggests that shared environmental effects tend to maintain some consistency between PO and SIB relationships, rather than showing completely opposite decay patterns.

Variance component analysis revealed strong correlations between PO and SIB pairs (Supplementary Fig. 5). The genetic variance component ($V_G$) showed robust correlation ($r = 0.883$, 95% CI: [0.814, 0.936]), while the shared environmental variance component ($V_S$) showed a moderate but significant positive correlation ($r = 0.440$, 95% CI: [0.159, 0.682]). Notably, these correlations strengthened substantially when we restricted our analysis to phenotypes exhibiting consistent decay patterns between PO and SIB pairs (slow, fast, and similar, excluding different). In these cases, the $V_S$ correlation increased markedly from $r = 0.440$ to $r = 0.713$ (95% CI: [0.402, 0.911]), and $V_G$ correlation showed a modest improvement from $r = 0.883$ to $r = 0.916$ (95% CI: [0.839, 0.966]). These findings suggest two key insights: first, genetic effects remain stable across different familial relationships, and second, shared environmental effects, while more variable, show strong consistency between PO and SIB pairs when their decay patterns align.

Additionally, impedance of whole body showed remarkably large shared environmental decaying values ($w_S \approx 100$) in both relationship types. This phenotype was highlighted in the previous section for having notably large shared environmental decaying value, reinforcing our previous observation of nuclear-family-specific shared environmental effects in this phenotype.

## Variance by genetic effect from BIGFAM and other methods

To validate the variance components estimated by BIGFAM, we compared our genetic effect estimates ($V_G$) with those from eight other methods. We classified these methods into three distinct categories based on their input data requirements. The first category, summary-based methods, estimates genetic variance components (heritability) using GWAS summary statistics. This category includes LDSC[24] and LDpred[25]. The second category, genotype-based methods, leverages individual-level genotype data from relatives to estimate variance components. This category comprises relatedness disequilibrium regression (RDR)[26], GCTA, and Haseman–Elston (HE) regression. For GCTA analysis, we utilized the big-K small-K method[27] to address potential bias from shared environmental effects in relatives[28]. This method provides two distinct estimates (GCTA-snp and GCTA-ped)

(Supplementary Methods). For HE regression[28], we employed regression using the square difference of the phenotypes for pairwise individuals (HE-SD) and using the cross product of the phenotypes for pairwise individuals (HE-CP). The third category, pedigree-based methods, estimates variance components using only phenotype and relationship information without genotype data. This category includes BIGFAM and structural equation model (SEM)[8]. To ensure fair comparison, both genotype-based and pedigree-based methods were applied to the same set of first- to third-degree relative pairs (81,326 relative pairs for UKB and 38,006 relative pairs for GS:SFHS). Detailed descriptions of each method are provided in the Supplementary Methods.

It is important to note that SNP-based heritability is not directly comparable to family-based heritability estimates. SNP-based methods focus on the heritability captured by genotyped variants, which is considered a lower bound of narrow-sense heritability due to ungenotyped markers and imperfect linkage disequilibrium (LD)[29,30]. In contrast, family-based methods, like BIGFAM, can capture broader genetic influences, including those not tagged by SNPs. Therefore, while comparing these methods, it is crucial to understand that they provide insights into different aspects of genetic variance. However, if BIGFAM effectively partitions genetic effects from shared environmental effects, its $V_G$ estimates should show high correlation with other methods despite these systematic differences in magnitude. To test this hypothesis, we analyzed 40 phenotypes from the UK Biobank (UKB) that showed significantly nonzero $V_G$ across all methods. The correlations ($r^2$) between methods were estimated through 1000 bootstrap resampling.

The results showed that methods within the same category demonstrated high concordance in their estimates (Fig. 3 and Supplementary Fig. 6). Interestingly, BIGFAM showed stronger correlations with methods from other categories (summary and genotype-based methods; $r^2$ ranges from 0.729 to 0.791) than with SEM (pedigree-based method; $r^2 = 0.628$). When examining correlations with summary-based methods, BIGFAM achieved comparable levels to those observed with genotype-based methods (Fig. 3). Specifically, among the pairwise comparison, LDpred2's correlation with BIGFAM was the second highest ($r^2 = 0.804$, 95% CIs: [0.620, 0.891]), while its correlation with SEM was the lowest ($r^2 = 0.587$, 95% CI: [0.339, 0.761]). These results demonstrate that BIGFAM shows higher concordance with other methods compared to existing genotype-free methods. It suggests that BIGFAM successfully partitions genetic and shared environmental effects, even without using genotype data.

Similar patterns of high method concordance were observed in the GS:SFHS dataset (Supplementary Fig. 7). In this dataset, correlations between methods were notably stronger than in UKB (all comparison showing median $r^2$ values >0.95). However, due to the small number of phenotypes (six phenotypes showing significantly nonzero $V_G$ across all methods), the confidence intervals were substantially wider. For instance, the correlation between GCTA-snp and GCTA-ped showed a high median value ($r^2 = 0.959$) but with wide confidence intervals (95% CI: [0.020, 1.000]).

## Comparison of BIGFAM with REML and HE-regression

Existing methods such as REML[31,32] and Haseman–Elston regression[28] can adapt to situations where genotype data is unavailable by replacing genotype-based relatedness measures with pedigree-based information. In these pedigree-based approaches, genetic correlation is determined by the degree of relatedness of pair: unrelated pairs are assigned 0, while $d$-th degree relatives are assigned $2^{-d}$ based on expected genetic correlation. However, this approach fits phenotypes to a single genetic relatedness matrix (GRM). When applied to relatives, such fitting can lead to biased estimates of $V_G$ due to shared environmental effects[28].

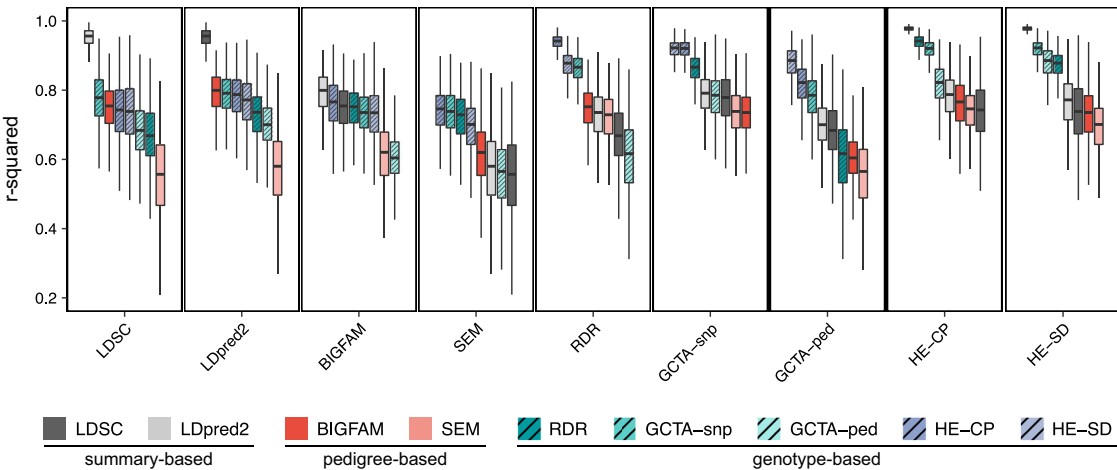

**Fig. 3 | Pairwise correlations between different methods for estimating genetic variance components.** Box plots show the distribution of correlations ($r^2$) between nine different methods for estimating genetic variance components, across $n = 40$ phenotypes from the UKB. Each panel compares the method on the x-axis with all other methods. Methods are categorized into three groups: summary-based methods (LDSC, LDpred2; gray), pedigree-based methods (BIG-FAM, SEM; red), and genotype-based methods (RDR, GCTA-snp, GCTA-ped, HE-CP, HE-SD; teal with diagonal patterns). Boxes represent the interquartile range (IQR), center lines show medians, whiskers extend to $1.5 \times$ IQR, and outliers have been omitted for clarity.

To address this issue, several methods have been proposed[2,26,27] that jointly fit additional information matrices to account for shared environments alongside SNP-based GRM. These methods effectively address the shared environmental issue when using SNP-based GRM. However, when using multiple relatedness matrices, substituting SNP-based GRM with pedigree-based GRM results in convergence issues, a problem not observed in single-GRM implementations (Supplementary Methods).

These convergence issues arise from three key factors. First, the incorporation of additional information matrices increases the number of parameters to be estimated, leading to numerical instability. Second, multicollinearity issues between the pedigree-based GRM and additional shared environmental matrices further complicate the estimation process. Specifically, the values in these matrices are structurally dependent: when a pair shows a specific degree of relationship in the GRM, it deterministically defines the values in all shared environmental matrices used in the analysis. Third, the inherent sparsity of the pedigree-based GRM contributes to this instability, as only a small fraction of pairs (those who are related) have nonzero values among all possible pairs of individuals. These factors collectively prevent the estimates from converging in both REML and HE regression implementations, as detailed in the Supplementary Methods.

In contrast, BIGFAM effectively addresses these limitations through two key innovations. First, BIGFAM utilizes summarized pairwise regression coefficients (FR-reg) for each degree of relationship (see Methods), unlike REML and HE regression, which use all pairwise information directly and are thus susceptible to the sparse nature of relatedness data. Second, it introduces a shared environmental decaying parameter ($w_S$), which not only reduces the number of parameters to estimate but also reduces the number of dependent matrices involved in the estimation process. This effectively mitigates both parameter estimation and multicollinearity problems present in existing methods. These features collectively enable stable and efficient variance component analysis, particularly in datasets with multiple degrees of relatives.

### Estimating variance component by the X chromosome in various shared environmental scenarios

To estimate variance components by X chromosome ($V_X$), BIGFAM utilizes a distinctive characteristic of X chromosome inheritance. Within each degree of relatedness, X chromosome effects vary substantially across specific familial relationships due to their unique inheritance patterns compared to autosomal effects and shared environmental effects. To leverage this property, BIGFAM computes FR-reg coefficients at the familial relationship ($l$) level, unlike the previous analysis, where we computed them for each degree of relatedness.

Using familial level FR-reg coefficients, BIGFAM follows a two-step approach to estimate $V_X$. In the first step, we group FR-reg coefficients by degree of relatedness ($d$) and compute residuals by subtracting the mean within each group. Since autosomal and shared environmental effects are expected to be similar within each degree group, this subtraction effectively removes these components while preserving the relationship-specific X chromosome effects (see Methods). The second step is a prediction step where we estimate $V_X$ by regressing these residuals on the X chromosome correlation of each familial relationship. However, this regression can produce wide confidence intervals, particularly when estimating small $V_X$ values (Supplementary Methods). To address this limitation, we incorporated an L2 penalty term into the loss function (see Methods).

The L2 penalty term helps improve the precision of our estimates, but its weight ($\alpha$) needs to be carefully chosen. As $\alpha$ increases, the variance of the estimate ($V_X$) decreases, but excessively large values can lead to underestimation of $V_X$. Through simulation studies comparing models with and without L2 penalty across a range of $\alpha$ values (from $-2$ to 4), we found that $\alpha = 2$ provided optimal results, effectively reducing CIs while maintaining estimation accuracy (Supplementary Fig. 8). Using this optimal penalty weight, we verified BIGFAM's performance across various combinations of variance components ($V_A$: 0.2 to 0.6, $V_X$: 0.001 to 0.02), confirming consistent and accurate estimation of $V_X$ across all parameter settings (Supplementary Fig. 9). For example, when the true $V_X$ was 0.005, the L2 penalty with $\alpha = 2$ reduced the width of 95% CIs by approximately 40% (0.004 with 95% CI: [$-0.018$, 0.030]) compared to the model without penalty (0.007 with 95% CI: [$-0.030$, 0.051]).

To further evaluate BIGFAM under different shared environmental scenarios, we tested the performance in estimating $V_X$ on four shared environmental scenarios. These scenarios included: a reference scenario with gradual environmental decay (SC1), a nuclear-family-specific scenario where shared environment only affects first-degree relatives (SC2), a maternal-effect scenario (SC3), and a scenario where shared environmental effects are strongest in second-degree relatives (SC4). For all scenarios, we set autosome, X chromosome, and shared

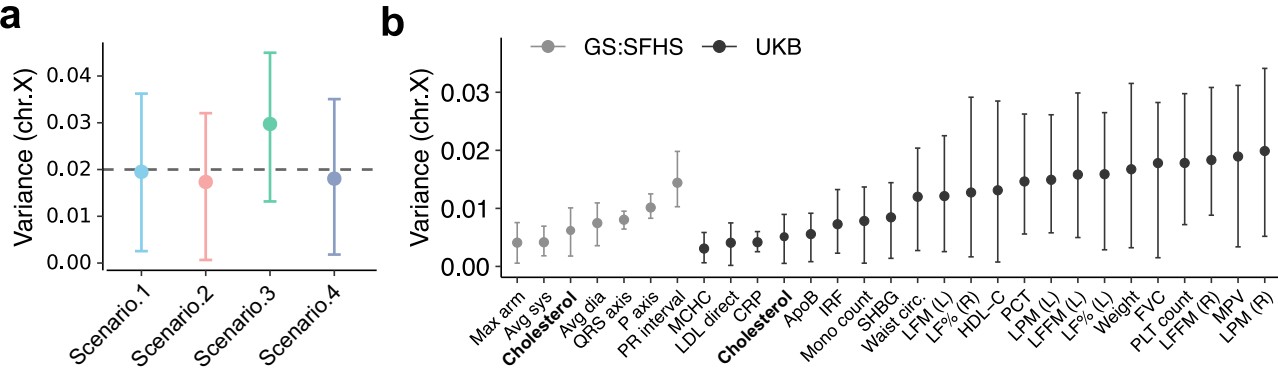

**Fig. 4 | Estimation of X chromosome variance components. a** Performance evaluation of BIGFAM's X chromosome variance component estimation across four shared environmental scenarios: reference scenario with gradual environmental decay (SC1), nuclear-family-specific scenario (SC2), maternal-effect scenario (SC3), and second-degree-specific scenario (SC4). For each scenario, points and error bars represent the median and 95% CI of estimates derived from 1000 resampling, assuming a standard error of 0.05 for FR-reg coefficients in each relationship. The black dashed line indicates the true $V_X$ value (0.02). **b** Estimated X chromosome variance components ($V_X$) for 29 phenotypes (7 from GS:SFHS and 22 from UKB). Points represent median estimates with 95% CIs, based on independent phenotype estimates from each cohort. Color represents cohort: GS:SFHS (gray) and UKB (black).

environmental variance components as $(V_A, V_X, V_S) = (0.4, 0.02, 0.2)$ (see Methods). BIGFAM demonstrated robust performance across most scenarios, accurately estimating the true $V_X$ value (Fig. 4a). In the maternal-effect scenario (*SC3*), we observed a slight overestimation (0.030 with 95% CIs: [0.013, 0.045]) compared to other scenarios. This overestimation likely occurs because maternal relationships tend to have both stronger shared environmental effects and higher X chromosome correlation compared to other relationships. For example, the X chromosome correlation of mother–son relationships ($\frac{1}{\sqrt{2}}$) is higher than father–son relationships (0). Since our model assumes these effects are independent, the correlation between them in maternal relationships can lead to slightly inflated $V_X$ estimates.

Additionally, we evaluated BIGFAM's robustness to relationship-specific shared environmental effects. Since shared environmental effects can vary substantially across different types of familial relationships, we simulated four different distributions of shared environmental correlation: normal, left-skewed (some relatives has weaker shared environmental effects), right-skewed (some relatives has stronger shared environmental effects), and uniform distributions (Supplementary Fig. 9). BIGFAM demonstrated consistent performance across all distributions, accurately estimating the true value ($V_X = 0.02$). While the exact values of median estimates and 95% confidence intervals showed slight variations across different distributions, all estimates remained close to the true value, confirming the method's robustness to different patterns of shared environmental effects (Supplementary Fig. 9).

**Variance by X chromosome in real data**
Using the Generation Scotland (GS:SFHS) and UK Biobank (UKB) dataset, we estimated the variance component by the X chromosome ($V_X$) using only phenotype data from relatives. In the GS:SFHS dataset, we identified 18,258 first-degree relative pairs with seven distinct familial relationships (involving 17,605 individuals), 15,114 second-degree relative pairs with 17 distinct familial relationships (involving 8253 individuals), and 4634 third-degree relative pairs with ten distinct familial relationships (involving 2686 individuals). In the UKB dataset, we identified 10,404 first-degree relative pairs with seven distinct familial relationships (involving 19,348 individuals), 2454 second-degree relative pairs with seven distinct familial relationships (involving 4743 individuals), and 14,754 third-degree relative pairs with eight distinct familial relationships (involving 27,069 individuals). In the UKB dataset, we leveraged marker-based kinship coefficients, age, and sex to infer familial relationships (see Methods). Phenotypes were regressed out by sex, age, squared age, sex × age, and sex × squared age.

The results of BIGFAM demonstrated that 29 phenotypes (22 phenotypes from UKB and seven phenotypes from GS:SFHS) showed significantly nonzero $V_X$ (Fig. 4b). The average X chromosome heritability was 0.011(SD = 0.005). Despite limited phenotype overlap between the two cohorts, cholesterol showed significant $V_X$ in both datasets with remarkably consistent estimates: 0.005 (95% CIs: [0.001, 0.009]) in UKB and 0.006 (95% CIs: [0.002, 0.010]) in GS:SFHS.

To further validate the consistency of BIGFAM, we compared its estimates with SNP heritability estimated from unrelated individuals in the same cohort (UKB). We used GCTA[31] to estimate X chromosome SNP heritability from unrelated individuals (Methods). Using BIGFAM with the optimal L2 weight ($\alpha = 2$) (Supplementary Fig. 10), we focused on 21 phenotypes that showed significantly nonzero $V_X$ in both approaches. The average X chromosome heritability estimates differed between methods: 0.013(SD = 0.005) from BIGFAM versus 0.005(SD = 0.003) from unrelated individuals (Fig. 5a). However, the strong correlation between these estimates ($r = 0.658$, 95% CIs: [0.397, 0.987]) supports the reliability of BIGFAM's results. The higher estimates from BIGFAM can be attributed to several factors, including its ability to capture additional genetic effects from pseudo-autosomal regions (PAR) and ungenotyped variants which are not in LD with genotyped variants[33–35].

We further extended our validation by examining sex-specific X chromosome heritability (Methods). Due to X chromosome inactivation, heritability on the X chromosome is expected to be higher in males than in females. To systematically compare sex-specific heritability estimates between BIGFAM and unrelated samples, we employed the dosage compensation ratio (DCR = $V_{X,\text{male}}/V_{X,\text{female}}$)[36], which quantifies the male-to-female ratio of X chromosome heritability. DCR provides a particularly useful metric as it has well-established theoretical expectations: under full dosage compensation (FDC), where one X chromosome in females is randomly inactivated, the DCR is 2, while under no dosage compensation (NDC), the DCR is 0.5.

The DCR comparison between BIGFAM and unrelated samples further supported BIGFAM's reliability. Among 24 phenotypes with significantly nonzero sex-specific $V_X$ in both approaches, the mean DCR values (1.882(SD = 0.504) from BIGFAM, 1.782(SD = 0.406) from unrelated samples) were not significantly different ($p$ value = 0.453) (Fig. 5b). The correlation between these DCR estimates was 0.637 (95% CI: [0.422, 0.885], based on 1000 resampling). Furthermore, DCR values remained close to the FDC value of 2 across different L2 weights ($\alpha$) (Supplementary Fig. 11), consistently indicating higher X chromosome heritability in males than females. These results demonstrate

# a

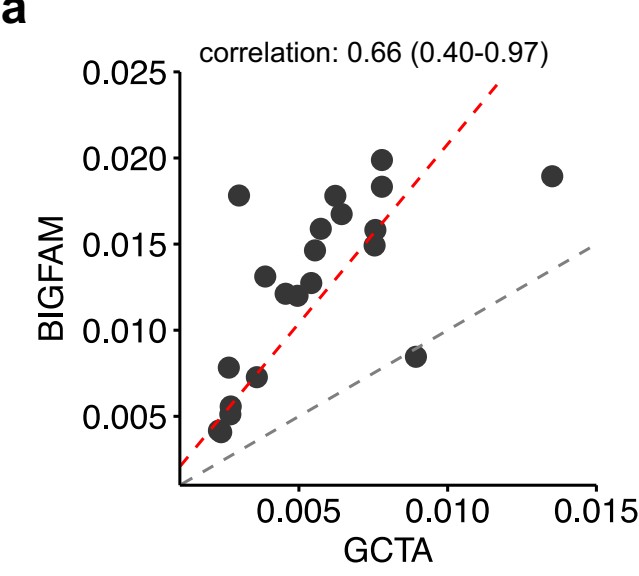

# b

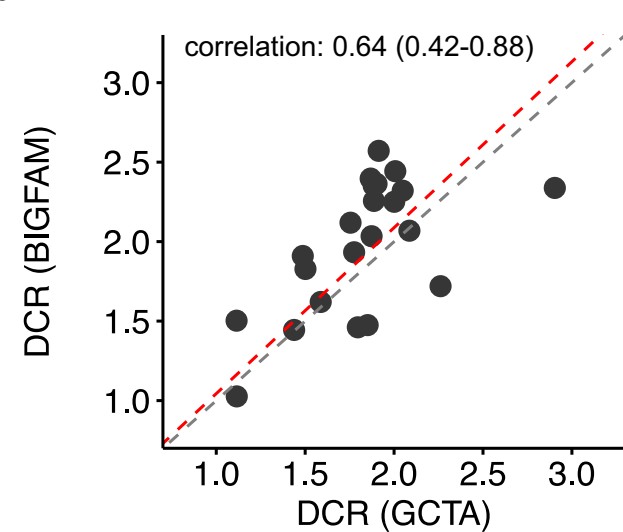

**Fig. 5 | Comparison of BIGFAM and GCTA estimates on X chromosome variance components. a** Comparison of X chromosome variance component ($V_X$) estimates between BIGFAM and GCTA for 21 phenotypes (black dots) with significantly nonzero estimates in both methods. The correlation of variance component estimates between BIGFAM and GCTA is 0.66 (95% CI: 0.40–0.97). Red dashed lines represent the regression line (without intercept), and gray dashed lines represent the reference ($y = x$) line. **b** Comparison of dosage compensation ratios (DCR = $V_{X,male}/V_{X,female}$) between BIGFAM and GCTA for 24 phenotypes (black dots) with significantly nonzero $V_X$ in both sexes. The correlation of variance component estimates between BIGFAM and GCTA is 0.64 (95% CI: 0.42–0.88). Red dashed lines represent the regression line (without intercept), and gray dashed lines represent the reference ($y = x$) line.

that BIGFAM can reliably detect both overall and sex-specific X chromosome heritability using phenotypic data alone, showing strong concordance with estimates using genotype data.

## Discussion

In this study, we introduce BIGFAM, a genotype-free framework designed to analyze variance components using only phenotypic and relationship information. This approach addresses limitations of traditional heritability estimation methods that rely on genetic data. By utilizing regression coefficients from each relationship (FR-reg) as

input, BIGFAM achieves two primary objectives. First, it partitions variance components into genetic ($V_G$) and shared environmental effects ($V_S$), incorporating a shared environmental decay parameter ($w_S$). Second, it estimates variance on the X chromosome leveraging relationship-specific genetic correlations on the X chromosome. Applied to real data from the Generation Scotland (GS:SFHS) and UK Biobank (UKB), BIGFAM revealed that 41% (53/130) of the phenotypes exhibit a distinct shared environmental decay parameter different from the genetic decay parameter. Notably, BIGFAM's genetic variance estimates showed stronger correlations with genotype-based methods ($r > 0.8$) than with other pedigree-based methods like SEM. We also identified phenotypes with high shared environmental effects, such as standing height (0.245). Additionally, BIGFAM's X chromosome heritability estimates showed a strong correlation ($r > 0.6$) with genotype-based X chromosome heritability estimates. These findings underscore that BIGFAM's ability to significantly enhance our understanding of genetic and environmental contributions to phenotypic variation, particularly in contexts where genotypic data is unavailable or incomplete.

BIGFAM offers distinct advantages over existing heritability estimation methods. Unlike genotype-based approaches[2,3,26,27] that depend on genotyped variants, BIGFAM operates without genetic data, significantly expanding its applicability. Compared to other genotype-free methods[1,5,6,8,37], BIGFAM makes two critical advances. First, it effectively partitions genetic and shared environmental components with stronger correlations to genotype-based methods than existing pedigree-based approaches. Second, it uniquely incorporates X chromosome analysis, addressing a significant gap in previous methodologies that focused solely on autosomes. This comprehensive approach provides insights into both autosomal and sex-linked genetic contributions, offering a more complete understanding of variance components across the entire genome.

BIGFAM, while innovative, has several limitations and raises questions that require further research. First, relationship information inferred from the UKB can be inaccurate. Despite using marker-based kinship coefficients, age, sex, and genetic correlation on the X chromosome, discrepancies between inferred and true relationships can arise. Additionally, heritability estimates may be affected by population stratification. BIGFAM does not directly account for genetic principal components for ancestry correction, which can lead to biased heritability estimates. Another limitation is the assumption about the consistency of shared environmental decay parameters ($w_S$) across degrees of relatedness. If $w_S$ varies by degree of relatedness, as shown in our simulations, the precision of heritability estimates in BIGFAM could be compromised. Furthermore, if shared environmental correlation aligns with genetic correlation on the X chromosome, estimates of X chromosome heritability from BIGFAM may be biased due to confounding factors. BIGFAM also currently lacks support for variance component analysis for binary traits. Heritability estimation for binary traits often relies on the liability threshold model, which may lead to biased estimates when applied to close relatives due to the influence of both additive and non-additive effects[38]. Therefore, further research is needed to develop methods for estimating heritability for binary traits using close relatives. Additionally, BIGFAM does not support bivariate analysis for computing the genetic correlation between traits. Unlike bivariate analysis with unrelated individuals, the phenotypes of close relatives are influenced by shared environmental effects. Consequently, the standard bivariate analysis model[39,40] cannot be directly applied to close relatives. Importantly, our current model cannot estimate the interaction between genetic and shared environmental effects. Recent studies have demonstrated that such interactions, particularly through genetic nurture effects, can significantly influence trait inheritance[41]. Future development of more sophisticated models could enable the estimation of these gene-environment interactions, along with genetic and shared

environmental correlations across multiple traits. We anticipate that further insights into shared environmental factors or the development of more sophisticated models could enable the estimation of genetic and shared environmental correlations across multiple traits. Lastly, BIGFAM does not account for assortative mating. Under assortative mating, the estimated genetic variance components can be inflated. Nevertheless, our model can still effectively partition the variance components by genetic and shared environmental effects. This is because, even under assortative mating, the variance component by genetic effect in the FR-reg coefficient ($\hat{\lambda}$) still decays as a factor of 2 (Supplementary Methods).

## Methods

This study was approved by the Institutional Review Board (IRB) at Seoul National University Hospital Biomedical Research Institute (IRB No. E-2302-003-1400). Access to individual-level data from the UK Biobank (Project ID 285388) and Generation Scotland: Scottish Family Health Study (GS23571) was obtained through their official data access procedures and conducted in accordance with their ethical guidelines. Information on participants' sex, age, and degree of relatedness was provided by the UK Biobank and Generation Scotland: Scottish Family Health Study. Sex information provided by the cohorts was used in the analysis, particularly for modeling sex-specific genetic variance components on the X chromosome. No additional sex or gender reassignment or determination was performed by the authors. Participant compensation followed the policies of the respective data providers.

### Traditional methods for estimating heritability

The variance component by genetic effects, commonly known as heritability, has traditionally been estimated using the shared genetic variance between relatives. One of the most basic approaches is parent-offspring regression (PO-reg), which estimates heritability by regressing offspring's phenotype values on their parents' phenotype values under an additive genetic model. We begin by explaining PO-reg as it forms the foundation for our extended framework.

The traditional additive genetic model describes the relationship between phenotype and genotype as $y = x^T\beta + \epsilon$. Here, $y$ represents a standardized phenotype value, and $x$ is a standardized genotype vector of size $M$, where $M$ denotes the number of causal variants. The effect size vector $\beta$ (of size $M$) follows a normal distribution $N(0, \frac{h^2}{M})$, where $h^2$ represents the variance by genetic effects (narrow-sense heritability). The environmental effect $\epsilon$ follows $N(0, 1 - h^2)$. Under this model, PO-reg estimates $h^2$ by analyzing the relationship between parent and offspring phenotypes. Specifically, when we regress offspring's phenotype value on parent's phenotype value ($y_{\text{offspring}} = \lambda \cdot y_{\text{parent}}$), the regression coefficient $\lambda$ multiplied by 2 provides an estimate of $h^2$. Below, we demonstrate that $2 \cdot E[\hat{\lambda}] = h^2$.

By the standard regression formula,

$$E\left[\hat{\lambda}\right] = E\left[\frac{Cov\left(y_o, y_p\right)}{Var\left(y_p\right)}\right] = \sigma\left(y_o, y_p\right) \quad (3)$$

where $y_o$ and $y_p$ denote the standardized phenotype of offspring and parent, respectively, and $\sigma(y_o, y_p)$ represents the population covariance between $y_o$ and $y_p$. Under the additive genetic model with independent environmental effects between relatives, $\sigma(y_o, y_p)$ can be expressed as

$$\sigma\left(x_o^T\beta + e_o, x_p^T\beta + e_p\right) = E\left[x_o^T x_p\right] \cdot E\left[\beta^T\beta\right] \quad (4)$$

since the environmental effects ($e_o, e_p$), genotypes ($x_o, x_p$), and effect sizes ($\beta$) are mutually independent. As offspring randomly inherits one of the alleles from the parent with a probability of 0.5, $E[x_o^T x_p] = \frac{1}{2}$.

Given that $\beta \sim N(0, \frac{h^2}{M})$, $E[\beta^T\beta] = h^2$. Therefore,

$$E\left[x_o^T x_p\right] \cdot E\left[\beta^T\beta\right] = \frac{M}{2} \cdot \frac{h^2}{M} = \frac{h^2}{2} \quad (5)$$

Consequently, the regression coefficient of PO-reg ($\hat{\lambda}$) is $\frac{h^2}{2}$, which is half of the heritability. Extending this derivation to $d$-degree relatives, the heritability estimated from the traditional method can be quantified as:

$$h^2_{\text{Trad}(d)} = 2^d \cdot \lambda_d \quad (6)$$

where $\lambda_d$ denotes the regression coefficient of PO-reg from $d$-degree relative pairs.

### Familial relationship regression (FR-reg)

As input to BIGFAM, we introduce Familial Relationship regression (FR-reg), which extends the parent-offspring regression (PO-reg) framework. The general framework of FR-reg is identical to PO-reg, wherein it regresses the phenotypes between relative pairs. However, when analyzing relative pairs, the choice of which relationship to use as the predictor versus the response can affect the results. For example, in a mother–son pair, using the mother's phenotype to predict the son's might yield different results than using the son's to predict the mother's.

To address this uncertainty and better capture the variance components between relatives, we include all possible predictor-response combinations within each relationship type in our regression analysis. Specifically, for a given relationship type *rel* (which can be either a degree of relatedness ($d$) or a specific familial relationship ($l$)), we model:

$$\begin{bmatrix} y_{m_1} \\ y_{m_2} \end{bmatrix} = \beta_0 + \lambda_{rel} \cdot \begin{bmatrix} y_{m_2} \\ y_{m_1} \end{bmatrix} \quad (7)$$

where $y_{m_1}$ and $y_{m_2}$ represent the standardized phenotypes of relative pair ($m_1, m_2$) and $\beta_0$ is the intercept. For instance, in parent-offspring pairs, both parent-to-offspring and offspring-to-parent directions are simultaneously considered in estimating $\lambda_{rel}$, ensuring our estimates capture the full extent of shared variance components between relatives.

Additionally, to minimize the potential bias from over-representation of certain families in the estimation process, we implement the following bootstrap resampling strategy. For each bootstrap iteration, we randomly select only one relationship per individual. This means if an individual has relationships with multiple relatives (e.g., a person who is both a parent and a sibling), only one of these relationships is randomly selected for each bootstrap iteration. This approach helps minimize potential effects from sample non-independence.

### Variance components in the FR-reg coefficient

To describe which variance components are captured in the FR-reg coefficient ($\lambda_{\text{rel}}$), we derive its expectation under the ACE model (A = additive genetic effects, C = shared (common) environment effects, E = unshared environment effects). In the ACE model, phenotype ($y$) is expressed as $y = x_G^T\beta_G + x_S^T\beta_S + \epsilon$ where $x_G$ is a standardized genotype vector of size $M_G$ (number of variants), and $\beta_G$ is its effect size vector with each component following $N(0, \frac{V_G}{M_G})$, where $V_G$ represents the variance by genetic effects. Similarly, $x_S$ represents standardized shared environmental factors of size $M_S$ with effect sizes $\beta_S \sim N(0, \frac{V_S}{M_S})$, where $V_S$ denotes the variance by shared environmental effects. The remaining environmental effect $\epsilon$ follows $N(0, 1 - V_G - V_S)$.

Under this model, the expected value of FR-reg coefficient ($E[\lambda_{rel}]$) can be decomposed into genetic ($G$) and shared

environmental (S) parts:

$$E[\lambda_{rel}] = G + S$$
$$= E[r_{G,rel}] \cdot V_G + E[r_{S,rel}] \cdot V_S \quad (8)$$

where $r_{G,rel}$ and $r_{G,rel}$ represent the genetic and shared environmental correlation between relatives with relationship *rel*, respectively.

## Objective 1. Partitioning variance components by genetic and shared environmental effects

To partition the variance components ($V_G$ and $V_S$) in FR-reg coefficients, we first conduct FR-reg for each degree of relatedness ($d$). As previously derived, the regression coefficient of FR-reg comprises two parts: the genetic part ($G$) and the shared environmental part ($S$). In the genetic part $G$, the genetic correlation between $d$-degree relatives ($r_{G,d}$) decreases by a factor of 2 as $d$ increases. In the shared environmental part ($S$), we assume that the shared environmental correlation between $d$-degree relatives ($r_{S,d}$) decays by a factor of $w_S$ as $d$ increases. Thus, $r_{S,d}$ can be expressed as $w_S^{-d+1}$. Based on this, the expectation of FR-reg coefficient is formulated as follows:

$$E\left[\hat{\lambda}_d\right] = G + S$$
$$= E[r_{G,d}] \cdot V_G + E[r_{S,d}] \cdot V_S \quad (9)$$
$$= 2^{-d} \cdot V_G + w_S^{-d+1} \cdot V_S$$

where $\hat{\lambda}_d$ represents the observed FR-reg coefficient from $d$-degree relative pairs.

**Step 1. Slope test**. In this step, we test whether genetic and shared environmental effects exhibit different decay patterns as degree of relatedness ($d$) increases. According to fundamental genetic inheritance patterns, genetic correlation decreases by a factor of 2 as degree of relatedness increases. If shared environmental correlation follows the same pattern ($w_S = 2$), both components would decay identically, making them statistically indistinguishable. However, if shared environmental effects decay at a different rate ($w_S \neq 2$), we can effectively decompose these components using FR-reg coefficients from multiple degrees of relatedness.

To test this, we formulate our null hypothesis as $H_0 : w_S = 2$, which represents the case where shared environmental correlation decays at the same rate as genetic correlation. Under this null hypothesis, the expected FR-reg coefficient can be expressed as $E[\lambda_d] = 2^{-d} \cdot (V_G + 2V_S)$. Taking the logarithm (base 2) of both sides yields:

$$\log_2 \lambda_d = 1 \cdot - d + \log_2(V_G + 2 \cdot V_S) \quad (10)$$

This equation indicates that under the null hypothesis, regressing $\log_2 \lambda_d$ on negative degree of relatedness ($-d$) should yield a coefficient of 1. Therefore, we test the null hypothesis by evaluating whether the regression coefficient estimates contain 1 within their 95% confidence intervals (CIs). These CIs are computed using resampled FR-reg coefficients. Rejection of this null hypothesis indicates distinct decay patterns between genetic and shared environmental effects, enabling us to estimate $V_G$ and $V_S$ separately in the next step.

**Step.2 Prediction variance components by genetic and shared environmental effects**. Based on the results from Step 1, we now aim to estimate their respective contributions ($V_G$ and $V_S$) and the rate of shared environmental decay ($w_S$). These parameters are estimated by finding the combination that best explains the observed FR-reg coefficients across multiple degrees of relatedness.

To find the optimal estimates, we define a loss function that measures the discrepancy between observed FR-reg coefficients ($\hat{\lambda}_d$)

and their predicted values ($\lambda_d$) under the ACE model:

$$Loss(V_G, V_S, w_S) = \sum_{d=1}^{D} \left( \log_2 \hat{\lambda}_d - \log_2 \lambda_d \right)^2 \quad (11)$$

where $D$ represents the set of degree of relatedness used in our analysis. By minimizing this loss function, we can identify the values of $V_G$, $V_S$, and $w_S$ that best explain the observed FR-reg coefficients.

To find optimal parameters, we employ two additional strategies:

- **Grid Search for $w_S$**: Based on the slope test results from Step 1, we systematically search for the optimal shared environmental decay rate ($w_S$). The search range for $w_S$ is determined by the results of the slope test. When the slope is significantly less than 1, we compute the loss function for $1/w_S$ in a range from 0.55 to 0.9 ($w_S$ from 1.82 to 1.11) with a 0.01 increment. When the slope is not significantly different from 1, we compute the loss function for $1/w_S$ in the range from 0.4 to 0.6 ($w_S$ from 1.67 to 2.5) with a 0.01 increment. When the slope is significantly larger than 1, we compute the loss function for $1/w_S$ in the range from 0.01 to 0.45 ($w_S$ from 2.22 to 100) with a 0.01 increment.

- **Cross-Validation**: To ensure our estimates are robust and not overfitted, we repeat the following process ten times: First, we randomly partition our resampled FR-reg coefficients into ten groups. For each group serving as a test set, we estimate $V_G$ and $V_S$ using the remaining groups as training data for each candidate $w_S$ value. We then select the parameter combination ($V_G$, $V_S$, $w_S$) that yields the minimum loss on the test set. This process generates 100 sets of parameter estimates (10 repeats × 10 folds), from which we compute the final estimates and their confidence intervals.

## Finding X chromosome heritability

To infer the variance components by X chromosome ($V_X$) from FR-reg coefficients, we first conduct FR-reg on each familial relationship ($l$). As previously derived, the regression coefficient of FR-reg comprises two parts: the genetic part ($G$) and the shared environmental part ($S$). To infer $V_X$, we split the genetic part ($G$) into two additional parts: the autosomal part ($A$) and the X chromosome part ($X$). The autosomal part ($A$) consists of genetic correlation on the autosome between the $l$-relationship ($r_{A,l}$) and the variance component by autosome ($V_A$). Notably, $r_{A,l}$ remains constant for all $l$-relationships within the same degree of relatedness. Similarly, the X chromosome part ($X$) also consists of genetic correlation on the X chromosome between the $l$-relationship ($r_{X,l}$) and the variance components by X chromosome ($V_X$). The shared environmental part ($S$) consists of shared environmental correlation between the $l$-relationship ($r_{S,l}$) and the variance component by shared environment ($V_S$). As the shared environment can exhibit variability across familial relationships, we model $r_{S,l}$ as a random variable. This variable is assumed to be independent and identically distributed for all $l$ within the same degree of relatedness. Meanwhile, $V_S$ remains constant for all $l$ within the same degree of relatedness.

With this decomposition, the expectation of the FR-reg coefficient is formulated as follows:

$$E\left[\hat{\lambda}_l\right] = A + X + S$$
$$= E[r_{A,l}] \cdot V_A + E[r_{X,l}] \cdot V_X + E[r_{S,l}] \cdot V_S \quad (12)$$

It is important to note that the genetic correlation ($r_{A,l}$ and $r_{X,l}$) are known for a given relationship $l$[11,12]. The derivation of $r_{X,l}$ for a certain familial relationship can be found in Supplementary Methods.

**Step 1. Residual of familial relationship regression coefficient**. In this step, we simplify the model parameters for subsequent analyses by

removing the mean effects within each degree of relatedness from FR-reg coefficients. Specifically, we stratify the FR-reg coefficients into bins based on the same degree of relatedness and regress out the average of FR-reg coefficients from the individual FR-reg coefficients ($\hat{\lambda}_l$) within each bin.

The average of FR-reg coefficients in the $d$-bin ($\lambda_d$) is computed as follows.

$$\lambda_d = \frac{1}{|L_d|} \sum_l^{L_d} \hat{\lambda}_l \tag{13}$$

where $L_d$ is the set of familial relationships in the $d$-bin and $|L_d|$ is the number of relationships in $L_d$. For instance, for $d = 1$, $L_d$ could include father–daughter, mother–son, sister, and brother relationships, resulting in $|L_d| = 4$.

Next, we formulate the expectation of residual of FR-reg coefficient for $l$ relation ($\hat{\lambda}_l^{res}$) as follows:

$$
\begin{aligned}
E\left[\hat{\lambda}_l^{res}\right] &= E\left[\hat{\lambda}_l\right] - E\left[\lambda_d\right] \\
&= \left(E[r_{A,l}] \cdot V_A + E[r_{X,l}] \cdot V_X + E[r_{S,l}] \cdot V_S\right) \\
&\quad - \left(E[r_{A,d}] \cdot V_A + E[r_{X,d}] \cdot V_X + E[r_{S,d}] \cdot V_S\right) \\
&= t_l \cdot V_X + \epsilon_S
\end{aligned}
\tag{14}
$$

where $t_l$ represents $E[r_{X,l}] - E[r_{X,d}]$ and $\epsilon_s$ is $(E[r_{S,l}] - E[r_{S,d}]) \cdot V_S$.

It is important to note that the variance component by autosome ($V_A$) is canceled out because the genetic correlation on the autosome is constant within each bin (i.e., $E[r_{A,l}] = E[r_{A,d}]$ for all pairs $l$ in degree $d$). In contrast, the X chromosome genetic correlation ($r_{X,l}$) varies within each bin depending on the specific type of relationship (e.g., mother–daughter has $r_{X,l} = 0.5$ vs father–daughter has $r_{X,l} = 1/\sqrt{2}$), even among relatives of the same degree. Additionally, due to our assumption that $r_{S,l}$ follows a distribution with a mean of $r_{S,d}$, the expected value of $\epsilon_S$ is zero. Hence, we can regard the variance component by relationship-specific shared environmental effects as a mean-zero error term. Since $r_{X,l}$ is well-documented[11,12], $t_l$ can be specified based on the relationship used in the analysis. Consequently, we can infer the variance component by X chromosome ($V_X$) by using only FR-reg coefficient.

**Step2. Prediction variance components by the X chromosome.** To compute a stable estimate ($V_X$) from $\hat{\lambda}_l^{res}$, we incorporate an additional strategy called the L2 penalty. It is well known that the optimization with L2 penalty shrinks the standard deviation of the estimate[42]. To achieve this, we establish the following loss function:

$$Loss(V_X|\alpha) = \sum_l^L \left(\hat{\lambda}_l^{res} - t_l \cdot V_X\right)^2 + \left(\frac{1}{h_{Trad}^2}\right)^\alpha \cdot (V_X)^2 \tag{15}$$

Here, $L$ represents the set of familial relationships used in BIGFAM, and $\alpha$ denotes the weight on the L2 penalty. The purpose of introducing $\alpha$ is to shrink the standard error of model parameters. The $h_{Trad}^2$ represents the heritability estimated from FR-reg coefficients. $h_{Trad}^2$ is calculated as inverse-variance weighting the estimate of $2^d \cdot \lambda_l$ in each familial relationship.

**Finding X chromosome heritability in males and females**
To achieve this, we slightly modified the variance component by X chromosome ($V_X$) in FR-reg coefficients with three X chromosome-related parameters: variance component by X chromosome in males ($V_{X,\text{male}}$), in females ($V_{X,\text{female}}$), and genetic correlation of X chromosome between males and females ($\rho_{X,mf}$).

To alleviate the effect of sex-specific variance components, in the first step, we stratified the FR-reg coefficients into bins based not only on the same degree of relatedness but also on the same sex pairs.

Within each bin, FR-reg coefficients ($\hat{\lambda}_l$) were regressed out by the average of FR-reg coefficients as follows:

$$\lambda_{d,ss} = \frac{1}{|L_{d,ss}|} \sum_l^{L_{d,ss}} \hat{\lambda}_l \tag{16}$$

where $L_{d,ss}$ is the set of familial relationships in the $d$-bin and same sex pair ($ss$). $ss$ can be male-male, female–female, and male–female pair.

Then, the expectation of residual of FR-reg coefficient for $l$ relation ($\hat{\lambda}_l^{res}$) can be formulated as follows:

$$
\begin{aligned}
E\left[\hat{\lambda}_l^{res}\right] &= E\left[\hat{\lambda}_l\right] - E\left[\lambda_{d,ss}\right] \\
&= \left(E[r_{A,l}] \cdot V_A + E[r_{X,l}] \cdot V_X + E[r_{S,l}] \cdot V_S\right) \\
&\quad - \left(E[r_{A,d,ss}] \cdot V_A + E[r_{X,d,ss}] \cdot V_X + E[r_{S,d,ss}] \cdot V_S\right) \\
&= t_l \cdot V_X + \epsilon_S
\end{aligned}
\tag{17}
$$

where $t_l$ represents $E[r_{X,l}] - E[r_{X,d,ss}]$ and $\epsilon_S$ is $(E[r_{S,l}] - E[r_{S,d,ss}]) \cdot V_S$.

In the second step, the loss function can be represented as follows.

$$Loss(V_X|\alpha) = \sum_l^L \left(\hat{\lambda}_l^{res} - t_l \cdot V_X\right)^2 + \left(\frac{1}{h_{Trad}^2}\right)^\alpha \cdot (V_X)^2 \tag{18}$$

where $V_X$ is $V_{X,\text{male}}$ if $l$ relationship is male pairs, is $V_{X,\text{female}}$ if $l$ relationship is female pairs, and is $\rho_{X,mf} \cdot \sqrt{V_{X,\text{male}} \cdot V_{X,\text{female}}}$ if $l$ relationship is male–female pairs. $\rho_{X,mf}$ is the genetic correlation of X chromosome between males and females. To make the square-root term positive, we constrain the scope of $V_{X,\text{male}}$ and $V_{X,\text{female}}$ within $1 \cdot 10^{-6}$ to 1. To find optimal parameters robustly, we grid search the X chromosome heritability while varying $\rho_{X,mf}$ in a range from $-1$ to 1 with 0.1 increment. In each value of $\rho_{X,mf}$, we select the parameters (($V_{X,\text{male}}, V_{X,\text{female}}, \rho_{X,mf}$)) which show the minimum loss function value.

**FR-reg coefficients generation in simulation**
To partitioning genetic and shared environmental variance components from FR-reg coefficients from a simulation, we generate the FR-reg coefficients for $d$-degree relatives ($\hat{\lambda}_d$) based on three true parameters ($V_G$, $V_S$, and $w_S$).

$$\hat{\lambda}_d = 2^{-d} \cdot V_G + w_S^{-d+1} \cdot V_S + \epsilon \tag{19}$$

where $V_G$ is the variance component by genetic effects, $V_S$ is the variance component by share environmental effects within first-degree relatives, $w_S$ is the shared environmental decaying factors, and $\epsilon$ is the error term following mean-zero normal distribution.

To infer the X chromosome heritability from FR-reg coefficients in a simulation, we generate the FR-reg coefficients for $l$ relationship ($\lambda_l$) based on three true parameters ($V_A$, $V_X$, and $V_S$).

$$\hat{\lambda}_l = 2^{-d} \cdot V_A + r_{X,l} \cdot V_X + r_{S,l} \cdot V_S + \epsilon \tag{20}$$

where $d$ is the degree of relatedness of $l$ relationship, $V_A$ is the variance component by autosomes, $V_X$ is the variance component by X chromosome, and $V_S$ is the variance component by shared environment. $r_{X,l}$ is the genetic correlation on the X chromosome on $l$ relationship pair and $r_{S,l}$ is the shared environmental correlation on $l$ relationship pair. $\epsilon$ is the error term which follows mean-zero normal distribution.

**FR-reg coefficients generation in four shared environmental scenarios**
To evaluate our method's performance under different shared environmental patterns, we simulated FR-reg coefficients for four distinct

scenarios. For each scenario, we generated coefficients by combining genetic effects ($V_G$) and scenario-specific shared environmental effects ($V_S$). The genetic component was modeled consistently across all scenarios, with genetic correlation decreasing by a factor of 2 as the degree of relatedness increases.

In the reference scenario (SC1), shared environmental effects followed a standard decay pattern where the correlation decreases by a factor of $w_S$ with each degree. For the nuclear-family-specific scenario (SC2), shared environmental effects were present only in first-degree relatives and set to zero for more distant relationships. The maternal-effect scenario (SC3) modeled enhanced environmental effects in mother-offspring relationships, where these relationships showed 1.5-fold stronger environmental correlation compared to other first-degree relationships. In the second-degree-dominant scenario (SC4), environmental effects were amplified (threefold) specifically for second-degree relatives while maintaining standard levels for first and third-degree relatives.

For each scenario, FR-reg coefficients were generated using the formula $\lambda_d = V_G \cdot (1/2)^d + V_S \cdot (1/w_S)^{d-1}$, where $d$ represents the degree of relatedness. The coefficients were simulated with a standard error of 0.005, and we performed 100 bootstrap resamples for each coefficient to account for estimation uncertainty.

### Inference of the relatives from the UK Biobank

The UK Biobank (UKB) dataset contains information on 81,326 relative pairs (123,418 individuals, Supplementary Fig. 12) and their kinship coefficients[43].

**Inference of the degree of relatedness.** The degree of relatedness of each relative pair was inferred using the marker-based kinship coefficients. We determined the first, second, and third-degree relatives whose kinship coefficients were within the range of $(0.4, 0.6)$, $(0.2, 0.3)$, and $(0.10, 0.15)$, respectively (Distribution of kinship coefficients are shown in Supplementary Fig. 13).

With these criteria, we identified 27,365 pairs (49,083 individuals) of first-degree relatives, 9447 pairs (17,240 individuals) of second-degree relatives, and 44,514 pairs (72,944 individuals) of third-degree relatives.

**Inference of the familial relationship.** To identify familial relationships within each degree of relatedness, we employed specific criteria (age difference, sex, and genetic correlation on the X chromosome) as follows. With these criteria, we identified 7, 7, and 8 familial relationships, respectively (Supplementary Table 1).

Specifically, for the first-degree relatives, we identified seven relationships (10,404 pairs with 19,348 individuals). If the age difference of the pair of relatives was larger than 20 and smaller than 28, we identified them as a parent–offspring relationship. If the age difference of the pair of relatives was larger than 1 and smaller than 5, we identified them as sibling relationship. After that, sex was used to specify the familial relationship.

For the second-degree relatives, we identified seven relationships (2454 pairs with 4743 individuals). We assumed that the possible second-degree relationships were only offspring's parent's sibling relationship, because the majority of age are distributed within 40 to 70 in the UKB. Therefore, we left relative pairs where the age difference was larger than 15 and smaller than 29. To identify specific familial relationships, we used genetic correlation on the X chromosome. Specifically, based on the X chromosome genetic correlation on the $l$-relationship ($r_{X,l}$), we inferred $l$-relationships where the $r_{X,l}$ was within $0.8 \cdot E[r_{X,l}]$ and $1.2 \cdot E[r_{X,l}]$. If $E[r_{X,l}]$ is 0, we inferred $l$-relationships where the $r_{X,l}$ was within $-0.05$ and $0.05$. After that, sex was used to specify the familial relationship.

For the third-degree relatives, we identified eight relationships (14,754 pairs with 27,069 individuals). We assumed that the possible

third-degree relationships were only offspring's parent's sibling's offspring relationship, because the majority of age are distributed within 40 to 70 in the UKB. Therefore, we left relative pairs where the age difference was smaller than 5. To identify specific familial relationships, we used genetic correlation on the X chromosome. Specifically, based on the X chromosome genetic correlation on the $l$-relationship ($r_{X,l}$), we inferred $l$-relationships where the $r_{X,l}$ was within $0.8 \cdot E[r_{X,l}]$ and $1.2 \cdot E[r_{X,l}]$. If $E[r_{X,l}]$ is 0, we inferred $l$-relationships where the $r_{X,l}$ was within $-0.05$ and $0.05$. After that, sex was used to specify the familial relationship.

### Inference of the relatives from Generation Scotland: Scottish Family Health Study

Scotland: Scottish Family Health Study Generation Scotland: Scottish Family Health Study (GS:SFHS) dataset contains pedigree information recruited from 4830 families. It includes 40,254 first to fourth-degree relative pairs (18,236 individuals, Supplementary Fig. 12)[44].

**Inference of the degree of relatedness.** The degree of relatedness of each relative pair was inferred based on pedigree information. Specifically, the first-degree relatives (18,258 pairs and 17,605 individuals) include parent-offspring and sibling relationships. The second-degree relatives (15,114 pairs and 8253 individuals) include avuncular, grandparent, and half-sibling relationships. The third-degree relatives (4634 pairs and 2686 individuals) include first cousins, half-avuncular, and great-grandparent relationships. We exclude fourth-degree relatives because of a small sample size (75 relative pairs).

**Inference of the familial relationship.** To identify specific familial relationships within each degree of relatedness group, we employed sex and pedigree information. For each degree of relatedness, we filtered out familiar relationships which has too small relative pairs (<100). As a result, we identified seven familial relationships in first-degree, 17 familial relationships in second-degree, and ten familial relationships in third-degree relatives (Supplementary Table 2).

### Genetic relatedness matrix for SNP heritability

The genetic relatedness matrix (GRM) is used in various SNP-based heritability estimation methods. The GRM is computed using GCTA software[31] with the following QC workflow. Using PLINK 2.0[45], we used SNPs with non-duplicated rsID, hard-called variants with imputation information (INFO) score >0.4, missing genotype rate <0.1, minor allele frequency (MAF) >0.01 and Hardy–Weinberg equilibrium (HWE) $p > $1e-6, as computed in European ancestry individuals. For the X chromosome, we used variants in the non-pseudo-autosomal region (NPX) (27–155 Mb, hg19) because the PAR has different properties compared with the non-PAR (NPX) region. After the QC workflow, 8,389,798 variants in the autosome and 269,956 variants in the X chromosome remained.

### Variance by genetic effect from other methods

To validate BIGFAM's estimates, we compared our results with eight other methods across three categories: summary-based methods (LDSC from https://nealelab.github.io/UKBB_ldsc and LDpred2 using *bigsnpr v.1.12*), genotype-based methods (RDR from https://github.com/AlexTISYoung/RDR, GCTA-snp, GCTA-ped, HE-CP, and HE-SD using *GCTA v.1.94*), and pedigree-based methods (BIGFAM and SEM using *statsmodels v.0.14.2*). For fair comparison, all methods analyzing relatives used the same set of first- to third-degree relative pairs from UK Biobank (81,326 pairs) and GS:SFHS (38,006 pairs). Detailed descriptions of each method are provided in the Supplementary Methods.

### Reporting summary

Further information on research design is available in the Nature Portfolio Reporting Summary linked to this article.

## Data availability

All data underlying the figures in this paper are provided in the Source Data. The BIGFAM software used in this study is available at https://github.com/jerrylee9310/BIGFAM. Summary statistics from LDSC are available at https://nealelab.github.io/UKBB_ldsc/index.html. UK Biobank data is available at https://www.ukbiobank.ac.kr. Generation Scotland: Scottish Family Health Study data is available at https://genscot.ed.ac.uk/. Access to these datasets requires institutional approval in accordance with their data access policies to ensure participant privacy and compliance with ethical and legal standards. Source data are provided with this paper.

## Code availability

Code for BIGFAM is available at https://github.com/jerrylee9310/BIGFAM and archived with https://doi.org/10.5281/zenodo.15386299.

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

## Acknowledgements

This work was supported by the National Research Foundation of Korea (NRF) (Grant number RS-2025-00553579 to B.H.) funded by the Korean government, Ministry of Science, and ICT. This work was also supported by the Creative-Pioneering Researchers Program (to B.H.) funded by Seoul National University and by the AI-Bio Research Grant through Seoul National University.

## Author contributions

J.J.L. conceived the study, performed the analyses, and wrote the manuscript. B.H. supervised the research and contributed to manuscript revisions.

## Competing interests

Buhm Han is the CEO of SpintoAI Inc., which had no commercial interests in outcomes of this study. The remaining author (J.J.L.) declares no competing interests.
