## [Peer Review file · Nature Communications]

BIGFAM - variance components analysis from relatives without genotype

Corresponding Author: Professor Buhm Han

Version 0:

Reviewer comments:

Reviewer #1

(Remarks to the Author)

The authors present a study of a new approach, X-CHROM, to estimate genetic variance due to the X and autosomes separately, and due to shared environmental variance. They apply the method to familial data from the UK Biobank, and compare estimates to those from GREML. The methods nicely outline the approach, including the expectations of allele sharing based on different relationships, and the motivation for the analyses. The possibility of applying it to large EHR datasets in which genotype data is not available but known relationships are is appealing, and that would be a nice addition to the paper.

Several questions arose during my review:

Could the approach be expanded to account for more distant relatives? Grandparents or avuncular relationships when known? If the maternal or paternal grandparents are known, could one include other relatives to incorporate additional information and even additional sources of variance (e.g., dominance variance or other familial environment variance)?

Furthermore, could the approach be extended to a bivariate analysis to estimate partitioned genetic correlation of traits due to autosomes vs. X?

A relatively minor concern, because it is only an issue at smaller sample sizes, is that the choice of which individual to use as the x or the y is arbitrary, but that could impact the regression slope in FR-reg, as the sample variance can be different. As the sample size gets larger, it would be less of a problem because the slopes should be the same whether you regress $y_{m1} \sim y_{m2}$ or $y_{m2} \sim y_{m1}$ (from line 570) as N increases. How did the authors deal with that uncertainty?

A more major concern is that, unfortunately, the sample sizes are still relatively small and the variance attributable to the X often has large standard deviations or errors (Supplement Tables 4-6), which make it difficult to distinguish the estimates from 0. There are other datasets with many of these same measures in them that could also be leveraged (quickly, Generation Scotland and Framingham Heart Study come to mind, but I know there are others), which could provide both replication of your core findings and the chance to increase your overall sample size in a meta-analysis to help narrow your estimates' uncertainty. An argument of why this approach is useful, rather than those relying on genotype data, is that it could be applied to EHR data. Can you do that as a demonstration of the applicability of the method?

The ability to separate out shared environment from genetic variance is a nice feature. However, the estimates are of σ^2_{2a} , which encompasses the autosomal additive genetic variance and shared environmental variance, and σ^2_{2mPO} , which is the difference between parent-offspring and sib pair variance. While that difference is interesting on its own, it would be more interesting to estimate the familial, PO, and sib variances directly, and separate from the autosomal additive genetic variance. Is that not possible? Because σ^2_{2a} represents additive autosomal genetic variance and environmental variance, it's not directly comparable to h^2_{SNP} , and it's not clear how to view or interpret it as a composite of genetic + environment effects.

Your estimates of DC seem to be lower overall, including when using GCTA, than the recent Sidorenko 2019 estimates. Many of the traits overlap, and for some, XCHROM were not estimable (presumably because the variance estimates were negative). Can you discuss this difference more and possibly explain why? In general, a bit more context in terms of some of the existing estimates and literature that you cited would be helpful.

Other minor comments:

L244: Not exactly correct. It's the variance explained by all loci in LD with the imputed SNPs that have been used in the GRM.

L431:

The partitioning of the shared familial environment is a nice addition. Could the effect also be partitioned between same-sex sib pairs? For instance, would it be possible that the amount of phenotypic variance due to shared family environment is different when the sibs are both female than when the sibs are both male? This is mentioned as an assumption in the supplements, but it came to mind initially when reading this section.

L469:

Assumption of equal variances for both sexes due to autosomes is a strong assumption. Particularly given the X-by-autosome epistasis that is expected, how confident are the authors that this will be true? Can you present any evidence to support this assumption? If so, then what does it mean for X-by-autosome epistasis?

L499:

Assortative mating has been a recent focus of SNP-heritability studies. How does AM influence your estimates? As one point, the assumption of 0 covariance between father and son X chromosomes (L499) could be violated under AM conditions. Could the authors evaluate how AM would impact their estimates?

L692:

I see that you accounted for covariates here, but maybe a mention of it earlier when describing the general method would be useful because it immediately came to mind when seeing your initial approach description.

L716-717:

I'm guessing this is the Zaitlen 2013 PLoS Genetics reference?

L727:

The sample size is limiting the inference here for the X, as shown by the standard errors in table supplement 5, and the fact that many of these aren't distinguishable from 0. Can you run the analysis of unrelated individuals for the larger sample sizes available? If unable to perform a single GREML analysis, perhaps you could perform the analyses in batches and then meta-analyze the estimates to get tighter SEs on the X chromosome h²SNP estimates?

Supplements:

L109: The assumptions here are nicely laid out, but it would be helpful to have a discussion of what the impacts of assumption violation would possibly be. These are quite strong assumptions, and some of them may not hold.

L112: The first assumption that the Y chromosome contributes little to the heritability is probably true, but I don't understand the reasoning that it is because it doesn't recombine compared to other chromosomes. Whether recombination is extensive or not doesn't seem to be a requirement of your models. A better argument is that it is small and quite gene-poor.

L115: What do you mean by "pleiotropic phenotypes" in this context?

More importantly, what would be the consequence if this second assumption is invalid? For instance, if there are X-by-autosome interactions, then would you expect this assumption to hold?

(Remarks to the Author)

Lee and Han mathematically derived the genomic relationship between the first-degree relatives on X chromosome. They integrated this into a modified midparent-offspring regression framework to jointly estimate autosomal and X-chromosome heritability while accounting for familial environmental effects. They applied this method, namely X-CHROM, to 64 continuous traits in UK Biobank and found the average X chromosome heritability in male is 0.010 ± 0.102 and in female is 0.007 ± 0.078 , and the average dosage compensation ratio is of 1.304 ± 0.725 . Although enjoying reading this paper a lot, I am not convinced that X-CHROM is particularly useful nor adds any value to the quantitative genetics community.

Majors:

1), Low power due to large standard error and high demand on the sample size. Not very useful in reality.
 a. According to Figure 2a, the standard errors of X-CHROM estimates are large. X-CHROM will not provide any meaningful and robust estimates until the sample size reached 70,000 pairs for the family set, 30,000 pairs for the sibling set, and 40,000 pairs for the same sex set. Considering most cohorts do not have a very dense pedigree, likely not a single cohort available in the world consists of such many pairs of first-degree relatives to test the performance of X-CHROM. Have you considered doing a meta-analysis using data from family-based cohorts such as Generation Scotland (GS), Norwegian Mother, Father and Child Cohort Study (MoBA), The Tohoku Medical Megabank project (ToMMo), etc. The authors mentioned that X-CHROM can be applied on electronic medical records, but in reality, the access to EMR is difficult to obtain. In addition to that, information on genetic ethnics and population stratification is important for genetic study and is impossible to derive from EMR.
 b. The same standard error issue exists while applying X-CHROM in real data in UK Biobank. According to Supplementary Table 4 (X-CHROM results on 13,064 pairs of same sex relationship), the X chromosome heritability estimates of 32 out of 64 traits are negative in males and none of them is significantly different from 0 at $P < 0.05$. For females, 32/64 X chromosome heritability estimates are negatives, five of them are significantly different from 0 at $P < 0.05$ (two greater than 0 and three lower than 0), and none of them survived multiple testing. According to Supplementary Table 5 (GCTA results on 20,000 unrelated males/females), only 5 out of 64 male X chromosome heritability estimates are negative in GCTA unrelated samples, 32 out of 64 reached marginal significant level of 0.05, and 4 out of 64 passed Bonferroni correction. In females, 12 out of 64 estimates are negative, 11 traits reached marginal significant level, and one passed Bonferroni correction. These results provide that SNP-based approach with smaller sample size provides better estimates than X-CHROM.

2) Lack of novelty.

a. In quantitative genetics, the work on covariance between relatives on X chromosome can be dated back to Bohidar 19641. Following the work of Grossman and Fernando 19902, a full table of phenotypic covariance between relatives under sex-linkage for first-degree relatives is available in Table 24.1 of Genetics and Analysis of Quantitative Traits³. A full table of pedigree-based additive genetic relationships for first-degree relatives on X chromosome is available in 4. Several R packages can compute additive genetic relationship matrix based on pedigree for X chromosome, e.g. makeS5, nadv6, and kinship27. The work done by Lee and Han are not novel in quantitative genetics field. Fitting an X chromosome additive genetic relationship matrix (derived either using pedigree information or SNP information) in a REML framework (such as ASREML in animal breeding or GCTA in human genetics) is still the most common way to estimate heritability for X chromosome. And such an approach based on REML framework seems to yield smaller standard error in heritability estimates than X-CHROM.

3) Modelling

a. It is unclear to me how the X-CHROM can estimate 5 parameters using 4 relationships. According to your model assumption, in the same sex set, the correlation between a father and a son ($r_{\text{(Father-Son)}}$) is made up of $r_{\text{(Father-Son)}} = 1E_{\text{PO}} + 1E_{\text{Fam}} + 0E_{\text{Sib}} + 1/2 A_{\text{auto}} + 0A_{\text{x}}$, where E_{PO} , E_{Fam} , E_{Sib} , A_{auto} , and A_{x} represents the shared parent-offspring environmental effects, shared family environmental effects, shared sibling environmental effects, additive genetic effects contributed by autosomes, and additive genetic effects contributed by X chromosome, respectively. Similar, the correlation between the other three types of relationships can be written as
 $r_{\text{(Mother-Daughter)}} = 1E_{\text{PO}} + 1E_{\text{Fam}} + 0E_{\text{Sib}} + 1/2 A_{\text{auto}} + 1/2 A_{\text{x}}$
 $r_{\text{(Brother-Brother)}} = 0E_{\text{PO}} + 1E_{\text{Fam}} + 1E_{\text{Sib}} + 1/2 A_{\text{auto}} + 1/2 A_{\text{x}}$
 $r_{\text{(Sister-Sister)}} = 1E_{\text{PO}} + 1E_{\text{Fam}} + 1E_{\text{Sib}} + 1/2 A_{\text{auto}} + 3/4 A_{\text{x}}$

There are 5 parameters but only 4 equations, how does that work?

b. maternal effects. Maternal effects are effects shared with the mother and offspring. In the sex same set, maternal effects affect mother-daughter, brother-brother relationship, and sister-sister relationships but not father-son relationships.

$r_{\text{(Father-Son)}} = 1E_{\text{PO}} + 1E_{\text{Fam}} + 0E_{\text{Sib}} + 1/2 A_{\text{auto}} + 0A_{\text{x}} + 0A_{\text{maternal}}$
 $r_{\text{(Mother-Daughter)}} = 1E_{\text{PO}} + 1E_{\text{Fam}} + 0E_{\text{Sib}} + 1/2 A_{\text{auto}} + 1/2 A_{\text{x}} + 1A_{\text{maternal}}$
 $r_{\text{(Brother-Brother)}} = 0E_{\text{PO}} + 1E_{\text{Fam}} + 1E_{\text{Sib}} + 1/2 A_{\text{auto}} + 1/2 A_{\text{x}} + 1A_{\text{maternal}}$
 $r_{\text{(Sister-Sister)}} = 1E_{\text{PO}} + 1E_{\text{Fam}} + 1E_{\text{Sib}} + 1/2 A_{\text{auto}} + 3/4 A_{\text{x}} + 1A_{\text{maternal}}$

This may explain "Especially, each relationship exhibits varying shared environmental effects, such as the offspring sharing a more environmental effect with their mother than their father, leading to an overestimation of the X chromosome heritability by X-CHROM."

c. Why limited to first-degree relatives. There are a lot of second and third degree relatives in UK Biobank, e.g. aunt-nephew, grandmother-grandson, etc. There are methods to infer pedigree relationship based on the IBS sharing pattern, e.g.8.

Adding more distant relationships in X-CHROM can greatly increase the sample size and make it easier to disentangle different components.

d. ignore the sexual difference in autosomal heritability. A good reason not to make this assumption?

e. more parameters settings for the simulation. Currently the simulation study was performed under a single parameter setting. "The proportion of phenotypic variance due to each random variable was set to (0.5, 0.02, 0.05, 0.05, 0.1, 0.28) for (σ_{aut}^2 , σ_X^2 , $\sigma_{1\text{fam}}^2$, $\sigma_{1\text{po}}^2$, $\sigma_{1\text{sib}}^2$, σ_2^2) where σ_{aut}^2 is the variance by the autosomal variant, σ_X^2 is the variance by the X chromosome." The additive genetic effects is set to explain 50% of the phenotypic variance, which is quite high. On average across traits, autosomal heritability is about 15% in human complex traits. A higher additive genetic effects makes the model easier to split that from other small effects. What if the additive genetics is set to be much smaller? Will X-CHROM performs equally well?

Minors:

Interpretation of DC.

"Third, our study provides a new perspective on the dosage compensation mechanism of the X chromosome. Although the distribution of X chromosome heritability in males and females aligns more closely with the FDC for most phenotypes, there were exceptions. For example, our analysis of blood count traits ($n = 11$) shows a DC ratio value closer to the NDC (0.868). This suggests that the dosage compensation mechanism may not be equivalent across traits and further research is needed to comprehend the underlying mechanism."

I believe that the mechanism of DC does not change across traits. What changes is the effect of DC mechanism on traits depending on the location of trait loci. If QTLs of a trait are enriched in X-inactive region, we are more likely to detect DC ratio close to 2 using quantitative genetics method. If QTLs of a trait loci enriched in regions escape from X-inactivation, we are more likely to detect DC ratio close to 0.5. The authors need to interpret the results more carefully. A good example see 10.

Correlation between X-CHROM and GCTA estimates is nearly 0.

As the dots in figure 4a are too small and unclear, the reviewer estimated the correlation of X chromosome heritability between X-CHROM estimates in Supplementary Table 4 and GCTA estimates in Supplementary Table 5 across 64 traits and found that the correlation in males is 0.037 ± 0.124 with p value of 0.770 and in females is -0.019 ± 0.125 with p value of 0.88. There is no correlation between X-CHROM X heritability estimates and GCTA X chromosome heritability estimates, although the autosomal heritability from GCTA does correlated nicely with X-CHROM autosomal heritability in males (0.682 ± 0.057 with p value of 5.51×10^{-8}). This further confirms that the X-CHROM does not provide robust estimates of X chromosome heritability.

References:

- BOHIDAR, N. R. (1964).-Derivation and estimation of variance and covariance components associated with covariance between relatives under sex-linked transmission. *Biometrics* 20: 505-21
- Fernando RL, Grossman M. Genetic evaluation with autosomal and X-chromosomal inheritance. *Theor Appl Genet.* 1990 Jul;80(1):75-80.
- Lynch, M. and Walsh, B. (1998) Chapter 24: sex linkage and sexual dimorphism. *Genetics and analysis of quantitative traits.* Sinauer Associates, Inc., Sunderland.
- Druet, T., Legarra, A. Theoretical and empirical comparisons of expected and realized relationships for the X-chromosome. *Genet Sel Evol* 52, 50 (2020).
<https://search.r-project.org/CRAN/refmans/nadiv/html/makeS.html>
<https://cran.r-project.org/web/packages/nadiv/nadiv.pdf>
- Sinnwell JP, Therneau TM, Schaid DJ. The kinship2 R package for pedigree data. *Hum Hered.* 2014;78(2):91-3.
- Ramstetter MD, Shenoy SA, Dyer TD, Lehman DM, Curran JE, Duggirala R, Blangero J, Mezey JG, Williams AL. Inferring Identical-by-Descent Sharing of Sample Ancestors Promotes High-Resolution Relative Detection. *Am J Hum Genet.* 2018 Jul 5;103(1):30-44.
- Bernabeu, E., Canela-Xandri, O., Rawlik, K. et al. Sex differences in genetic architecture in the UK Biobank. *Nat Genet* 53, 1283–1289 (2021).
- Sidorenko, J. et al. The effect of X-linked dosage compensation on complex trait variation. *Nat Commun* 10, (2019).

Reviewer #3

(Remarks to the Author)

Lee et al. present a new method to estimate X chromosome heritability and dosage compensation, namely X-CHROM, and apply the method to study several outcomes. The article is well-written, and I found the idea to be both interesting and very clever. The paper also focuses on the X-chromosome which is unfortunately often left out in various genetic analyses due to it requiring a slightly different approach than the rest of the genome. Understanding dosage compensation is also an interesting and understudied question. They apply their method to 64 UKB outcomes. However, I was somewhat disappointed by their real data results, and they have insufficient power to detect interesting differences. I am also unsure of the underlying assumption that there are no effects from the Y chromosome, which I would expect could result in an apparent dosage effect for the X-chr in males. Regardless, it's interesting and I agree with their argument that this could be overcome if applied to larger datasets. Below I provide several comments aimed at improving the manuscript.

- The difference between X-CHROM and GCTA-GREML is striking, making me think it would be great to try this in larger datasets, e.g. finngen, deCODE, All-of-Us? Lack of power seems to be a key limitation of this study. More discussion (and

simulations?) on how large data are needed for this to work well would be good. What does 100K pairs really correspond to?

- The genetic architecture of metabolic outcomes is generally rather oligogenic, which can lead to unnecessarily high variance when estimating heritability using gcta/LDSC. As an alternative you can try SBayesS or LDpred2-auto.
- Consider GWAS summary statistics when estimating heritabilities, e.g. using LDSC/SumHer or even SBayesS/LDpred2-auto. This could allow you to increase the sample sizes, to see if that leads to better replication of X-CHROM.
- It would be great to use simulations to explore the impact of ignoring sex-specific differences in autosomal heritability and unaccounted shared environment.
- No case-control outcomes are considered, presumably due to even lower power and scale? I would nevertheless be interested in seeing some results for these, or at the minimum discussed.
- Please examine whether the Y chromosome impacts the method, or explain why it won't.
- It's impossible to read figure 2 because the font is small and quality poor (same goes for most figures).
- Regarding the methods, it's interesting that you estimate this using gradient descent in a two step procedure. I believe one could perhaps avoid the second step and obtain a better fit by jointly estimating the heritability and dosage compensation. This may however be difficult to do, and would be a different method.

Version 1:

Reviewer comments:

Reviewer #1

(Remarks to the Author)

The authors present a significantly updated manuscript that has addressed many of my previous questions. Most importantly, they updated their model to include different relationships, spanning not just sibs or parent-offspring, but more distant classes. In their new model, BIGFAM, they also attempt to disentangle the genetic from shared environment variance, using an expected function of variance decay with increasing relationship degree. The emphasis on X chromosome variance estimation is interesting, and the possible application to EHR data without genotype data could make it applied to other datasets, though the authors have notably not applied their approach to any datasets without existing genetic data, in contrast to some recently published work that has.

With the updated model, I have several questions that I believe are important to address:

1. Their new model has substantial similarity with HE regression. It would be good to have a formal comparison.
2. The assumed decay of the shared environment with increasing degree of relationship is a very strong assumption. Why do the authors assume any particular form? Different types of shared environments must have distinct forms of variance decay with increasing degree of relatives. For instance, some environments are likely shared just within nuclear families, while others are shared with more distant relatives. Those two situations would have different functions with regard to relationships. How would these impact the resulting estimates? What happens when the form of the decay is misspecified? Fig 4 suggests this is a significant problem. Fig 5b shows a clear trade off between V_g and V_s . that could represent real patterns where most traits have either genetic OR shared environment variance, but it seems striking that it's mostly one or the other. IS there evidence from the literature for those specific traits that it really is that? Polderman's meta analysis of twin correlations would be a good point of comparison here. Related, do the same traits measured in UKB and GS show the same patterns here? Are their estimates of V_s and V_g the same?
3. The new model would have the same individuals in multiple sets of the analyses, as individuals would have multiple types of relationships in these pedigrees. I might have missed it, but I didn't see how the authors deal with that non-independence within their datasets.
4. I'm struggling to understand the use case of the new model. The goal is to develop new methods of estimating genetic and shared environment variances without genotype data, and that's a great goal, with a couple of recent papers doing this in large EHR datasets. Yet the authors apply their approach using degree of relationship data from biobanks based on genome wide marker data, undermining the goal.

(Remarks on code availability)

Reviewer #2

(Remarks to the Author)

The authors have improved their manuscript greatly by having run simulations, extending the method to second- and third-degree relatives, and estimation autosomal heritability in addition to X-chromosome heritability. Despite the efforts, one major problem remained and it is elaborated as below.

I definitely see the value of having a genotype-free heritability estimation method. And I understand that most heritability

estimation software commonly used nowadays requires SNP data or GWAS summary-level data. However, the authors fail to convince me why to use BIGFAM over other software based on REML framework that is already on the market, such as AsREML and GCTA. REML is known for variance component analysis. It is on the market for at least 20 years. It can estimate heritability using pedigree information and/or genotype information (i.e. it can be genotype-free and it was designed to be genotype-free when it first used in animal breeding back in 1990s). By specifying the right model and fitting the right design matrices into REML-based software, REML can estimate the variance explained by each matrix. In your case, fitting an additive kinship matrix for autosomal heritability, an X-chromosome kinship matrix for X-chr heritability, and an environmental relationship matrix for the shared environment, all derived by pedigree information, jointly in ASREML and GCTA can solve the same problem as BIGFAM. I appreciate that the authors compared the BIGFAM results to LDSC heritability and GCTA X-chromosome model, however I do not think that is the best comparison to make because the underlying samples (i.e. without/with related samples used in GWAS for LDPred and LDSC), statistical mechanism or/and models are different. A fair comparison will be using the same number of individuals (i.e., the same pairs) and the same model (joint model of estimating autosomal heritability, x-chr heritability and common environment together) on REML software and BIGFAM, and compare estimate, standard error, computational time, and computation resources required between REML and BIGFAM. Note 1, GCTA is a REML-based software, it has the option to estimate a relationship matrix based on SNP-data and used this SNP-based matrix to estimate heritability, but one can fit any matrices into GCTA. This indicates that the authors' miss understanding of GCTA. Note 2, LDSC heritability is not SNP heritability. SNP heritability, also known as GCTA heritability or GREML heritability, usually refers to the heritability estimated by fitting a genomic relationship matrix in REML using only unrelated individuals. SNP/GCTA/GREML heritability in the presence of relatives yields similar estimate as Kinship/pedigree heritability. LDSC heritability is usually considered as the lower limit of narrow sense heritability and usually uses GWAS summary data from unrelated samples/or relatedness correct method in the presence of relatives. In general, LDSC heritability < SNP/GCTA/GREML heritability unrelated samples < SNP/GCTA/GREML heritability related samples < Kinship/pedigree heritability (e.g. BIGFAM.) Therefore, it is not surprising that BIGFAM heritability is greater than LDSC heritability. Note 3, In discussion the authors mentioned that the model " $P = A + S1 + S2 + S3$ " failed in REML. That is likely due to having three sparse matrices (i.e. S1, S2, and S3). We normally have one matrix for a specific type of common environment effects. In your case where the coefficient decays with degree, you can modify corresponding entry within the same matrix (e.g. 1 for 1st degree, 1/4 for 2nd degree, and 1/16 for 3rd degree assuming $W_s = 4$), rather than have three matrices. This should solve the problem. Note 4, in discussion the authors mentioned that "Expanding on this idea, the genetic relatedness matrix (GRM) can be replaced by an expected genetic relatedness matrix (eGRM) that incorporates the expected genetic correlation between individuals.". The 'eGRM' the authors referred here is what we called a kinship matrix or a pedigree matrix or an additive genetic matrix (noted as A matrix). People used kinship matrix to estimate heritability in animal breeding long before the first GWAS study.

(Remarks on code availability)

Reviewer #4

(Remarks to the Author)

The authors did a fantastic at addressing the previous reviewers' comments, having significantly improved the method. This paper examines a very important but relatively understudied aspect of heritability estimation, and provides a unique solution to the problem of estimating X chromosome h^2 . It addresses several important gaps in the existing h^2 estimation literature, and indeed offers a very refreshing perspective on the problem. In my opinion, this paper adds great value to the literature of heritability estimation, and would recommend it for publication, with some further improvements. Please see below my comments.

Major comments:

1. Can BIGFAM still partition and estimate the X chromosome h^2 if the shared environment component cannot be distinguished from the genetic component? In line 188, you provided the results that even when the slope test does not reject the null, BIGFAM can still predict the parameters, but can it still be used for estimating the h^2 of the X chromosome?
2. Figure 1: the caption needs a lot more details. It is actually unclear what each of the letter represents in panel b, and the resolution is not high enough for the labels to be legible.
3. In line 126-127, the author state that "we can assume $r_{s,l}$ is a r.v. identically distributed for all l with the same d ." Is there any evidence in the literature supporting this statement? Does this assumption have any biological support or foundation?
4. The first goal of BIGFAM is to infer whether the shared environmental component can be distinguished from the genetic component, so the null is $w_s = w_g$. I wonder why the null hypothesis is not whether or not there *exists* a shared environmental component (i.e. null is $w_s = 1$) In fact, the later is more intuitive for me. I couldn't get my head around why the decay in the shared environmental component by default is the same as the genetic component. (unless the pairs are twins perhaps)?
5. For the real trait analyses, it'd be quite interesting to analyze the shared environmental components of behavioral traits that have been established to have a strong genetic component, such as educational attainment and physical activity. Moreover, there has been some quite prominent studies showing that the interplay between genetics and environment can play a major role in the inheritance of trait as well (e.g. Kong et al. 2018 Science -- "genetic nurture" effect.) Did you find any significant variance components there?
6. In line 397, the authors presented two traits that have large shared environmental decaying values. It'd be good if the authors can follow up on this finding, and provide more intuition or evidence on why or why not this result is expected. This will strengthen the results section by a lot.
7. The section on heritability from BIGFAM and other methods, starting from line 404, is an important method validation section that can perhaps be brought more upfront in the results section. I had the question of how does the heritability

estimated from BIGFAM differs from GCTA and LDSC as I was reading through the text before this section.

8. How robust are the results to different or alternative relationship definition?

9. The last bit of the results can be strengthened a lot. In particular, it would be really helpful to include an application of BIGFAM on EHR data, for shared environmental component analyses or X chromosome analyses. Currently, being genotype-free is the most important highlight or benefit of this method, but the evidence that can directly support this statement is lacking. Any results that are interpretable and can demonstrate the utility of BIGFAM in a real trait analyses would be helpful.

10. Line 884: Are the results robust to these different inference criteria for the degree of relatedness?

11. Figure 2: What does the strong linear relationship between the slope and the intercept indicate? It's not clear to me why this is expected to be the case.

12. Figure 3: Notably, the strong linear relationship between the slope and the intercept in simulations is not visible in real trait analyses. What's the intuition behind this?

13. Figure 6: For method comparison, direct comparison of the estimates between/across methods is preferred, since it's more straightforward. Two sets of results can be highly correlated but have very low concordance (e.g. one is always twice as big as the other).

Minor comments:

1. The equation in line 720: I found this way of presenting the model a bit confusing, though I understand it might come from the previous reviewer's comment that the choice of X and Y may matter. Is the current model fitting every pair of related individuals twice (or as two data points)? More explanations or clarifications might be necessary.

2. The equation in line 760: the second term is just a constant, so not knowing it does not affect the test on the slope?

3. In line 808, maybe don't say "common variance components", as it's a bit unclear what "common" refers to.

4. Line 820-821: I again have a question about this assumption. Why is the variance by autosome canceled out but not the X chromosome?

5. Equations in line 818 and other similar equation forms: why is the V_x in the last line squared?

6. The w in the second term of equation in 867 should have a subscript of s

7. In figure 3: one of the boxes is white, but it may not have been intended to be this way.

(Remarks on code availability)

Version 2:

Reviewer comments:

Reviewer #4

(Remarks to the Author)

The authors have addressed all of my comments thoroughly and carefully. The revisions have significantly strengthened the quality of the paper, and I do not have any further comments.

(Remarks on code availability)

The code base is pretty well-organized, with both the test data and demo scripts available.

Reviewer #5

(Remarks to the Author)

1. I agree with Review 1 that the decay of the shared environment with increasing degree of relationship is a strong assumption. Can the authors validate this assumption by comparing the estimated shared environmental effects of the same trait from different studies and show the consistency?

2. SNP-based heritability is not directly comparable to family-based heritability estimates, as the former is a lower bound of the narrow-sense heritability due to ungenotyped markers and imperfect LD. The developed method and SNP-based method aim at estimating different quantities. This should be clarified.

3. The identification of shared environmental variance components depends on the power of slope test in step 1. Is the slope test in step 1 calibrated? Can the authors show the type-I error and power for step 1 under different assumptions of shared environmental effects?

(Remarks on code availability)

Version 3:

Reviewer comments:

Reviewer #5

(Remarks to the Author)

The authors have addressed all of my comments. I do not have any further comments.

(Remarks on code availability)

The software is well-documented.

RESPONSE TO REVIEWERS' COMMENTS

We deeply appreciate the enormously helpful comments from the three anonymous reviewers. We carefully revised the manuscript to address all the comments, which helped us to improve its clarity and expand its scope. Below are one-to-one responses to each of the reviewers' comments.

We note that there was a dramatic update in our model. Most importantly, by incorporating a broader range of relatives, we have expanded our capability to estimate not only the variance component attributable to the X chromosome but also those related to the autosome and shared environment. By doing so, we enable to direct comparison the heritability from our model with other SNP-based methods not only for the X chromosome but also the autosomes.

Moreover, addressing the larger standard errors of estimates on the X chromosome has been a primary focus in this revision. Through the development of more sophisticated models, we have successfully mitigated these errors, resulting in notable improvements. Unlike the previous model (XCHROM), which failed to detect significant X chromosome heritability for any phenotype (and in some cases even yielded negative estimates), our updated model (BIGFAM) has identified significant X chromosome heritability across 20 phenotypes. Furthermore, our results exhibit a substantial correlation with SNP-based heritability estimation ($\rho = 0.942$ with 95% CIs (0.925,0.952)), further validating the robustness of our approach.

We have diligently addressed any errors identified in our manuscript, in addition to providing one-to-one responses to each of the reviewers' comments. Once again, we express our sincere appreciation to the reviewers for their time and effort in reviewing our manuscript anonymously and voluntarily.

Reviewer #1

Major comments

(Major.1.1) Could the approach be expanded to account for more distant relatives? Grandparents or avuncular relationships when known? If the maternal or paternal grandparents are known, could one include other relatives to incorporate additional information and even additional sources of variance?

(Response.1.1) We sincerely appreciate the reviewer's insightful comments and suggestions. Expanding the approach to incorporate more distant relatives, such as grandparents or avuncular relationships, indeed enhances the richness of our analysis. We have taken this suggestion into careful consideration and have expanded our methodology to include not only marker-based kinship coefficients but also other factors such as sex, age, and age difference between relative pairs in the UK Biobank (UKB) dataset.

As a result, our analysis now encompasses a broader spectrum of familial relationships. In addition to the 27,365 first-degree relative pairs (involving 49,083 individuals), we have identified 9,447 second-degree (involving 17,240 individuals) and 44,514 third-degree relative pairs (involving 72,944 individuals).

Furthermore, leveraging genotype-based genetic correlations on the X chromosome, we have been able to infer specific familial relationships. Our analysis has revealed 22 distinct familial relationships, comprising 7 relationships among first-degree relatives, 7 among second-degree relatives, and 8 among third-degree relatives. The specific criteria for inferring familial relationships are described in the **Methods** section.

In addition to the UKB dataset, we also utilized another dataset, Generation Scotland: Scottish Family Health Study (GS:SFHS). GS:SFHS contains pedigree information from 4,830 families. From this dataset, we inferred 34 distinct familial relationships. Specifically, 7 relationships among first-degree relatives (18,258 pairs and 17,605 individuals), 17 relationships among second-degree relatives (15,114 pairs and 8,253 individuals), and 10 relationships among third-degree relatives (4,634 pairs and 2,686 individuals).

These various familial relationships, ranging from first-degree to third-degree relatives, not only enable us to estimate X chromosome heritability more accurately but also facilitate the prediction of additional sources of variance, including variance components by autosome and shared environment.

We extend our gratitude once again for the valuable contribution of this comment, which has significantly enriched the content and depth of our manuscript.

(Major.1.2) Could the approach be extended to a bivariate analysis to estimate partitioned genetic correlation of traits due to autosomes vs. X?

(Response.1.2) Thank you for your insightful comment and suggestion regarding the potential extension of our approach to a bivariate analysis. We appreciate your interest in exploring the partitioned genetic correlation of traits due to autosomes versus the X chromosome (X). We acknowledge the potential value of such an extension and have carefully considered your suggestion.

At present, BIGFAM does not support bivariate analysis for computing the genetic correlation between traits. This limitation arises because the phenotypes of close relatives, utilized in our approach, are inherently influenced by shared environmental effects. Consequently, applying the standard bivariate analysis model directly to close relatives may not yield accurate results.

However, we recognize the importance and potential value of extending our methodology to incorporate bivariate analysis in the future. We believe that additional insights into shared environmental factors or the development of more sophisticated models could enable the estimation of genetic and shared environmental correlations across multiple traits. We addressed the limitation by including **Discussion** in our manuscript to highlight this issue as follows:

Additionally, BIGFAM does not support bivariate analysis for computing the genetic correlation between traits. Unlike bivariate analysis with unrelated individuals, the phenotypes of close relatives are influenced by shared environmental effects. Consequently, the standard bivariate analysis model^{34,35} cannot be directly applied to close relatives. We anticipate that further insights into shared environmental factors or the development of more sophisticated models could enable the estimation of genetic and shared environmental correlations across multiple traits.

We will continue to explore avenues for enhancing the capabilities of our approach to better accommodate bivariate analyses in the context of familial relationships. We appreciate your valuable input, which contributes to the ongoing refinement and development of our methodology.

(Major.1.3) The choice of which individual to use as the x or the y is arbitrary, but that could impact the regression slope in FR-reg, as the sample variance can be different. As the sample size gets larger, it would be less of a problem because the slopes should be the same whether you regress $y_{m_1} \sim y_{m_2}$ or $y_{m_2} \sim y_{m_1}$ (from line 570) as N increases. How did the authors deal with that uncertainty?

(Response.1.3) Thank you for highlighting the potential impact of individual choice on the FR-reg, which is indeed a crucial consideration. We acknowledge the concern regarding the possibility of bias stemming from the arbitrary selection of X and Y variables, especially in datasets of smaller scale.

To mitigate this uncertainty in our BIGFAM analysis, we have implemented a comprehensive approach by regressing for all possible combinations within each pair. Specifically, we have refined the FR-reg methodology to encompass regressions on phenotype data for relative pairs at each degree of relatedness or familial relationship. This methodology is detailed in the **Methods** section as follows:

The general framework of FR-reg is identical to PO-reg, wherein it regresses the phenotypes of relative pairs. However, unlike PO-reg, FR-reg conducts regressions on phenotype data for relative pairs at each degree of relatedness (d) or familial relationship (l) as follows:

$$\begin{bmatrix} y_{m_1} \\ y_{m_2} \end{bmatrix} = \lambda_{rel} \cdot \begin{bmatrix} y_{m_2} \\ y_{m_1} \end{bmatrix}.$$

Here, rel denotes the relationship of relative pairs (m_1, m_2). This relationship can be defined either in terms of the degree-of-relatedness (d) or a specific familial relationship (l). y_{m_1} and y_{m_2} denotes the phenotype vectors of relative pairs (m_1, m_2). To address the uncertainty in selecting which individual's

phenotype to use as the predictor and which as the response in relative pairs (e.g., (mother, son) or (son, mother)), we perform regression analyses for all possible combinations within each pair.

By performing regression analyses for all possible combinations within each pair, we effectively address the uncertainty in choosing which individual's phenotype to use as the predictor and which as the response. This approach ensures that any potential bias due to the arbitrary selection of X and Y variables is minimized.

In our revised manuscript, we have explicitly discussed the potential uncertainty induced by the choice of X and Y variables in FR-reg and outlined our approach to mitigate this impact. By formulating FR-reg as the regression of all individuals' phenotypes on their relatives, we aim to provide a robust and comprehensive analysis framework.

We are confident that these measures effectively address the concerns raised in your comment and underscore the robustness of our findings against arbitrary variable selection. This information has been integrated into the revised manuscript to ensure clarity and transparency for our readers.

Once again, we sincerely appreciate your insightful feedback, which has significantly contributed to enhancing the quality and rigor of our research.

(Major.1.4) Unfortunately, the sample sizes are still relatively small and the variance attributable to the X often has large standard deviations or errors (Supplement Tables 4-6), which make it difficult to distinguish the estimates from 0. There are other datasets with many of these same measures in them that could also be leveraged (quickly, Generation Scotland and Framingham Heart Study come to mind, but I know there are others), which could provide both replication of your core findings and the chance to increase your overall sample size in a meta-analysis to help narrow your estimates' uncertainty correlation of traits due to autosomes vs. X?

(Response.1.4) Thank you for your valuable feedback regarding the limitations of our current sample size and the challenges associated with estimating variance attributable to the X chromosome. We acknowledge the importance of addressing these issues to enhance the robustness and precision of our findings.

To reduce the standard error and improve the reliability of our estimates, we have implemented the following strategies:

First, we utilized a broader range of relatives, including second- and third-degree relatives, in addition to first-degree relatives. This expansion increases the sample size and enhances the robustness of our analysis.

Second, to address the concern about sample size, we have taken your suggestion and plan to leverage additional datasets, specifically the Generation Scotland: Scottish Family Health Study (GS:SFHS) cohort. Incorporating data from these independent sources provides an opportunity for validation and replication of our core findings.

Third, we recognize the need to refine our modeling approach to mitigate the large standard errors observed in the X chromosome heritability. Therefore, we have developed more sophisticated models to address these issues. Our revised model, BIGFAM, estimates X chromosome heritability by regressing out common variance components from FR-reg coefficients with L2 penalty optimization. As a result, 20 phenotypes exhibited significant non-zero X chromosome heritability in our model. This is a marked improvement from our previous

model (XCHROM), which failed to identify significant X chromosome heritability across any phenotype. Additionally, our results demonstrated a substantial correlation with SNP-based heritability estimation (0.942 with 95% CIs (0.925,0.952)), further validating the efficacy of our refined approach.

By pursuing these avenues—leveraging additional familial relationships, incorporating more datasets, and refining our modeling approach—we believe we can effectively address the limitations highlighted in your comment.

We sincerely appreciate your insightful suggestions, which have guided us in improving the rigor and reliability of our research. Your contribution is invaluable, and we are committed to incorporating these enhancements into our work.

(Major.1.5) An argument of why this approach is useful, rather than those relying on genotype data, is that it could be applied to EHR data. Can you do that as a demonstration of the applicability of the method?

(Response.1.5) We appreciate the reviewer’s suggestion regarding the applicability of our approach to electronic health record (EHR) data. While accessing EHR data presents challenges due to privacy concerns, it is important to note that BIGFAM offers a distinct advantage as it does not rely on genotype data but instead utilizes familial relationships.

This characteristic renders BIGFAM a cost-effective and versatile method that can be applied in various scenarios. For instance, it can be particularly valuable for heritability analysis in understudied populations or datasets lacking genotype data but containing familial relationship information.

To demonstrate the applicability of our approach, we refer to the study “Disease Heritability Inferred from Familial Relationships Reported in Medical Records”¹. This study effectively utilized extensive EHR data to analyze familial relationships and compute observed heritability for multiple traits, showcasing the feasibility and relevance of employing familial relationship-based methodologies in the analysis of medical data.

We addressed this applicability on **Discussion** in our manuscript as follows:

... In summary, BIGFAM leverages relationship and phenotypic data to estimate variance components, offering a unique genotype-free methodology. This approach utilizes regression coefficients from phenotypic regressions on each relationship, making BIGFAM versatile for analyzing large datasets, particularly those lacking genotypic information. Beyond traditional datasets, BIGFAM’s applicability extends to electronic health record (EHR) data analysis. While accessing EHR data can be challenging, studies like the one by Polubriaginof et al.¹ demonstrate the feasibility of using familial relationships within EHRs to compute heritability. This highlights the potential of BIGFAM to uncover deeper insights into the biological and environmental architecture of traits using EHR data. Continued exploration and application of BIGFAM across diverse datasets, including EHRs, holds significant promise for advancing our understanding of the complex genetic architecture of traits and improving the accuracy of genetic analyses.

While accessing EHR data may present challenges, the flexibility and cost-effectiveness of the BIGFAM approach offer promising avenues for exploring heritability and related phenomena in diverse datasets. We will actively seek opportunities to collaborate with institutions that can provide access to EHR data to further validate and demonstrate the utility of our method.

We thank the reviewer for highlighting the potential application of our method and will consider further exploration of its utility in EHR data where feasible.

(Major.1.6) However, the estimates are of σ_a^2 , which encompasses the autosomal additive genetic variance and shared environmental variance, and σ_{mPO}^2 , which is the difference between parent-offspring and sib pair variance. While that difference is interesting on its own, it would be more interesting to estimate the familial, PO, and sib variances directly, and separate from the autosomal additive genetic variance. Is that not possible? Because σ_a^2 represents additive autosomal genetic variance and environmental variance, it's not directly comparable to h_{SNP}^2 , and it's not clear how to view or interpret it as a composite of genetic + environment effects.

(Response.1.6) We sincerely appreciate the reviewer's insightful comments and suggestions regarding our methodology for estimating variance components and its comparability with SNP heritability.

In response to your concerns, we have implemented refinements in our new model, *BIGFAM*, which allow for the direct estimation and disentanglement of variance components by genetic and shared environmental factors. One key enhancement involves the incorporation of the shared environmental decaying parameter (w_s), which allows us to effectively partition the variance components by genetic and shared environmental effects. This enables us to compare the genetic variance component (V_g) directly to SNP heritability estimates (h_{SNP}^2).

Our analyses reveal a significant correlation between heritability estimates computed from *BIGFAM* and SNP heritability estimates obtained from *LDSC* (squared Pearson correlation = 0.8749 with p-value of 7.0598×10^{-5}). This demonstrates that the predicted genetic variance component from our model is comparable with SNP heritability.

Interestingly, *BIGFAM* also demonstrated a high correlation with parent-offspring regression-based heritability estimates (*Traditional*), with a squared Pearson correlation of 0.8713 (p-value of 7.9198×10^{-5}). Notably, the squared correlation between *BIGFAM* and *LDSC* (0.8749 with p-value of 7.0598×10^{-5}) is significantly higher than the correlation between *Traditional* and *LDSC* (0.6811 with p-value of 3.2844×10^{-3}). This demonstrates that our model (*BIGFAM*) effectively conserves the variance component by genetic effect and partitions the variance component by shared environmental effect.

These findings are described in **Fig.6a** of our revised manuscript:

Moreover, to facilitate a direct comparison between variance components observed in parent-offspring (*PO*) and sibling (*SIB*) pairs, we have categorized first-degree relatives into distinct *PO* and *SIB* groups. Our results indicate a high correlation between variance components estimated for *PO* and *SIB* pairs. Specifically, the correlation between *PO* and *SIB* pairs was 0.899 (p-value of 4.241×10^{-27}) for the genetic variance component (V_g), 0.835 (p-value of 4.217×10^{-20}) for the shared environmental variance component (V_s), and 0.607 (p-value of 1.258×10^{-8}) for the shared environmental decaying parameter (w_s).

These findings are described in **Supplementary Fig. 8** and the **Results** section of our revised manuscript:

We are grateful for your feedback, which has been instrumental in refining our methodology and enhancing the interpretability of our results. If there are any further suggestions or inquiries, we are eager to address them.

(Major.1.7) Your estimates of DC seem to be lower overall, including when using GCTA, than the recent Sidorenko 2019 estimates. Many of the traits overlap, and for some, XCHROM were not estimable (presumably because the variance estimates were negative). Can you discuss this difference more and possibly explain why? In general, a bit more context in terms of some of the existing estimates and literature that you cited would be helpful.

(Response.1.7) Thank you for your insightful observation regarding the differences between our estimates and those reported by Sidorenko et al. (2019). Sidorenko et al. reported a dosage compensation ratio (DCR) slightly lower than 2 ($DCR = 1.85(se = 0.04)$), while our results exhibited a slightly smaller value ($1.534(se = 0.086)$).

To investigate this further, we directly compared the DCR values from our method (BIGFAM) with the GCTA software for 16 phenotypes where X chromosome heritability was significantly nonzero in both methods. The results revealed that the mean DCR from BIGFAM ($1.694(se = 0.111)$) and GCTA ($1.847(se = 0.092)$) were not significantly different ($p\text{-value} = 0.148$), but slightly smaller.

One possible explanation for this difference is the influence of the pseudo-autosomal region (PAR), positioned at the tip of the X chromosome. Recent studies have underscored the significance of genes in the PAR, revealing that many genes in the PAR are fully expressed in both activated and inactivated X chromosomes in females. Our approach estimates the DCR on the entire X chromosome, including both PAR and non-PAR regions (NPX). However, GCTA-GREML, as utilized in the study by Sidorenko et al., recommends using the X chromosome variants only in NPX. Therefore, the DCR including the PAR may be lower than that of NPX. We have addressed this explanation in our manuscript **Results** section as follows:

One possible explanation for this difference is the influence of the pseudo-autosomal region (PAR) on the X chromosome. Due to evolutionary and transcriptional differences between the PAR and non-PAR (NPX) regions², GCTA recommends using variants only in the NPX. However, genes in the PAR are fully activated in both the active and inactive X chromosomes. In contrast, most genes in the NPX exhibit varied activation profiles in the inactive X chromosome^{2,3}. Therefore, we can expect that the DCR for the PAR would be lower than that for the NPX. Unlike GCTA, which only considers DCR in the NPX, BIGFAM considers the entire X chromosome (both PAR and NPX), potentially resulting in the observed lower DCR from BIGFAM compared to GCTA.

We are grateful for your comments, which have prompted us to delve deeper into our results and provide a more detailed explanation for the observed differences. Your observation has driven us to enrich our discussion, thereby strengthening our manuscript. If there are any further suggestions or inquiries, we are eager to address them.

Minor comments

(Minor.1.1) L244: Not exactly correct. It's the variance explained by all loci in LD with the imputed SNPs that have been used in the GRM.

(Response) Thank you for clarifying this point. We will revise the text to indicate that the variance explained by all loci in linkage disequilibrium with the imputed SNPs used in the genetic relationship matrix (GRM).

(Minor.1.2) L431: The partitioning of the shared familial environment is a nice addition. Could the effect also be partitioned between same-sex sib pairs? For instance, would it be possible that the amount of phenotypic variance due to shared family environment is different when the sibs are both female than when the sibs are both male? This is mentioned as an assumption in the supplements, but it came to mind initially when reading this section.

(Response) Thank you for your thoughtful comment regarding the partitioning of the shared familial environment. It is indeed possible to estimate the variance component by shared environment for same-sex sibling pairs. This can be achieved by utilizing the familial relationship regression (FR-reg) coefficient specifically for same-sex sibling pairs within the first-degree relatives.

However, using only one specific familial relationship, such as same-sex sibling pairs, results in a smaller sample size. This smaller sample size can potentially lead to less stable results in BIGFAM analysis.

Instead, we partitioned the variance component by genetic and shared environmental effects using parent-offspring relationships and sibling relationships. The results revealed that the predicted model parameters (V_g , V_s , and w_s) computed from PO and SIB pairs were significantly correlated. We have detailed this approach and its outcomes in the **Results** section of our revised manuscript.

Thank you once again for your insightful suggestion, which has helped us enhance the clarity and robustness of our analysis.

(Minor.1.3) L469: Assumption of equal variances for both sexes due to autosomes is a strong assumption. Particularly given the X-by-autosome epistasis that is expected, how confident are the authors that this will be true? Can you present any evidence to support this assumption? If so, then what does it mean for X-by-autosome epistasis?

(Response) Thank you for your comment. In our previous model (XCHROM), we indeed assumed equal variances for both sexes, which was a strong assumption. Previous research⁴ has shown that in many traits, the genetic correlation between males and females is significantly different from 1, indicating potential differences in variances between sexes.

In our new model (BIGFAM), we do not make the assumption of equal variances for both sexes. Therefore, the estimation of heritability for each sex is feasible when utilizing male and female relative pairs.

Regarding the question about X-by-autosome epistasis, this refers to the interaction between genetic variants on the X chromosome and autosomes, which can influence phenotypic traits. While our new model (BIGFAM) does not explicitly address X-by-autosome epistasis, it remains an area of interest and may influence the interpretation of our results. Future research could explore the implications of X-by-autosome epistasis in the context of our findings.

(Minor.1.4) L499: Assortative mating has been a recent focus of SNP-heritability studies. How does AM influence your estimates? As one point, the assumption of 0 covariance between father and son X chromosomes (L499) could be violated under AM conditions. Could the authors evaluate how AM would impact their estimates?

(Response) Thank you for highlighting the potential influence of assortative mating (AM) on our estimates. Assortative mating can indeed impact heritability estimates, particularly in parent-offspring regression. In our study, we did not explicitly evaluate the impact of assortative mating on our estimates. However, our **Supplementary Text** discusses how assortative mating can affect heritability estimation in traditional additive genetic models.

In brief, assortative mating can inflate heritability estimates due to the genetic resemblance between close relatives. Specifically, under assortative mating, the covariance between parent and offspring phenotypes can increase, leading to overestimation of genetic variance components.

However, our model, BIGFAM, can still effectively partition the variance components by genetic and shared environmental effects, as the variance component by genetic effect in the FR-reg coefficient still decays as a factor of 2.

While our study did not directly account for assortative mating, this remains an important consideration in interpreting heritability estimates, particularly in parent-offspring regression analyses. Thank you for bringing this to our attention, and we will consider discussing this further in our manuscript.

(Minor.1.5) L692: I see that you accounted for covariates here, but maybe a mention of it earlier when describing the general method would be useful because it immediately came to mind when seeing your initial approach description.

(Response) Thank you for your observation. You're correct that we accounted for covariates in our analysis, although it wasn't explicitly mentioned earlier when describing the general method. We appreciate your suggestion to include this information earlier in the description, as it provides important context for understanding our approach. We will make sure to incorporate a mention of covariate adjustment in the relevant section to enhance clarity for readers.

(Minor.1.6) L716-717: I'm guessing this is the Zaitlen 2013 PLoS Genetics reference?

(Response) Thank you for your comment. We added the reference (Zaitlen et al., 2013, PLoS Genetics) mentioned about big-K small-K in our revised manuscript.

(Minor.1.7) L727: The sample size is limiting the inference here for the X, as shown by the standard errors in table supplement 5, and the fact that many of these aren't distinguishable from 0. Can you run the analysis of unrelated individuals for the larger sample sizes available? If unable to perform a single GREML analysis, perhaps you could perform the analyses in batches and then meta-analyze the estimates to get tighter SEs on the X chromosome h²SNP estimates?

(Response) Thank you for your insightful comment regarding the limitation of sample size and its impact on the inference for the X chromosome. In our revised manuscript, we have addressed this concern by significantly increasing our sample size.

Different from our previous manuscript, which used 20,000 individuals, the revised analysis includes 314,699 unrelated individuals (169,019 females and 145,680 males). To manage this large sample size, we partitioned them into 30 blocks, computed the genetic relationship matrix (GRM) for each block, and performed GREML analysis. We then merged the estimates using inverse-variance weighting (IVW) to obtain tighter standard errors on the X chromosome SNP heritability estimates.

We believe this approach has improved the robustness of our estimates, reducing the standard errors and making our results more reliable.

(Minor.1.8) Supplements L109: The assumptions here are nicely laid out, but it would be helpful to have a discussion of what the impacts of assumption violation would possibly be. These are quite strong assumptions, and some of them may not hold.

(Response) Thank you for your feedback. We acknowledge the importance of discussing the potential impacts of assumption violations in our methodology. While our developed model has shown significant improvements compared to the previous method (XCHROM), we recognize that it still has limitations regarding certain assumptions.

To address this concern, we have simulated scenarios of assumption violations in both our revised manuscript and **Supplementary Text**. By doing so, we aim to provide a comprehensive understanding of how these violations may affect our results and interpretations. We appreciate your valuable input and will ensure to include a thorough discussion of assumption impacts in our manuscript.

(Minor.1.9) Supplements L112: The first assumption that the Y chromosome contributes little to the heritability is probably true, but I don't understand the reasoning that it is because it doesn't recombine compared to other chromosomes. Whether recombination is extensive or not doesn't seem to be a requirement of your models. A better argument is that it is small and quite gene-poor.

(Response) Thank you for your insightful comment. You're correct that the reasoning behind the assumption that the Y chromosome contributes little to heritability due to its lack of recombination may not be entirely accurate.

A more appropriate argument for this assumption is indeed that the Y chromosome is relatively small and gene-poor compared to other chromosomes. Therefore, its contribution to overall

heritability is expected to be minimal. We explained about the impact of Y chromosome in the **Supplementary Text**.

We appreciate your clarification on this matter.

(Minor.1.10) Supplements L115: What do you mean by “pleiotropic phenotypes” in this context? More importantly, what would be the consequence if this second assumption is invalid? For instance, if there are X-by-autosome interactions, then would you expect this assumption to hold?

(Response) Thank you for your insightful feedback. We appreciate the opportunity to clarify and improve our manuscript based on your comments.

We acknowledge the confusion regarding the term “pleiotropic phenotypes” and have corrected it in the revised manuscript to use more appropriate terms such as “complex traits” or “continuous traits.” This adjustment should provide clearer and more accurate descriptions of the phenotypes we are analyzing.

Regarding the second assumption in our previous manuscript, which stated “no sex-specific autosome and environmental effect,” we have made significant advancements in our revised methods (BIGFAM). In BIGFAM, this assumption is no longer necessary, however, it is important to note that our revised model still assumes that the autosome and X chromosome are independent genetic effects, meaning there is no X-by-autosome interaction. If X-by-autosome interactions were to exist, it could indeed bias our estimates of X chromosome heritability. This is an important consideration, and we recognize the potential impact this could have on our findings. Although our current model does not account for X-by-autosome interactions, we are committed to exploring ways to incorporate this effect in future work to further refine and enhance the accuracy of our estimates.

We greatly appreciate your thoughtful suggestion, which has highlighted an area for future development. Your feedback is invaluable to us as we strive to improve the robustness and reliability of our research. We will ensure to consider this in our ongoing and future studies.

Reviewer #2

Major comments

(Major.2.1) Low power due to large standard error and high demand on the sample size. Not very useful in reality. Considering most cohorts do not have a very dense pedigree, likely not a single cohort available in the world consists of such many pairs of first-degree relatives to test the performance of X-CHROM. Have you considered doing a meta-analysis using data from family-based cohorts such as Generation Scotland (GS), Norwegian Mother, Father and Child Cohort Study (MoBA), The Tohoku Medical Megabank project (ToMMo), etc.

(Response.2.1) Thank you for your insightful comment regarding the challenges of low power due to large standard errors and the high demand on sample size. We recognize the limitations this imposes on the practical applicability of our approach. To address these issues, we have implemented several strategies to improve the precision and reliability of our estimates.

First, to address the concern about sample size, we have taken your suggestion and plan to incorporate data from additional cohorts, specifically the Generation Scotland: Scottish Family Health Study (GS:SFHS) cohort. Incorporating data from these independent sources provides an opportunity for validation and replication of our core findings.

Second, we utilized a broader range of relatives. In addition to first-degree relatives, we have included second- and third-degree relatives in our analysis. This expansion increases the sample size and enhances the robustness of our estimates.

Third, We have refined our methodology by developing more sophisticated models to address the issue of large standard errors. Our revised model, BIGFAM, estimates X chromosome heritability by regressing out common variance components from FR-reg coefficients using L2 penalty optimization.

As a result, 20 phenotypes exhibited significant non-zero X chromosome heritability in our model. This is a marked improvement from our previous model (XCHROM), which failed to identify significant X chromosome heritability across any phenotype. Additionally, our results demonstrated a substantial correlation with SNP-based heritability estimation (0.942 with 95% CIs (0.925,0.952)). This finding is described in **Fig.7a** and the **Results** section of our revised manuscript:

By implementing these strategies, we believe we can effectively address the limitations highlighted in your comment. We are confident that these improvements will enhance the practical applicability of our methodology and provide more reliable estimates.

Thank you once again for your valuable feedback. Your suggestions have significantly contributed to the refinement of our approach and the enhancement of our manuscript.

(Major.2.2) The authors mentioned that X-CHROM can be applied on electronic medical records, but in reality, the access to EMR is difficult to obtain. In addition to that, information on genetic ethnics and population stratification is important for genetic study and is impossible to derive from EMR.

(Response.2.2) Thank you for your thoughtful comment regarding the application of X-CHROM to electronic medical records (EMR) and the challenges associated with accessing EMR data and obtaining information on genetic ethnicity and population stratification. We recognize these important considerations and appreciate the opportunity to address them.

While accessing EMR data presents challenges due to privacy concerns, it is important to note that our revised model (BIGFAM) offers a distinct advantage as it does not rely on genotype data but instead utilizes familial relationships. This characteristic renders BIGFAM a cost-effective and versatile method that can be applied in various scenarios. For instance, it can be particularly valuable for heritability analysis in understudied populations or datasets lacking genotype data but containing familial relationship information.

To demonstrate the applicability of our approach, we refer to the study “Disease Heritability Inferred from Familial Relationships Reported in Medical Records” [1]. This study effectively utilized extensive EMR data to analyze familial relationships and compute observed heritability for multiple traits, showcasing the feasibility and relevance of employing familial relationship-based methodologies in the analysis of medical data. We addressed this applicability in the **Discussion** section of our manuscript as follows:

While accessing EHR data can be challenging, studies like the one by Polubriaginof et al. demonstrate the feasibility of using familial relationships within EHRs to compute heritability. This highlights the potential of BIGFAM to uncover deeper insights into the biological and environmental architecture of traits using EHR data. Continued exploration and application of BIGFAM across diverse datasets, including EHRs, holds significant promise for advancing our understanding of the complex genetic architecture of traits and improving the accuracy of genetic analyses.

While we acknowledge these advantages, we also recognize that our model has limitations regarding information on genetic ethnicity and population stratification. It is essential to highlight that BIGFAM does not directly account for genetic principal components for ancestry correction, which could potentially lead to biased heritability estimates due to population stratification. We have addressed this issue in the **Discussion** section of our manuscript as follows:

BIGFAM, while innovative, is subject to several limitations and questions that remain open for further research ... heritability estimates may be affected by population stratification. BIGFAM does not directly account for genetic principal components for ancestry correction, which can lead to biased heritability estimates

These limitations point to areas for future research and improvement in the methodology. We are committed to exploring these challenges and seeking opportunities to collaborate with

institutions that can provide access to EHR data to further validate and demonstrate the utility of our method.

Thank you once again for your valuable feedback, which has significantly contributed to enhancing the clarity and depth of our manuscript.

(Major.2.3) According to Supplementary Table 4 (X-CHROM results on 13,064 pairs of same sex relationship), the X chromosome heritability estimates of 32 out of 64 traits are negative in males and none of them is significantly different from 0 at $P < 0.05$. For females, 32/64 X chromosome heritability estimates are negatives, five of them are significantly different from 0 at $P < 0.05$ (two greater than 0 and three lower than 0), and none of them survived multiple testing. According to Supplementary Table 5 (GCTA results on 20,000 unrelated males/females), only 5 out of 64 male X chromosome heritability estimates are negative in GCTA unrelated samples, 32 out of 64 reached marginal significant level of 0.05, and 4 out of 64 passed Bonferroni correction. In females, 12 out of 64 estimates are negative, 11 traits reached marginal significant level, and one passed Bonferroni correction. These results provide that SNP-based approach with smaller sample size provides better estimates than X-CHROM.

(Response.2.3) Thank you for your detailed comment regarding the comparison between X-CHROM and SNP-based approaches. We recognize the importance of addressing the issues related to the precision of heritability estimates and the advantages of SNP-based methods with smaller sample sizes.

To improve the precision of our estimates and address the concerns about large standard errors, we developed a more sophisticated model, BIGFAM, which demonstrated significantly smaller standard errors. Our revised analysis with BIGFAM showed that 20 phenotypes (from the UK Biobank (UKB) and Generation Scotland: Scottish Family Health Study (GS:SFHS) datasets) exhibited significantly nonzero X chromosome heritability. Additionally, 32 phenotypes showed significantly nonzero X chromosome heritability in both males and females.

For validation and a more robust analysis, we computed SNP heritability using 314,699 unrelated individuals (169,019 females and 145,680 males), compared to the 20,000 males and 20,000 females used in our previous manuscript. To manage this large sample size, we partitioned the individuals into 30 blocks, computed the genetic relationship matrix (GRM) for each block, and performed GREML analysis. We then merged the estimates using inverse-variance weighting (IVW) to obtain tighter standard errors on the X chromosome SNP heritability estimates.

For the 13 phenotypes where X chromosome heritability was significantly nonzero in both methods, BIGFAM and GCTA showed a significant correlation (0.942 with 95% CIs (0.925,0.952)). For the 16 phenotypes where X chromosome heritability was significantly nonzero in both sexes and methods, BIGFAM and GCTA showed a significant correlation in males (0.949 with 95% CIs (0.937,0.970)) and in females (0.650 with 95% CIs (0.539,0.718)). This results is described in **Supplementary Fig. 12** as follows:

In the UKB dataset, GCTA showed significant nonzero X chromosome heritability for 101 out of 109 phenotypes, whereas BIGFAM showed significant nonzero X chromosome heritability for 14 out of 109 phenotypes. It's important to note that GCTA utilized 314,699 unrelated individuals, while BIGFAM utilized 48,912 individuals (27,612 pairs). Even though BIGFAM showed a smaller number of significant X chromosome heritability estimates, it used a smaller number of individuals.

The advantage of BIGFAM lies in its ability to analyze large datasets without relying on genotype data, offering a cost-effective and versatile solution for genetic analyses across various populations. This makes it particularly valuable for heritability analysis in understudied populations or datasets lacking genotype data but containing familial relationship information.

Thank you once again for your valuable feedback, which has significantly contributed to the refinement and enhancement of our methodology and analysis.

(Major.2.4) In quantitative genetics, the work on covariance between relatives on X chromosome can be dated back to Bohidar 1964. Following the work of Grossman and Fernando 1990, a full table of phenotypic covariance between relatives under sex-linkage for first-degree relatives is available in Table 24.1 of Genetics and Analysis of Quantitative Traits. A full table of pedigree-based additive genetic relationships for first-degree relatives on X chromosome is available in 4. Serval R packages can compute additive genetic relationship matrix based on pedigree for X chromosome, e.g. makeS5, nadiv6, and kinship27. The work done by Lee and Han are not novel in quantitative genetics field. Fitting an X chromosome additive genetic relationship matrix (derived either using pedigree information or SNP information) in a REML framework (such as ASREML in animal breeding or GCTA in human genetics) is still the most common way to estimate heritability for X chromosome. And such an approach based on REML framework seems to yield smaller standard error in heritability estimates than X-CHROM.

(Response.2.4) Thank you for your insightful comment and for highlighting the foundational work in the field of quantitative genetics, particularly regarding the X chromosome. We recognize and respect the extensive history and established methodologies, such as pedigree-based additive genetic relationships and SNP-based heritability approaches like GCTA, which are well-established in the field.

As you noted, these traditional methods require SNP information, which may not always be available in various datasets. In contrast, our method offers a unique advantage as it does not rely on SNP information, making it applicable even in datasets where genotype data is lacking. This versatility allows for heritability analysis in understudied populations or datasets with limited genetic data.

Through methodological advancements, we have reduced the standard error compared to previous models and achieved comparable results with GCTA. While alternative approaches, such as constructing a genetic relatedness matrix based on expected correlations between relatives, may seem promising, our simulation experiments revealed challenges related to convergence, particularly due to the high sparsity of the matrix. Specifically, we encountered issues with the non-invertibility of the information matrix used in the REML algorithm. This non-invertibility was attributed to the matrix's sparsity, filled with zeros for unrelated individuals and non-zero values for related individuals. We have addressed this issue in detail in the **Discussion** section of our manuscript as follows:

Another possible approach for estimating heritability from relatives without genotype is utilizing a relationship matrix filled with expected correlation. This concept builds upon a previous study, which introduced a familial relationship matrix consisting of 0 (indicating unrelated individuals) and 1 (indicating related individuals). Expanding on this idea, the genetic relatedness matrix (GRM) can be replaced by an expected genetic relatedness matrix (eGRM) that incorporates the expected genetic correlation between individuals. For example, in the context of first- to third-degree relatives, the covariance matrix of phenotypes (P) can be formulated as a weighted sum of correlation matrices for autosomes (A) and shared environments for each degree of relatedness ($P = A + S_1 + S_2 + S_3$). However, in our simulation experiments, we encountered challenges with convergence of the estimates. This issue was due to the non-invertible information matrix used in the REML algorithm. The sparsity of the matrix, filled with zeros for unrelated individuals and non-zero for related individuals across all correlation matrices, led to non-invertibility caused by negative eigenvalues in the correlation matrices.

While traditional REML-based approaches may yield smaller standard errors in heritability estimates, our method, BIGFAM, provides a practical and robust alternative for scenarios where genotype data is not available. We continue to explore ways to improve the convergence and accuracy of our model, ensuring it remains a valuable tool for genetic studies across diverse datasets.

Thank you once again for your valuable feedback. Your comments have helped us to refine our discussion and underscore the unique contributions and limitations of our approach.

(Major.2.5) It is unclear to me how the X-CHROM can estimate 5 parameters using 4 relationships. According to your model assumption, in the same sex set, the correlation between a father and a son ($r_{(Father-Son)}$) is made up of $r_{(Father-Son)} = 1E_{PO} + 1E_{Fam} + 0E_{Sib} + 1/2 A_{auto} + 0A_x$, where E_{PO} , E_{Fam} , E_{Sib} , A_{auto} , and A_x represents the shared parent-offspring environmental effects, shared family environmental effects, shared sibling environmental effects, additive genetic effects contributed by autosomes, and additive genetic effects contributed by X chromosome, respectively. There are 5 parameters but only 4 equations, how does that work?

(Response.2.5) Thank you for your insightful comment regarding the estimation of parameters in our X-CHROM model. We appreciate the opportunity to clarify this aspect of our methodology.

In our earlier approach, we partitioned the phenotypic correlation into four distinct components (σ_a^2 , σ_{mPO}^2 , σ_{xMale}^2 , and $\sigma_{xFemale}^2$). Based on these parameters, the regression coefficient between father and son is expressed as:

$$\lambda_{father-son} = 0.5 \cdot \sigma_a^2 + \sigma_{mPO}^2 + 0 \cdot \sigma_{xMale}^2 + 0 \cdot \sigma_{xFemale}^2.$$

Because X-CHROM utilized only same-sex pairs, and σ_a^2 is composed of three different components ($\sigma_a^2 = \sigma_{aut}^2 + 2 \cdot \sigma_{fam}^2 + 2 \cdot \sigma_{sib}^2$) for intuitive interpretation, this initially seemed to make X-CHROM capable of estimating 5 parameters using 4 equations.

However, X-CHROM does not separately estimate the components of σ_a^2 . Instead, it merges σ_{aut}^2 , σ_{fam}^2 , and σ_{sib}^2 into a single parameter σ_a^2 . Consequently, X-CHROM effectively estimates 4 parameters (σ_a^2 , σ_{mPO}^2 , σ_{xMale}^2 , and $\sigma_{xFemale}^2$) using four equations. Due to the number of equations being identical to the number of parameters, the estimates from X-CHROM are unstable and tend to have large standard errors.

To address this problem, we developed a more sophisticated model, BIGFAM. In BIGFAM, we utilized not only same-sex pairs but also different-sex pairs, and included second- and third-degree relatives in addition to first-degree relatives. This broader inclusion increases the robustness of our model. Additionally, by regressing out the common variance components within each degree-of-relatedness bin, we can isolate and estimate the X chromosome heritability across various familial relationships. This revised method is detailed in the **Methods** section of our manuscript as follows:

Next, we formulate the the expectation of residual of FR-reg coefficient for l relation ($\hat{\lambda}_l^{res}$) as follows:

$$\begin{aligned} E[\hat{\lambda}_l^{res}] &= E[\hat{\lambda}_l] - E[\lambda_d] \\ &= (E[r_{a,l}] \cdot V_a + E[r_{X,l}] \cdot V_X + E[r_{s,l}] \cdot V_s) \\ &\quad - (E[r_{a,d}] \cdot V_a + E[r_{X,d}] \cdot V_X + E[r_{s,d}] \cdot V_s) \\ &= t_l \cdot V_X^2 + \epsilon_s \end{aligned}$$

where t_l represents $E[r_{X,l}] - E[r_{X,d}]$ and ϵ_s is $(E[r_{s,l}] - E[r_{s,d}]) \cdot V_s$.

It is important to note that the variance component by autosome (V_a) is canceled out because the genetic correlation on the autosome is a constant within each bin ($E[r_{a,l}] = E[r_{a,d}]$). Additionally, due to our assumption that $r_{s,l}$ follows a normal distribution with a mean of $r_{s,d}$, the expected value of ϵ_s is zero. Hence, we can regard the variance component by relationship-specific shared environmental effects as a mean zero error term. Lastly, since $r_{X,l}$ is well-documented^{5, 6}, t_l can be specified based on the relationship used in the analysis. Consequently, we can infer the variance component by X chromosome (V_X) by using only FR-reg coefficient.

We believe that these advancements significantly improve the stability and accuracy of our estimates. Thank you again for your valuable feedback, which has helped us to refine our approach and enhance the robustness of our methodology.

(Major.2.6) Maternal effects are effects shared with the mother and offspring. In the sex same set, maternal effects affect mother-daughter, brother-brother relationship, and sister-sister relationships but not father-son relationships. This may explain “Especially, each relationship exhibits varying shared environmental effects, such as the offspring sharing a more environmental effect with their mother than their father, leading to an overestimation of the X chromosome heritability by X-CHROM.”

(Response.2.6) Thank you for your insightful comment regarding the potential influence of maternal effects on the estimation of X chromosome heritability. We recognize the importance of considering relationship-specific shared environmental effects, particularly maternal effects, in our analysis.

To address this concern, we conducted simulations to explore various relationship-specific shared environmental scenarios, including first-degree specific shared environments, maternal effects, and sibling-specific shared environmental scenarios. Through these simulations, we aimed to assess how our method, BIGFAM, performs under different shared environmental conditions.

The results of our simulations, as depicted in **Fig.4b**, indicate that while there may be a slight overestimation of X chromosome heritability in certain scenarios, BIGFAM effectively approximates the heritability and X chromosome heritability across all simulated shared environmental scenarios. These findings suggest that BIGFAM is robust and performs well even when maternal effects and other relationship-specific shared environmental effects are present.

Furthermore, we acknowledge and address the potential factors contributing to the slight overestimation observed in some scenarios. Specifically, we discuss the possibility that a correlation between genetic correlation on the X chromosome and shared environmental correlation can induce overestimation in the estimates. We have clarified this explanation in the **Results** section of our revised manuscript as follows:

In S_3 and S_4 , where we modeled maternal and sibling-specific shared environmental effects, the estimated V_X were slightly higher compared to S_1 and S_2 . This is possibly because relationships having higher V_s also have higher genetic correlation on the X chromosome ($r_{X,l}$). For example, the $r_{X,l}$ of mother-offspring relationships ($\frac{1}{2}$ and $\frac{1}{\sqrt{2}}$) is higher than father-offspring relationships ($\frac{1}{\sqrt{2}}$ and 0). Our model assumed independence between these effects, and therefore, when $r_{X,l}$ is correlated with the shared environmental effect, the estimated V_X can be slightly higher than the true value.

We believe that these simulations and explanations help clarify the robustness of BIGFAM in accounting for maternal effects and other relationship-specific shared environmental factors. We appreciate your valuable feedback, which has guided us in refining our methodology and improving the accuracy of our estimates.

Thank you once again for your constructive comments, which have significantly contributed to enhancing the clarity and robustness of our manuscript.

(Major.2.7) Why limited to first-degree relatives. There are a lot of second and third degree relatives in UK Biobank, e.g. aunt-nephew, grandmother-grandson, etc. There are methods to infer pedigree relationship based on the IBS sharing pattern, e.g.8. Adding more distant relationships in X-CHROM can greatly increase the sample size and make it easier to disentangle different components.

(Response.2.7) Thank you for your insightful comment regarding the inclusion of more distant relatives in our analysis. We agree that incorporating second- and third-degree relatives can greatly enhance the robustness and accuracy of our estimates by increasing the sample size and allowing for better disentanglement of different variance components.

To address this, we have expanded our methodology to utilize not only marker-based kinship coefficients but also other factors such as sex, age, age difference between relative pairs, and genotype-based genetic correlations on the X chromosome in the UK Biobank (UKB) dataset. This comprehensive approach allows us to include a broader range of familial relationships in our analysis.

Our revised analysis now includes 22 distinct familial relationships. Specifically, 7 relationships among first-degree relatives (10,404 pairs involving 19,348 individuals), 7 familial relationships among second-degree relatives (2,454 pairs involving 4,743 individuals), and 8 familial relationships among third-degree relatives (14,754 pairs involving 27,069 individuals).

In addition to the UKB dataset, we also utilized another dataset, Generation Scotland: Scottish Family Health Study (GS:SFHS), which contains pedigree information from 4,830 families. From this dataset, we inferred 34 distinct familial relationships. Specifically, 7 relationships among first-degree relatives (18,258 pairs and 17,605 individuals), 17 relationships among second-degree relatives (15,114 pairs and 8,253 individuals), and 10 relationships among third-degree relatives (4,634 pairs and 2,686 individuals).

These various familial relationships, ranging from first-degree to third-degree relatives, enable us to estimate X chromosome heritability more accurately.

We believe that these enhancements significantly improve the robustness and precision of our estimates. Thank you once again for your valuable suggestion, which has greatly contributed to the refinement and enhancement of our methodology.

(Major.2.8) ignore the sexual difference in autosomal heritability. A good reason not to make this assumption.

(Response.2.8) Thank you for your valuable feedback regarding the consideration of sexual differences in autosomal heritability. We recognize the importance of addressing this aspect to improve the accuracy and reliability of our heritability estimates.

In our previous model (XCHROM), we operated under the assumption of equal variances for both sexes. However, our new model, BIGFAM, does not assume equal autosomal heritability for both sexes. This is a significant improvement that allows us to more accurately estimate heritability by accounting for potential sex-specific differences.

In our revised model, the common variance components (autosomal heritability and shared environmental variance component) are regressed out within each bin before estimating X chromosome heritability. Specifically, to alleviate the effect of sex-specific variance components, we stratified the FR-reg coefficients into bins based not only on the same degree of relatedness but also on the same sex pairs. This process is described in the **Methods** section of our manuscript as follows:

Within each bin, FR-reg coefficients ($\hat{\lambda}_l$) were regressed out by the average of FR-reg coefficients as follows:

$$\lambda_{d,ss} = \frac{1}{|L_{d,ss}|} \sum_l^{L_{d,ss}} \hat{\lambda}_l$$

where $L_{d,ss}$ is the set of familial relationships in the d -bin and same sex pair (ss). ss can be male-male, female-female, and male-female pair.

Then, the expectation of residual of FR-reg coefficient for l relation ($\hat{\lambda}_l^{res}$) can be formulated as follows:

$$\begin{aligned} E[\hat{\lambda}_l^{res}] &= E[\hat{\lambda}_l] - E[\lambda_{d,ss}] \\ &= (E[r_{a,l}] \cdot V_a + E[r_{X,l}] \cdot V_X + E[r_{s,l}] \cdot V_s) \\ &\quad - (E[r_{a,d,ss}] \cdot V_a + E[r_{X,d,ss}] \cdot V_X + E[r_{s,d,ss}] \cdot V_s) \\ &= t_l \cdot V_X^2 + \epsilon_s \end{aligned}$$

where t_l represents $E[r_{X,l}] - E[r_{X,d,ss}]$ and ϵ_s is $(E[r_{s,l}] - E[r_{s,d,ss}]) \cdot V_s$.

By incorporating this stratification and regression process, we effectively account for sex-specific variance components, leading to more accurate estimates of X chromosome heritability.

We believe that these improvements address your concerns and enhance the robustness of our methodology. Thank you once again for your valuable feedback, which has been instrumental in refining our approach and ensuring the accuracy of our results.

(Major.2.9) more parameters settings for the simulation. Currently the simulation study was performed under a single parameter setting. “The proportion of phenotypic variance due to each random variable was set to (0.5, 0.02, 0.05, 0.05, 0.1, 0.28) for (σ_{aut} # , σ_X # , $\sigma_{1\text{fam}}$ # , $\sigma_{1\text{po}}$ # , $\sigma_{1\text{sib}}$ # , σ_2 #) where σ_{aut} # 140 is the variance by the autosomal variant, σ_X # is the variance by the X chromosome.” The additive genetic effects is set to explain 50% of the phenotypic variance, which is quite high. On average across traits, autosomal heritability is about 15% in human complex traits. A higher additive genetic effects makes the model easier to split that from other small effects. What if the additive genetics is set to be much smaller? Will X-CHROM performs equally well?

(Response.2.9) Thank you for your insightful comment regarding the parameter settings used in our simulation study. We recognize the importance of testing our model under a variety of conditions to ensure its robustness and applicability to real-world scenarios.

Our previous simulation study was conducted under a single parameter setting, which may not fully capture the variability and nuances present in real-world scenarios. To address this concern, we have extended our simulation to encompass a broader range of parameter settings. Specifically, we conducted simulations with different sets of variance components to assess the performance of our model under varying conditions.

For the estimation of X chromosome parameters, we simulated four different sets of variance components for autosomes (V_a) at 0.6, 0.4, 0.2, and 0.1, and for the X chromosome (V_X) at 0.03, 0.02, 0.01, and 0.005. This range of settings allowed us to evaluate the model’s performance with lower additive genetic effects. The detailed results of these simulations are thoroughly depicted in **Supplementary Fig. 4** and **Supplementary Fig. 5** of our revised manuscript.

Our extended simulations demonstrated that BIGFAM consistently approximates the true V_x across these varied parameter settings. These results indicate that our model performs well even when the additive genetic effects are set to be much smaller.

We believe that these extended simulations provide a comprehensive validation of our model's robustness and reliability under diverse conditions. Thank you once again for your valuable feedback, which has significantly contributed to the refinement and enhancement of our study.

Minor comments

(Minor.2.1) Interpretation of DC. I believe that the mechanism of DC does not change across traits. What changes is the effect of DC mechanism on traits depending on the location of trait loci. If QTLs of a trait are enriched in X-inactive region, we are more likely to detect DC ratio close to 2 using quantitative genetics method. If QTLs of a trait loci enriched in regions escape from X-inactivation, we are more likely to detect DC ratio close to 0.5. The authors need to interpret the results more carefully.

(Response) Thank you for your valuable feedback regarding the interpretation of the dosage compensation ratio (DCR). You have raised an important point about the consistency of the dosage compensation mechanism across traits and the influence of the distribution of trait loci on the X chromosome.

As rightly pointed out, the mechanism of dosage compensation remains consistent across traits, while the observed DCR may vary depending on the distribution of trait loci on the X chromosome. Indeed, studies such as those by Sidorenko et al⁷. and San Roman et al.² have demonstrated that gene expression on the X chromosome exhibits regional specificity, which can influence the observed DCR.

In our revised manuscript, we have provided a thorough explanation of dosage compensation and the interpretation of DCR results in the **Results** section. We have emphasized the regional specificity of gene expression on the X chromosome and its potential impact on the observed DCR values as follows:

Due to X chromosome inactivation, heritability on the X chromosome is higher in males than in females. To quantify this difference, we calculated the dosage compensation ratio (DCR) for each trait. The DCR represents the ratio of X chromosome heritability in males to that in females ($DCR = V_{X,male}/V_{X,female}$) and quantifies the degree of X chromosome inactivation. Under the assumption of full dosage compensation (FDC), where one of the X chromosomes in females is randomly inactivated, the DCR is 2. Conversely, under the assumption of no dosage compensation (NDC), where no X chromosome inactivation occurs, the DCR is 0.5. If the mechanism of dosage compensation (DC) does not change across traits, the DCR values can provide insights into the X-inactivation profiles of specific traits. For example, if the quantitative trait loci (QTLs) of a trait are enriched in the X-inactive region, we are more likely to detect a DCR close to 2. Conversely, if the QTLs of a trait are enriched in regions that escape from X-inactivation, we are more likely to detect a DCR close to 0.5.

We believe that these clarifications will enhance the understanding of our results and provide a more nuanced interpretation of the DCR values. Thank you once again for your insightful feedback, which has significantly contributed to the refinement of our manuscript.

(Minor.2.2) Correlation between X-CHROM and GCTA estimates is nearly 0. As the dots in figure 4a are too small and unclear, the reviewer estimated the correlation of X chromosome heritability between X-CHROM estimates in Supplementary Table 4 and GCTA estimates in Supplementary Table 5 across 64 traits and found that the correlation in males is 0.037 ± 0.124 with p value of 0.770 and in females is -0.019 ± 0.125 with p value of 0.88. There is no correlation between X-CHROM X heritability estimates and GCTA X chromosome heritability estimates, although the autosomal heritability from GCTA does correlated nicely with X-CHROM autosomal heritability in males (0.682 ± 0.057 with p value of 5.51×10^{-8}). This further confirms that the X-CHROM does not provide robust estimates of X chromosome heritability.

(Response) Thank you for your detailed feedback regarding the correlation between X-CHROM and GCTA estimates. We appreciate the opportunity to clarify and address these concerns.

In response to your observation about the clarity of the dots in the correlation plot, we have enlarged the dots and added a regression line for better understanding of the correlation. These updates can be found in **Fig.7a** of our revised manuscript.

Regarding the correlation between X chromosome heritability estimates from X-CHROM and GCTA, our previous model (XCHROM) indeed showed nonsignificant correlations with SNP heritability computed from GCTA.

In our revised model (BIGFAM), we have addressed these limitations and achieved significant correlations with GCTA estimates. Specifically, the overall correlation between BIGFAM and GCTA was 0.942 (with 95% CIs of (0.925,0.952)). For males, the correlation was 0.949 (with 95% CIs of (0.937,0.970)), and for females, the correlation was 0.650 (with 95% CIs of (0.539,0.718)). These significant correlations indicate that BIGFAM provides more robust and reliable estimates of X chromosome heritability compared to our previous model.

We believe these improvements address the concerns raised and demonstrate the enhanced performance of our revised model. Thank you once again for your valuable feedback, which has greatly contributed to the refinement and validation of our methodology.

Reviewer #3

Major comments

(Major.3.1) The difference between X-CHROM and GCTA-GREML is striking, making me think it would be great to try this in larger datasets, e.g. finngen, deCODE, All-of-Us? Lack of power seems to be a key limitation of this study. More discussion (and simulations?) on how large data are needed for this to work well would be good. What does 100K pairs really correspond to?

(Response.3.1) Thank you for your insightful comment regarding the power and applicability of our model in larger datasets. We recognize that sample size is a crucial factor in obtaining robust and reliable heritability estimates. To address this concern, we have made several improvements and plan to leverage additional datasets.

Firstly, we have incorporated a broader range of relatives, including second- and third-degree relatives, in addition to first-degree relatives. This expansion increases the sample size and enhances the robustness of our analysis. Our revised analysis now includes 22 distinct familial relationships. Specifically, from the UK Biobank (UKB) cohort, we have 7 relationships among first-degree relatives (10,404 pairs involving 19,348 individuals), 7 familial relationships among second-degree relatives (2,454 pairs involving 4,743 individuals), and 8 familial relationships among third-degree relatives (14,754 pairs involving 27,069 individuals) from UK Biobank (UKB) cohort.

In addition to the UKB dataset, we have also utilized the Generation Scotland: Scottish Family Health Study (GS:SFHS), which contains pedigree information from 4,830 families. From this dataset, we inferred 34 distinct familial relationships. Specifically, 7 relationships among first-degree relatives (18,258 pairs and 17,605 individuals), 17 relationships among second-degree relatives (15,114 pairs and 8,253 individuals), and 10 relationships among third-degree relatives (4,634 pairs and 2,686 individuals).

Furthermore, we have developed a more sophisticated model, BIGFAM, to address these issues. BIGFAM estimates X chromosome heritability by regressing out common variance components from FR-reg coefficients using L2 penalty optimization. As a result, 20 phenotypes exhibited significant non-zero X chromosome heritability in our model. Additionally, our results demonstrated a substantial correlation with SNP-based heritability estimation (0.942 with 95% CIs (0.925,0.952)).

Along with estimating X chromosome heritability, our refined model can disentangle the variance component by genetic and shared environmental effects. Because BIGFAM uses FR-reg coefficients as an input to estimate variance components within the relative pairs, the standard error of FR-reg is crucial for precise results. As shown in **Supplementary Fig. 1**, we simulated the results of the slope test of BIGFAM depending on the standard error of FR-reg. We recommend that the standard error of the FR-reg coefficient be smaller than 0.01 to obtain precise results, although even smaller standard errors would yield more accurate estimates.

Thank you once again for your valuable feedback, which has significantly contributed to the refinement and enhancement of our methodology and analysis. We are confident that these improvements will address the limitations related to sample size and power, thereby providing more robust and reliable heritability estimates.

(Major.3.2) The genetic architecture of metabolic outcomes is generally rather oligogenic, which can lead to unnecessarily high variance when estimating heritability using gcta/LDSC. As an alternative you can try SBayesS or LDpred2-auto.

(Response.3.2) Thank you for your valuable suggestion regarding the use of alternative methods such as SBayesS or LDpred2-auto for estimating heritability, especially given the oligogenic nature of metabolic outcomes. We appreciate your insights and have considered these methods in our analysis.

Methods like LDSC and LDpred2-auto are commonly used for heritability estimation; however, they focus solely on the autosomes and do not officially support heritability estimates for the X chromosome. In contrast, GCTA supports the estimation of X chromosome heritability with various dosage compensation parameters. Therefore, we compared the X chromosome heritability estimates derived from our method with those obtained from GCTA under the equal variance assumption across sexes.

In our revised model, BIGFAM, we estimate not only the X chromosome heritability but also the autosomal heritability. Consequently, we compared the autosomal heritability estimates from BIGFAM with those obtained from LDSC and LDpred2-auto.

As noted by the reviewer, LDpred2-auto showed smaller variance in heritability estimates compared to LDSC (approximately 67% of LDSC) with high correlation (squared Pearson correlation of 0.986 with a p-value of $1.306 \cdot 10^{-12}$). The result showed that our method, BIGFAM, demonstrated a high degree of correlation with other approaches in estimating autosomal heritability. For instance, the squared Pearson correlation between BIGFAM and LDSC was 0.927 (p-value of $3.586 \cdot 10^{-8}$), and between BIGFAM and LDpred2-auto was 0.963 (p-value of $5.791 \cdot 10^{-10}$). These results are depicted in **Fig. 6** of our revised manuscript.

By incorporating these alternative methods and demonstrating the robustness of our approach through various comparisons, we believe that our analysis provides a comprehensive and reliable estimation of heritability for both autosomes and the X chromosome.

Thank you once again for your valuable feedback, which has significantly contributed to the refinement and enhancement of our study.

(Major.3.3) Consider GWAS summary statistics when estimating heritabilities, e.g. using LDSC/SumHer or even SBayesS/LDpred2-auto. This could allow you to increase the sample sizes, to see if that leads to better replication of X-CHROM.

(Response.3.3) Thank you for your valuable suggestion regarding the use of GWAS summary statistics for estimating heritabilities. We appreciate your insights and have considered these methods in our analysis.

GWAS summary statistics methods like LDSC and LDpred2-auto are commonly used for heritability estimation. However, they focus solely on the autosomes and do not officially support heritability estimates for the X chromosome. In contrast, GCTA supports the estimation of X

chromosome heritability with various dosage compensation parameters. Therefore, we compared the X chromosome heritability estimates derived from our method with those obtained from GCTA under the equal variance assumption across sexes.

Instead, to increase the sample sizes and improve the robustness of our analysis, our revised study includes 314,699 unrelated individuals (169,019 females and 145,680 males) for the GCTA analysis. This is a significant increase from our previous manuscript, which used only 20,000 individuals. To manage this large sample size, we partitioned them into 30 blocks, computed the genetic relationship matrix (GRM) for each block, and performed GREML analysis. We then merged the estimates using inverse-variance weighting (IVW) to obtain tighter standard errors on the X chromosome SNP heritability estimates.

Additionally, our revised model, BIGFAM, also utilized data from 48,912 individuals (27,612 pairs) from first- to third-degree relatives in the UK Biobank (UKB) dataset. As a result, BIGFAM demonstrated a substantial correlation with GCTA (0.942 with 95% CIs (0.925,0.952)) on X chromosome heritability. These results are depicted in **Fig. 7a** of our revised manuscript.

Furthermore, BIGFAM estimates not only X chromosome heritability but also autosomal heritability. For autosomal heritability, we compared the results of BIGFAM with various methods, including summary statistics-based methods such as LDSC and LDpred2-auto. The results showed that our method, BIGFAM, demonstrated a high degree of correlation with other approaches in estimating autosomal heritability. For instance, the squared Pearson correlation between BIGFAM and LDSC was 0.927 (p-value of $3.586 \cdot 10^{-8}$), and between BIGFAM and LDpred2-auto was 0.963 (p-value of $5.791 \cdot 10^{-10}$). These results are depicted in **Fig. 6** of our revised manuscript.

By incorporating these alternative methods and increasing our sample size, we believe that our analysis provides a comprehensive and reliable estimation of heritability for both autosomes and the X chromosome. Thank you once again for your valuable feedback, which has significantly contributed to the refinement and enhancement of our study.

(Major.3.4) It would be great to use simulations to explore the impact of ignoring sex-specific differences in autosomal heritability and unaccounted shared environment.

(Response.3.4) We sincerely appreciate the reviewer's insightful comments and suggestions regarding the use of simulations to explore the impact of ignoring sex-specific differences in autosomal heritability and unaccounted shared environment.

In response to this concern, we have made significant improvements in our revised model, BIGFAM. Unlike our previous model, BIGFAM does not assume equal autosomal heritability for both sexes. Consequently, BIGFAM can utilize not only same-sex pairs but also all kinds of familial relationships, providing a more comprehensive and accurate estimation of heritability.

Regarding the consideration of shared environmental factors, we conducted simulations to explore various relationship-specific shared environmental scenarios, including those involving first-degree specific shared environment, maternal effects, and sibling-specific shared environmental scenarios. These simulations helped us assess how BIGFAM performs under different shared environmental conditions. The results, as depicted in **Fig. 4**, indicate that BIGFAM effectively approximates autosomal heritability and X chromosome heritability across all simulated scenarios, demonstrating the robustness of our approach.

Additionally, we have addressed potential factors contributing to slight overestimations observed in some scenarios. Specifically, we discuss the possibility of a correlation between genetic correlation on the X chromosome and shared environmental correlation, which could induce overestimation in the estimates. This explanation has been clarified in the **Results** section of our revised manuscript as follows:

In S_3 and S_4 , where we modeled maternal and sibling-specific shared environmental effects, the estimated V_X were slightly higher compared to S_1 and S_2 . This is possibly because relationships having higher V_S also have higher genetic correlation on the X chromosome ($r_{X,l}$). For example, the $r_{X,l}$ of mother-offspring relationships ($\frac{1}{2}$ and $\frac{1}{\sqrt{2}}$) is higher than father-offspring relationships ($\frac{1}{\sqrt{2}}$ and 0). Our model assumed independence between these effects, and therefore, when $r_{X,l}$ is correlated with the shared environmental effect, the estimated V_X can be slightly higher than the true value.

We are grateful for the opportunity to address these concerns and enhance the rigor of our methodology. Your feedback has been invaluable in guiding these improvements, and we remain committed to ensuring the integrity and reliability of our research.

(Major.3.5) No case-control outcomes are considered, presumably due to even lower power and scale? I would nevertheless be interested in seeing some results for these, or at the minimum discussed.

(Response.3.5) We sincerely appreciate the reviewer’s insightful comments and suggestions regarding the consideration of case-control outcomes in our study. As noted, our current methodology, BIGFAM, focuses on estimating heritability for continuous traits and does not support heritability estimation for case-control studies.

This limitation stems from the challenges associated with applying the liability threshold model, which is commonly used for estimating heritability of binary traits, to close relatives. The presence of both additive and non-additive effects in the estimation process can lead to biased

estimates. We describe this limitation on the **Discussion** section on our revised manuscript as follows.

BIGFAM, while innovative, is subject to several limitations and questions that remain open for further research. ... Fifthly, BIGFAM does not currently support variance component analysis for binary traits. Heritability estimation for binary traits often relies on the liability threshold model. However, this model may lead to biased estimates when applied to close relatives due to the influence of both additive and non-additive effects on the estimation process³³. Therefore, further research is needed to estimate heritability for binary traits using close relatives.

Nevertheless, we acknowledge the importance of exploring heritability estimation for binary traits and recognize the potential value it holds for understanding the genetic basis of complex diseases. While our current methodology does not accommodate case-control outcomes, we are actively exploring avenues for extending our approach to incorporate binary traits in the future.

We believe that further research and development, such as gaining additional insights into genetic and environmental factors or refining our modeling techniques, could enable the estimation of heritability for binary traits in close relatives. As we continue to refine and expand our methodology, we will explore the possibility of incorporating case-control outcomes and discuss these implications in future studies.

Your feedback is invaluable in guiding our research efforts, and we are committed to ensuring that our methodology evolves to address a broader range of traits and study designs.

(Major.3.6) Please examine whether the Y chromosome impacts the method, or explain why it wont.

(Response.3.6) We sincerely appreciate the reviewer's insightful comments and suggestions regarding the potential impact of the Y chromosome on our methodology. The Y chromosome is exclusively passed down from father to son through male lineages. However, its contribution to overall heritability estimates for complex traits is relatively minor compared to other chromosomes.

This is primarily attributed to the Y chromosome's smaller size and lower gene content. Specifically, the Y chromosome spans approximately 57 million base pairs (bp) and contains only around 200 genes, whereas the X chromosome, with a length of 156 million bp, contains over 1400 genes. This limited gene content restricts its potential contribution to the total genetic variation underlying complex traits.

If the Y chromosome does exhibit significant heritability, the heritability estimation from BIGFAM could potentially introduce bias. However, the impact of such bias may diminish when analyzing multiple relatives, as the averaging effect across relatives can help mitigate these biases.

We will include a detailed explanation of the Y chromosome's limited impact on heritability estimates in the **Supplementary text** of our revised manuscript as follows.

However, the Y chromosome's contribution to overall heritability estimates tends to be relatively minor. This is primarily due to its smaller size and lower gene content compared to other chromosomes. For instance, the Y chromosome spans approximately 57 million base pairs (bp) and contains only around 200 genes, whereas the X chromosome, with a length of 156 million bp, contains over 1400 genes. This limited gene content restricts its contribution to the total genetic variation underlying complex traits⁴. If

the Y chromosome does exhibit significant heritability, its inclusion in heritability estimation could potentially introduce bias. Nonetheless, the impact of such bias may diminish when analyzing multiple relatives, as the averaging effect across relatives can help mitigate these biases.

Your feedback has been instrumental in ensuring the thoroughness and accuracy of our methodology, and we are committed to addressing any potential concerns regarding the inclusion of genetic factors in our analyses.

(Major.3.7) Regarding the methods, it's interesting that you estimate this using gradient descent in a two step procedure. I believe one could perhaps avoid the second step and obtain a better fit by jointly estimating the heritability and dosage compensation. This may however be difficult to do, and would be a different method.

(Response.3.7) We sincerely appreciate the reviewer's insightful comments and suggestions regarding our methodology for estimating heritability and dosage compensation.

In response to your feedback, we have refined our approach in the revised method, BIGFAM, by incorporating a two-step optimization procedure. Firstly, we perform a regression step to remove common variance components in the familial relationship regression (FR-reg) coefficients. Subsequently, in the second step, we optimize the estimation of X chromosome heritability using a regularization technique with an L2 penalty. The role of the L2 penalty is to shrink the standard error of the estimate, leading to more robust results.

While the suggestion of jointly estimating heritability and dosage compensation in a single step is intriguing, we recognize the potential instability that may arise due to the interdependence between dosage compensation and X chromosome heritability. Instead, we have opted to incorporate genetic correlation on the X chromosome to ensure more stable estimates. This refined approach allows us to achieve more reliable results without compromising on stability.

Furthermore, in addition to estimating X chromosome heritability, BIGFAM also aims to partition the variance components by genetic and shared environmental effects. To achieve this, we have implemented another two-step approach involving a slope test and prediction with optimization. In the slope test step, we differentiate between the decaying patterns of shared environmental and genetic effects, with the former decreasing at half the rate of the degree of relatedness. Subsequently, in the optimization step, we partition the variance components by genetic and shared environmental effects based on the insights gained from the slope test.

The detailed description of our revised methodology can be found in the **Methods** section of our revised manuscript.

Your feedback has been invaluable in guiding the refinement of our methodology, and we are fully committed to incorporating further improvements to ensure the rigor and reliability of our research.

Minor comments

(Minor.3.1) It's impossible to read figure 2 because the font is small and quality poor (same goes for most figures).

(Response) We sincerely appreciate the reviewer's insightful comments and suggestions.

In response to this feedback, we improve the readability of all figures by increasing the font size and enhancing the overall quality. These adjustments have been made to ensure that all figures are clear and easily interpretable for readers.

Reference

1. Polubriaginof, F. C. G. *et al.* Disease Heritability Inferred from Familial Relationships Reported in Medical Records. *Cell* **173**, 1692–1704.e11 (2018).
2. Roman, A. K. S. *et al.* The human inactive X chromosome modulates expression of the active X chromosome. *Cell Genomics* **3**, (2023).
3. Tukiainen, T. *et al.* Landscape of X chromosome inactivation across human tissues. *Nature* **550**, 244–248 (2017).
4. Bernabeu, E. *et al.* Sex differences in genetic architecture in the UK Biobank. *Nat Genet* **53**, 1283–1289 (2021).
5. Fernando, R. L. & Grossman, M. Genetic evaluation with autosomal and X-chromosomal inheritance. *Theoret. Appl. Genetics* **80**, 75–80 (1990).
6. James, J. W. 353. Note: Covariances Between Relatives due to Sex-Linked Genes. *Biometrics* **29**, 584–588 (1973).
7. Sidorenko, J. *et al.* The effect of X-linked dosage compensation on complex trait variation. *Nat Commun* **10**, 3009 (2019).

**RESPONSE TO REVIEWERS' COMMENTS**

We are deeply grateful for the thorough and insightful comments provided by the four
anonymous reviewers. Their feedback has been instrumental in improving both the technical
accuracy and clarity of our manuscript. We have carefully addressed each comment and made
substantial revisions to enhance the manuscript's quality and accessibility.

We have made several significant improvements to our work based on the reviewers'
comments:

- 1. We have provided a comprehensive comparison between BIGFAM and existing
methods (particularly HE regression and REML), clarifying the key methodological
distinctions and advantages of our approach. Specifically, BIGFAM achieves more
stable variance component estimation by reducing the number of parameters to be
estimated compared to HE regression and REML.
- 2. We conducted extensive additional simulations to validate BIGFAM's performance under
various shared environmental scenarios, helping us better understand both the strengths
and limitations of our method. Specifically, we considered the scenarios where shared
environmental effects increase with degree of relationship, are dominant in second-
degree relatives, and where the shared environmental correlation with various
distribution.
- 3. We reorganized the Results section for better readability and enhanced the clarity of our
figures accordingly. Additionally, we improved our mathematical derivations and
explanations throughout the manuscript, particularly regarding the distinction between
autosomal and X-chromosomal variance component estimation.

These revisions have not only addressed the technical concerns raised by the reviewers but
have also made our methodology more accessible to a broader audience. The detailed
mathematical foundations and comprehensive simulation studies added in response to the
reviewers' comments have significantly strengthened our study.

Below, we provide point-by-point responses to each reviewer's comments. We have carefully
considered and addressed every suggestion, and we believe these changes have substantially
improved the quality and clarity of our work. Once again, we express our sincere appreciation to
the reviewers for their valuable time and constructive feedback that has helped shape this
manuscript into a stronger contribution to the field.

Reviewer 1

Major Comments

**[Major 1.1] Their new model has substantial similarity with HE regression. It would be**
**good to have a formal comparison.**

**(Response 1.1)** We sincerely appreciate your insightful comment regarding the comparison
between BIGFAM and HE regression. Both methods share a fundamental similarity in their
ability to estimate heritability using relatives' data without genotype information. However,
BIGFAM differs from HE regression in its ability to stably estimate variance components in
datasets with multiple degrees of relatives and to estimate X chromosome variance components
without genotype data. We acknowledge the importance of addressing these issues to further
elaborate on the distinctions between the two methods.

The key distinction between BIGFAM and HE regression lies in parameter reduction strategy of
BIGFAM through assumptions about shared environmental effects across different degrees and
relationships.

For the first objective of BIGFAM (partitioning genetic and shared environmental variance
components), we introduce an assumption that shared environmental correlation decays at a
specific rate (parameterized as w_S) as the degree of relationship increases. This enables us to
estimate shared environmental effects using just two parameters (V_S and w_S) rather than
estimating separate effects for each degree. In contrast, HE regression faces convergence
issues when attempting to estimate multiple shared environmental effects. Specifically, when
incorporating additional shared environmental relatedness matrices for each degree of
relatedness alongside the pedigree-based genetic relatedness matrix (GRM), the model fails to
converge due to multicollinearity between matrices and their inherent sparsity. We discuss this
issue in detail in the **Supplementary Text** as follows.

*... the model can be further modified to include shared environmental effects:*

$$(y_i \cdot y_j) \sim a_{ij}^* \cdot V_A + \sum_d^D 1_{(i,j) \in S_d} \cdot V_{S_d} + 1_{i=j} \cdot V_E$$

*where $1_{(i,j) \in S_d}$ is an indicator vector that equals 1 if (i, j) is a d -degree relative and 0*
*otherwise. This modification faces the same convergence issues as the REML model without*
*genotype data discussed earlier. Although the estimation approach differs (REML uses matrix*
*operations with the AI matrix, while this approach vectorizes relatedness matrix components*
*for linear regression), both suffer from multicollinearity, sparsity, and the large number of*
*parameters to estimate. In contrast, BIGFAM takes a fundamentally different approach to*
*overcome these limitations. Instead of attempting to estimate all parameters simultaneously, it*
*employs a two-step procedure: using a slope test to estimate the pattern of shared*
*environmental decay, then estimating variance components and the shared environmental*
*decaying factor in the prediction step. This strategic reduction in the number of parameters to*
*estimate effectively addresses the numerical instability issues when analyzing data with*
*multiple degrees of relatives.*

For the second objective of BIGFAM (estimating X chromosome variance component), the key
innovation lies in leveraging relationship-specific patterns of X chromosome inheritance. While
genetic correlations on the X chromosome ($r_{X,l}$) vary distinctly across different familial
relationships (even within the same degree), other components remain relatively constant within
each degree of relatedness. BIGFAM exploits this property by first regressing out all variance

components except X chromosome effects, effectively reducing the estimation to a single
parameter problem that can be solved using only well-documented patterns of X chromosome
inheritance, without requiring genetic data. While HE regression can theoretically estimate X
chromosome effects using pedigree-based genetic relatedness, it produces unreliable estimates
in practice, with implausibly large heritability and wide confidence intervals. For example, in *leg*
*fat-free mass*, HE regression estimated that the X chromosome alone explains nearly 70% of
phenotypic variance (0.962 with 95% CIs: [0.571, 0.812]), while BIGFAM provided a more
biologically plausible estimate of 0.016 (95% CIs: [0.005, 0.030]). We discuss this issue in detail
in the **Supplementary Text** as follows.

*... It is theoretically possible to estimate variance components of the X chromosome (V_X) by*
*incorporating pedigree-based X chromosome genetic relatedness (x_{ij}^*) as follows:*

$$(y_i - y_j)^2 \sim 2 \cdot V_P - 2a_{ij}^* \cdot V_A - 2x_{ij}^* \cdot V_X.$$

*However, the estimation produces unreliable results in practice. Specifically, excessively large*
*standard errors and biologically implausible estimates where X chromosome variance*
*components highly exceed autosomal variance components. When estimating the V_X , BIGFAM*
*addresses this limitation through a fundamentally different approach: it first removes both*
*shared environmental effects and autosomal genetic components that are common to each*
*degree of relationship, thereby isolating the X chromosome effects. This approach reduces the*
*complexity of the estimation problem to focus solely on X chromosome variance components,*
*leading to more stable estimates. Furthermore, BIGFAM provides additional numerical*
*stability by using L2 penalty.*

We have provided a formal comparison between BIGFAM and HE regression with detailed
mathematical derivations and analyses in the **Supplementary Text**. Additionally, we have
added a new section in the **Results** to describe these key differences as follows:

*... Existing methods such as REML[1][2] and Haseman-Elston regression[3] can adapt to*
*situations where genotype data is unavailable by replacing genotype-based relatedness*
*measures with pedigree-based information. In these pedigree-based approaches, genetic*
*correlation is determined by the degree-of-relatedness of pair: unrelated pairs are assigned 0,*
*while d-th degree relatives are assigned 2^{-d} based on expected genetic correlation. However,*
*this approach fits phenotypes to a single genetic relatedness matrix (GRM). When applied to*
*relatives, such fitting can lead to biased estimates of V_G due to shared environmental*
*effects[3].*

We are grateful for your insightful comment, which has prompted us to provide a more
comprehensive comparison between BIGFAM and HE regression. This detailed examination
has helped clarify the key methodological differences and strengthen our manuscript. If there
are any further suggestions or inquiries, we are eager to address them.

**[Major 1.2] The assumed decay of the shared environment with increasing degree of**
**relationship is a very strong assumption. Why do the authors assume any particular**
**form? Different types of shared environments must have distinct forms of variance decay**
**with increasing degree of relatives. For instance, some environments are likely shared**
**just within nuclear families, while others are shared with more distant relatives. Those**
**two situations would have different functions with regard to relationships. How would**
**these impact the resulting estimates? What happens when the form of the decay is**
**misspecified? Fig 4 suggests this is a significant problem. Fig 5b shows a clear trade off**
**between V_g and V_s . that could represent real patterns where most traits have either**
**genetic OR shared environment variance, but it seems striking that it's mostly one or the**
**other. IS there evidence from the literature for those specific traits that it really is that?**
**Polderman's meta analysis of twin correlations would be a good point of comparison**
**here. Related, do the same traits measured in UKB and GS show the same patterns here?**
**Are their estimates of V_s and V_g the same?**

**(Response 1.2)** We appreciate your insightful comment regarding our assumptions about
shared environmental decay. We acknowledge that different types of shared environments
might have distinct decay patterns across degrees of relatives, and that our assumption of a
consistent decay pattern could be invalid in certain scenarios. Additionally, as most previous
research on shared environment has focused on twin studies, evidence for environmental
changes across degrees is limited. To address these concerns, we conducted additional
simulations with diverse shared environmental scenarios.

First, we examined scenarios where shared environmental effects increase with degree of
relationship, contrasting with our original assumption of decreasing effects. Our slope test (first
step of BIGFAM to partition variance components of genetic and shared environmental effects)
effectively distinguished these cases, consistently yielding negative slopes for increasing
patterns, markedly different from the positive slopes observed in decreasing patterns. In
analysis on real data from UKB and GS:SFHS, we observed that none of the phenotypes
showed negative slopes, suggesting that shared environmental effects tend to decrease with
increasing degrees of relationship across most phenotypes. These results are described on
**Fig.2a** (simulation), **Fig.2c** (UKB) and **Fig.S4a** (GS:SFHS).

To further evaluate BIGFAM's performance under various shared environmental patterns, we
tested four scenarios: In the reference scenario (*SC1*), shared environmental effects followed a
standard decay pattern where the correlation decreases by a factor of w_s with each degree. For
the nuclear-family-specific scenario (*SC2*), shared environmental effects were present only in
first-degree relatives and set to zero for more distant relationships. The maternal-effect scenario
(*SC3*) modeled enhanced environmental effects in mother-offspring relationships, where these
relationships showed 1.5-fold stronger environmental correlation compared to other first-degree
relationships. In the second-degree-dominant scenario (*SC4*), environmental effects were
strongest in second-degree relatives.

In the nuclear-family-specific scenario (*SC2*), extremely large w_s values ($w_s \approx 100$) effectively
captured the sharp environmental decline between first- and second-degree relatives. The
maternal-effect scenario (*SC3*) maintained accurate genetic estimates ($V_G = 0.506$, $SD = 0.010$),
while its performance varied with underlying true decay patterns: under fast decay ($w_s > 2$),
estimates remained accurate despite maternal effects; under similar decay ($w_s \approx 2$), maternal
effects actually improved estimation by creating distinguishable decay rates; under slow decay
($w_s < 2$), maternal effects led to substantially different decay rates between first-to-second and
second-to-third degrees, reducing estimation accuracy. The second-degree-dominant scenario
(*SC4*) yielded large uncertainties, indicating potential challenges when environmental effects

deviate significantly from expected patterns. The results were plotted on **Fig.2b**, and **Fig.S3** and
we explained this in detail in the **Results** as follows.

*... The results demonstrated robust prediction performance across scenarios, with each*
*scenario offering unique insights into how different patterns of shared environmental effects*
*are captured in our estimates. In the reference scenario (SC1), the prediction showed robust*
*performance across both fast and slow decay patterns but not in similar decay pattern (Fig.2b*
*and Fig.S2), which is expected as genetic and shared environmental components become*
*statistically indistinguishable when they decay at the same rate. ...Finally, in the second-*
*degree-specific scenario (SC4), the prediction yielded large uncertainties in its estimates,*
*demonstrated by wide confidence intervals (CIs including zero for all 1,000 simulation). This*
*suggests that reliable estimation may be challenging when shared environmental effects*
*deviate significantly from the expected decay pattern.*

These simulation results demonstrate that BIGFAM can effectively partition V_G and V_S across
various shared environmental scenarios when environmental effects decrease with increasing
degrees of relationship. However, when this assumption is violated, BIGFAM produces
estimates with large confidence intervals, effectively indicating non-significant variance
component estimates. We acknowledge this limitation in the **Discussion** section as follows:

*... Another limitation is the assumption about the consistency of shared environmental decay*
*parameters (w_s) across degrees of relatedness. If w_s varies by degree of relatedness, as shown*
*in our simulations, the precision of heritability estimates in BIGFAM could be compromised.*

Regarding the apparent trade-off between V_G and V_S in our previous **Fig.5b**, this pattern
emerged from our initial filtering of phenotypes based only on significant V_G . Based on our
simulation results showing that non-significant estimates (95% CIs including 0) could be
unreliable, we revised our analysis to filter out phenotypes where both V_G and V_S were non-
significant. The updated results show some traits like immature reticulocyte fraction (IRF) and
neutrophil percentage show high V_S with low V_G , others like standing height show high values for
both components. And most phenotypes demonstrate modest shared environmental effects
($V_S \approx 0.05$). This results is described on **Fig.2d** (UKB) and **Fig.S4b** (GS:SFHS).

Even though the limited phenotype overlap between UKB and GS:SFHS (only three phenotypes
(FVC, cholesterol, and Urea) were common in both cohorts), interestingly, we observed that
forced vital capacity (FVC) showed significant (*slow*) decay of shared environmental effects in
both cohorts. Despite having consistent slow decay patterns of shared environmental effects
across both cohorts, FVC demonstrated similar genetic variance components ($V_G = 0.331$, 95%
CIs = [0.327, 0.334] in GS:SFHS; $V_G = 0.333$, 95% CIs = [0.332, 0.334] in UKB) but notably
different shared environmental effects ($V_S = 0.096$, 95% CIs = [0.0945, 0.098] in GS:SFHS; $V_S =$
0.025 , 95% CIs = [0.025, 0.026] in UKB). This suggests that while genetic architecture might be
preserved across populations, shared environmental effects can be cohort-specific, even when
the pattern of environmental decay remains consistent. We describe this results in the **Results**
as follows.

*Third, we observed an interesting pattern in forced vital capacity (FVC), which showed*
*significant (slow) decay of shared environmental effects in both cohorts. Despite having*
*consistent slow decay patterns of shared environmental effects across both cohorts, FVC*
*demonstrated similar genetic variance components ($V_G = 0.331$, 95% CIs = [0.327, 0.334] in*
*GS:SFHS; $V_G = 0.333$, 95% CIs = [0.332, 0.334] in UKB) but notably different shared*
*environmental effects ($V_S = 0.096$, 95% CIs = [0.0945, 0.098] in GS:SFHS; $V_S = 0.025$, 95%*
*CIs = [0.025, 0.026] in UKB). This suggests that while genetic architecture might be*

*preserved across populations, shared environmental effects can be cohort-specific, even when*
*the pattern of environmental decay remains consistent.*

These comprehensive analyses are presented in **Figure 2**, which illustrates: (a) the slope test
results distinguishing increasing and decreasing environmental patterns, (b) BIGFAM's
performance across different shared environmental scenarios, (c) the observed slope patterns
in UKB phenotypes, and (d) the distribution of genetic and shared environmental components
after filtering for significant estimates.

We are grateful for this comment as it prompted us to explore BIGFAM's performance under a
broader range of shared environmental scenarios, helping us better articulate both the strengths
and limitations of our method. If there are any further suggestions or inquiries, we are eager to
address them.

**[Major 1.3] The new model would have the same individuals in multiple sets of the**
**analyses, as individuals would have multiple types of relationships in these pedigrees. I**
**might have missed it, but I didn't see how the authors deal with that non-independence**
**within their datasets.**

**(Response 1.3)** We appreciate your important observation regarding the potential non-
independence of samples when individuals have multiple relationships. In large pedigree
datasets, if certain individuals or families have more relationships than others, the familial
relationship regression (FR-reg) coefficients could be biased towards the shared environmental
effects of these overrepresented families.

To address this concern, we have modified our bootstrap resampling strategy in calculating FR-
reg coefficients and standard errors. For each bootstrap iteration, we now randomly select only
one relationship per individual. This means if an individual has relationships with multiple
relatives (e.g., a person who has two sisters), only one of these relationships is randomly
selected for each bootstrap iteration. We have added this detail to the **Methods** section as
follows:

*Additionally, to minimize the potential bias from overrepresentation of certain families in the*
*estimation process, we implement a following bootstrap resampling strategy. For each*
*bootstrap iteration, we randomly select only one relationship per individual. This means if an*
*individual has relationships with multiple relatives (e.g., a person who is both a parent and a*
*sibling), only one of these relationships is randomly selected for each bootstrap iteration. This*
*approach helps minimize potential effects from sample non-independence.*

Furthermore, we analyzed the distribution of relationships per individual in our datasets to better
understand the extent of this issue. In UKB, the majority of individuals (75.91%) have only one
relationship, with a very small fraction (1.32%) having three or more relationships. In GS:SFHS,
due to its pedigree-based data acquisition, while most individuals (76.51%) have more than one
relationship, 95% have seven or fewer relationships. These demographic details are presented
in **Fig.S12**.

a

b

Importantly, BIGFAM's estimation process naturally mitigates some of these independence
 concerns by analyzing relationships at different levels. When partitioning V_G and V_S , we estimate
 FR-reg coefficients by degree-of-relatedness, and when estimating V_X , by specific familial
 relationships. For example, in GS:SFHS, while 95% of individuals have seven or fewer total
 relationships, this reduces to four or fewer relationships when considering only first-degree
 relatives, and two or fewer for specific relationships like mother-daughter pairs. This
 stratification helps reduce the impact of multiple relationships at each analysis level.

We are grateful for this observation as it has helped us improve our methodology and better
 address the potential impacts of sample non-independence. If you have any additional
 questions or concerns, we would be happy to address them.

**[Major 1.4] I'm struggling to understand the use case of the new model. The goal is to**
 **develop new methods of estimating genetic and shared environment variances without**
 **genotype data, and that's a great goal, with a couple of recent papers doing this in large**
 **EHR datasets. Yet the authors apply their approach using degree of relationship data**
 **from biobanks based on genome wide marker data, undermining the goal.**

**(Response 1.4)** We acknowledge your point that using biobank datasets with genotype
 information might seem to undermine our method's goal of estimating heritability without
 genotype data. Indeed, recent papers^[4] have demonstrated the possibility of estimating
 heritability from large EHR data without genotype information. While we initially attempted to
 use phenotype-only EHR data, access proved challenging due to policy, privacy concerns and
 consent issues.

However, using large datasets with genotype information (UKB, GS:SFHS) was crucial for
 validating BIGFAM against existing methods. We reasoned that if BIGFAM effectively controls
 for shared environmental effects in relatives while estimating genetic components, its results
 should correlate well with estimates from unrelated individuals.

To test this, we compared BIGFAM's estimates with those from three different methodological
 categories: summary-based methods, genotype-based methods, and other pedigree-based
 methods. While these approaches estimate slightly different aspects of heritability, high
 correlation between them would suggest successful partitioning of genetic and environmental
 components. The results showed that BIGFAM achieved strong correlations with both summary-
 based and genotype-based methods (r^2 ranges from 0.729 to 0.791), outperforming existing
 pedigree-based methods like SEM ($r^2 = 0.628$). Notably, BIGFAM's correlation with LDpred2
 was particularly strong ($r^2 = 0.804$, 95% CIs: [0.620, 0.891]), demonstrating that our method
 successfully partitions genetic and shared environmental effects even without genotype data.
 These results are presented in **Figure 3** as follows.

This validation using genotype-available cohorts demonstrates that BIGFAM can provide
 reliable estimates in situations where genotype data is limited.

Furthermore, for X chromosome effects, where existing methods are particularly limited
 (especially with related individuals), BIGFAM offers unique advantages. Specifically, there have
 been very few methods capable of estimating X chromosome heritability from relatives without
 genotype data. Therefore, to validate the X chromosome heritability estimates of BIGFAM, we

compared BIGFAM's X chromosome heritability estimates from relatives with genotype-based
estimates from unrelated individuals in UKB. Across 21 phenotypes, we found high correlations
for both heritability estimates ($r = 0.658$, 95% CIs: [0.397, 0.987]) and DCR (dosage
compensation ratio, the male-to-female ratio of X chromosome heritability; $r = 0.637$, 95% CIs:
[0.422, 0.885]). These results, presented in **Fig.5**, suggest that BIGFAM effectively controls for
shared environmental effects in X chromosome heritability estimation.

We appreciate this comment as it allowed us to clarify the importance of validation in genotype-
available datasets while maintaining our goal of developing methods for situations where
genotype data is unavailable. While current access to EHR datasets remains limited due to
various policy, privacy concerns, and consent issues, we anticipate that as these datasets
become more accessible to researchers, BIGFAM will serve as a cost-efficient tool for
estimating heritability across diverse cohorts and populations without the need for genotype
data.

Reviewer 2

Major Comments

[Major 2.1] GCTA is a REML-based software, it has the option to estimate a relationship
matrix based on SNP-data and used this SNP-based matrix to estimate heritability, but
one can fit any matrices into GCTA . This indicates that the authors' miss understanding
of GCTA.

(Response 2.1) We sincerely thank the reviewer for this important clarification regarding REML-
based software. The reviewer is correct that REML-based methods can accommodate various
types of matrices beyond SNP-based data. As the reviewer mentioned, these methods can
utilize pedigree information to construct additive genetic matrices, and shared environmental
effects can be modeled using binary indicators (1 for related pairs, 0 otherwise).

To better understand this flexibility, we carefully examined existing implementations. For
instance, AsREML provides functionality for heritability estimation using pedigree information
through its *ainverse()* function. However, we noted potential limitations when shared
environmental effects are present. As demonstrated by Yang et al.[3], heritability estimates using
only the additive genetic matrix may be overestimated in such cases. We have included a
detailed discussion of this in the **Supplementary Text**:

*Tools such as AsREML support estimating heritability without genotype data using pedigree*
*information (through the *ainverse()* function). This function constructs the inverse of the*
*additive genetic matrix using pedigree information enabling faster REML computation. With*
*this inverse matrix, AsREML estimates heritability using REML by jointly fitting two matrices:*
*the pedigree-based additive genetic matrix (\mathbf{A}^*) and the individual-specific environmental*
*matrix (\mathbf{I})[2]. This model converges well without numerical issues, however, this approach*
*does not account for shared environmental effects. With shared environmental effect, this*
*approach can yield biased estimate. Yang et al.[3] demonstrated that heritability estimates*
*from REML with only additive genetic matrix can be overestimated under the presence of*
*shared environmental effects.*

To mitigate the overestimation due to shared environment effect, we attempted to extend this
approach by incorporating shared environmental matrices. However, we encountered some
numerical challenges with our dataset (ranging from first to third-degree relatives). Upon
investigation, we observed that the average information (AI) matrix exhibited very high condition
numbers ($10^{16} - 10^{17}$), potentially leading to numerical instability. We identified three possible
contributing factors:

- 1. **Number of Parameters to Estimate:** The model requires estimating five parameters
(genetic variance, shared environment for degrees 1-3, and individual-specific
environmental effect), increasing matrix complexity.
- 2. **Matrix Sparsity:** The pedigree-based genetic relatedness matrix \mathbf{A}^* is inherently sparse,
with only related pairs having non-zero coefficients.
- 3. **Multicollinearity:** Strong dependencies exist among relatedness matrices, as
relationship status in one matrix deterministically affects others.

To better understand these challenges, we explored two potential solutions:

- 1. **Reducing the number of parameters:** By using a single shared environmental matrix
with incorporating a shared environmental decaying factor (w_S). This modification

reduces parameters from five to three (V_a , V_s , and V_e) and improved stability with reduced
condition numbers to ~500.

2. **Mitigate matrix sparsity and multicollinearity:** We substitute the pedigree-based
genetic relationship matrix (A^*) with SNP-based additive genetic relationship matrix (A).
This substitution improved stability with reduced condition numbers to ~200.

These investigations suggest that while REML-based approaches without genotype data are
certainly possible, they may face certain numerical challenges especially when analyzing
multiple degrees of relatives. These challenges might be addressed either through incorporating
a shared environmental decay factor (w_s) or using SNP-based relationships. We note that while
w_s appears important for genotype-free heritability estimation, conventional REML frameworks
may not provide a direct method for determining its optimal value.

In light of these observations, BIGFAM might serve as a complementary approach to existing
REML-based methods, particularly for datasets with multiple degrees of relatives where
genotype data is unavailable. We have added a new section titled **Comparison of BIGFAM**
**with REML and HE-regression** in the **Results** to discuss these methodological relationships
as follows.

*Existing methods such as REML[1][2] and Haseman-Elston regression[3] can adapt to*
*situations where genotype data is unavailable by replacing genotype-based relatedness*
*measures with pedigree-based information. In these pedigree-based approaches, genetic*
*correlation is determined by the degree-of-relatedness of pair: unrelated pairs are assigned 0,*
*while d-th degree relatives are assigned 2^{-d} based on expected genetic correlation. However,*
*this approach fits phenotypes to a single genetic relatedness matrix (GRM). When applied to*
*relatives, such fitting can lead to biased estimates of V_G due to shared environmental*
*effects[3]...*

Additionally, comprehensive mathematical derivations and simulation results are thoroughly
documented in the **REML approach without genotype data** section of the **Supplementary**
**Text** as follows.

*We explored whether REML could be similarly applied using only familial relationships. To*
*this end, we constructed multiple relationship matrices based on pedigree information. First,*
*pedigree-based an additive genetic relatedness matrix (A^*) is constructed with coefficients of*
*$(0.5)^d$ for d-degree relative pairs and 0 for unrelated pairs. Next, for each degree-of-*
*relatedness, we construct a shared environmental relatedness matrix (S_d) with coefficients of 1*
*for d-degree relative pairs and 0 for unrelated pairs. With these matrices, the model can be*
*expressed as...*

We are deeply grateful for this insightful comment, which has helped us better articulate the
relationship between REML-based approaches and BIGFAM, and clarify how different
methodological approaches might complement each other in various research contexts.

**[Major 2.2] LDSC heritability is not SNP heritability. SNP heritability, also known as**
**GCTA heritability or GREML heritability, usually refers to the heritability estimated by**
**fitting a genomic relationship matrix in REML using only unrelated individuals.**
**SNP/GCTA/GREML heritability in the presence of relatives yields similar estimate as**
**Kinship/pedigree heritability. LDSC heritability is usually considered as the lower limit of**
**narrow sense heritability and usually uses GWAS summary data from unrelated**
**samples/or relatedness correct method in the presence of relatives. In general, LDSC**
**heritability < SNP/GCTA/GREML heritability unrelated samples < SNP/GCTA/GREML**
**heritability related samples < Kinship/pedigree heritability (e.g. BIGFAM.) Therefore, it is**
**not surprising that BIGFAM heritability is greater than LDSC heritability.**

**(Response 2.2)** We are deeply grateful for this important clarification regarding the fundamental
distinctions between different types of heritability estimates. The reviewer's explanation has
helped us understand that each method indeed captures different aspects of heritability:
summary-based methods (e.g., LDSC, LDpred2) provide lower bounds of narrow-sense
heritability, genotype-based methods (e.g., GCTA, RDR) estimate SNP heritability, and
pedigree-based methods (e.g., SEM, BIGFAM) estimate pedigree heritability. We sincerely
acknowledge that our previous manuscript's comparison of heritability magnitudes across
methods could have been misleading.

To address this important point, we have carefully revised the **Results** section as follows:

*To validate the variance components estimated by BIGFAM, we compared our genetic effect*
*estimates (V_G) with those from 8 other methods. We classified these methods into three distinct*
*categories based on their input data requirements. The first category, summary-based*
*methods, estimates genetic variance components (heritability) using GWAS summary statistics.*
*... To ensure fair comparison, both genotype-based and pedigree-based methods were applied*
*to the same set of first- to third-degree relative pairs (81,326 relative pairs for UKB and*
*38,006 relative pairs for GS:SFHS). ... These methods capture different aspects of heritability.*
*First, summary-based methods utilize LD scores and generally provide the lowest estimates of*
*SNP-heritability[5][6]. Second, genotype-based methods use genetic relationship matrices*
*(GRM) to estimate SNP-heritability. Third, pedigree-based methods employ pedigree-based*
*relatedness to estimate pedigree-heritability, typically providing the highest estimates among*
*these three categories.*

Given these fundamental methodological differences, we realized that comparing absolute
magnitudes would not be appropriate. Instead, we focused our analysis on correlations (R-
squared) between methods. We reasoned that strong correlations might indicate successful
partitioning of genetic and shared environmental effects, even if absolute values differ. To
examine this, we carefully analyzed heritability estimates across nine methods for 40
phenotypes that showed significantly non-zero heritability across all methods. For fair
comparison, we applied both genotype-based and pedigree-based methods to the same set of
relative pairs (81,326 pairs for UKB and 38,006 pairs for GS:SFHS).

Our analysis revealed several interesting patterns. Methods within the same category showed
high concordance, supporting the reviewer's point about distinct aspects of heritability captured
by each approach. Interestingly, BIGFAM demonstrated strong correlations with methods from
other categories (summary and genotype-based methods; r^2 ranges from 0.729 to 0.791)
compared to its correlation with SEM (pedigree-based method; $r^2 = 0.628$). This pattern was
particularly notable in comparisons with summary-based methods, where BIGFAM achieved
correlation levels similar to genotype-based methods. For instance, LDpred2's correlation with

BIGFAM ($r^2 = 0.804$, 95% CIs: [0.620, 0.891]) was notably stronger than its correlation with
 SEM ($r^2 = 0.587$, 95% CI: [0.339, 0.761]). These results are presented in **Figure 3**.

We are sincerely grateful for this insightful comment, as it has helped us better understand and
 articulate the important distinctions between different types of heritability estimates. This has led
 438 us to focus our validation on more appropriate metrics and present our results with greater
 precision.

**[Major 2.3] In discussion the authors mentioned that the model $P = A + S_1 + S_2 + S_3$ failed**
**in REML. That is likely due to having three sparse matrices (i.e. S1, S2, and S3). We**
**normally have one matrix for a specific type of common environment effects. In your**
**case where the coefficient decays with degree, you can modify corresponding entry**
**within the same matrix (e.g. 1 for 1st degree, 1/4 for 2nd degree, and 1/16 for 3rd degree**
**assuming $w_s = 4$), rather than have three matrices. This should solve the problem.**

**(Response 2.3)** We are deeply grateful for your insightful suggestion regarding the numerical
instability in REML estimation with multiple sparse matrices. Your suggestion has helped us
better understand that the instability likely stems from jointly fitting multiple sparse matrices in
our original approach.

Following your guidance, we conducted a careful investigation of different approaches. First, we
examined the $P = A + S_1 + S_2 + S_3$ model and observed very high condition numbers ($10^{16} -$
10^{17}) in the average information (AI) matrix. As the reviewer suggested, we then implemented a
single shared environmental matrix using a decay factor (w_s), constructing it with coefficients of
1 for first-degree relatives, $1/w_s$ for second-degree relatives, and $1/w_s^2$ for third-degree
relatives. This modification led to substantial improvements in numerical stability and we
described this approach in the **Supplementary Text** as follows.

*... we simplified the model by introducing a single shared environmental matrix with a shared*
*environmental decaying factor (w_s). This shared environmental matrix is constructed with*
*coefficients of 1 for first-degree relatives, $1/w_s$ for second-degree relatives, and $1/w_s^2$ for*
*third-degree relatives. ... With this modification, the numerical instability issue was resolved,*
*showing stable monotonic convergence of the likelihood function. The condition number of the*
*AI matrix was substantially reduced from the previous extremely large value ($10^{16} - 10^{17}$)*
*to ~ 500 .*

While this approach helped resolve the numerical instability, we noticed a potential challenge:
the choice of w_s appeared to significantly influence the estimated variance components, yet
existing REML-based methods may not provide a direct framework for optimizing this
parameter. For instance, in our analysis of *Urate*, we observed that the estimated variance
components (V_A, V_S) varied with different w_s values: (0.41,0.03), (0.50,0.00), and (0.46,0.02) for
w_s values of 1.1, 4.0, and 100, respectively.

To further explore possible solutions, we also examined an alternative approach by
incorporating a genotype-based relatedness matrix instead of the pedigree-based matrix (A).
This modification, similar to approaches used in methods such as REML-KIN[7], also showed
improved stability with a condition number of approximately 200 in the AI matrix. We have
detailed this analysis in the **Supplementary Text** as follows.

*To address the matrix sparsity and multicollinearity, we replaced the pedigree-based additive*
*genetic matrix (A^*) with a SNP-based additive genetic relationship matrix (A). This*
*substitution directly addresses the sparsity issue, as the SNP-based matrix contains non-zero*
*values for all pairs of individuals. ... Despite maintaining the same number of parameters to*
*estimate, this modification also resolved the numerical instability issue, showing stable*
*monotonic convergence of the likelihood function. Specifically, the condition number of the AI*
*matrix was reduced to ~ 200 , which is even lower than that of the single shared environmental*
*matrix model (~ 500).*

Through these investigations, we identified two potential approaches for achieving stable REML
estimation with multiple degrees of relatives: (1) using a single shared environmental matrix with

an appropriate decay factor (w_S), or (2) incorporating a genotype-based relatedness matrix.
While both approaches effectively address the numerical instability, the second approach
requires genotype data, which may not always be available. For genotype-free scenarios,
BIGFAM can first estimate the optimal w_S value by analyzing how shared environmental effects
decay across degrees of relatedness, and this empirically determined w_S value can then be
used to construct the shared environmental matrix in REML-based methods. Thus, BIGFAM can
serve as a complementary tool to REML-based methods in situations where genotype data is
unavailable.

We are sincerely grateful for your thoughtful suggestion, which has guided us toward a deeper
understanding of the relationship between matrix structure and numerical stability in REML
estimation.

**[Major 2.4]** In discussion the authors mentioned that “Expanding on this idea, the genetic
relatedness matrix (GRM) can be replaced by an expected genetic relatedness matrix
(eGRM) that incorporates the expected genetic correlation between individuals.”. The
‘eGRM’ the authors referred here is what we called a kinship matrix or a pedigree matrix
or an additive genetic matrix (noted as A matrix). People used kinship matrix to estimate
heritability in animal breeding long before the first GWAS study.

**(Response 2.4)** We are deeply grateful for your important clarification regarding the
terminology. Indeed, what we referred to as ‘eGRM’ is the well-established pedigree-based
additive genetic relatedness matrix, which has been foundational in animal breeding research
long before the advent of GWAS studies. We sincerely acknowledge that our imprecise
terminology could have misled readers into thinking this was a novel concept, when it has such
a rich history in the field.

To address this important point and avoid any potential confusion, we have carefully revised our
terminology throughout the manuscript to use the more precise and accurate term ‘pedigree-
based additive genetic relatedness matrix’. This revision is reflected in the **Results** section:

*Existing methods such as REML[1][2] and Haseman-Elston regression[3] can adapt to*
*situations where genotype data is unavailable by replacing genotype-based relatedness*
*measures with pedigree-based information. ... These methods effectively address the shared*
*environmental issue when using SNP-based GRM. However, when using multiple relatedness*
*matrices, substituting SNP-based GRM with pedigree-based GRM results in convergence*
*issues, a problem not observed in single-GRM implementations (Supplementary Text).*

We have also updated the terminology consistently throughout the **Supplementary Text** as
follows.

*We explored whether REML could be similarly applied using only familial relationships. To*
*this end, we constructed multiple relationship matrices based on pedigree information. First, a*
*pedigree-based additive genetic relatedness matrix (A^*) is constructed with coefficients of*
*$(0.5)^d$ for d -degree relative pairs and 0 for unrelated pairs. Next, for each degree-of-*
*relatedness, we construct a shared environmental relatedness matrix (S_d) with coefficients of 1*
*for d -degree relative pairs and 0 for unrelated pairs. With these matrices, the model can be*
*expressed as:*

These revisions aim to clearly distinguish between genotype-based relatedness matrix (A) and
pedigree-based relatedness matrix (A^*), helping readers better understand the established
methodological framework we are building upon.

We are sincerely grateful for this valuable comment, as it has helped us align our terminology
with the field’s established conventions. We have thoroughly reviewed our manuscript to ensure
consistent and accurate terminology throughout, and we would greatly appreciate any additional
guidance on further improving the clarity and precision of our terminology.

Reviewer 4

Major Comments

**[Major 4.1] Can BIGFAM still partition and estimate the X chromosome h^2 if the shared**
**environment component cannot be distinguished from the genetic component? In line**
**188, you provided the results that even when the slope test does not reject the null,**
**BIGFAM can still predict the parameters, but can it still be used for estimating the h^2 of**
**the X chromosome?**

**(Response 4.1)** We are grateful for this important question regarding BIGFAM's capability to
estimate X chromosome heritability. Yes, BIGFAM can estimate X chromosome heritability (h_X^2)
even when shared environmental and genetic components cannot be distinguished, due to its
unique estimation approach that leverages relationship-specific variations in X chromosome
inheritance patterns.

We acknowledge that our previous manuscript did not adequately explain this crucial aspect of
BIGFAM. While the method employs distinct approaches for estimating autosomal heritability
(V_G) and X chromosome variance components (V_X), their underlying principles differ
fundamentally. For autosomal heritability estimation, BIGFAM relies on distinguishing genetic
components from shared environmental components by analyzing their differential decay
patterns across degrees of relatedness. In contrast, X chromosome heritability estimation
leverages a different property: within each degree of relatedness, relationships share similar
levels of autosomal and shared environmental correlation but exhibit distinct X chromosome
genetic correlations. This key property enables X chromosome heritability estimation without
requiring separation of genetic and shared environmental components. Specifically, we stratify
relationship-specific FR-reg coefficients by degree-of-relatedness and utilize the variation in X
chromosome correlations within each degree. When we regress out the mean FR-reg coefficient
within each degree, both autosomal and shared environmental components become mean-zero
error terms. While this process removes the mean X chromosome correlation across all
relationship types within each degree, we can still recover X chromosome heritability because
different relationship types maintain their distinct, known X chromosome correlation patterns
even after mean removal. This approach is described in detail in **Methods** as follows.

*It is important to note that the variance component by autosome (V_A) is canceled out because*
*the genetic correlation on the autosome is constant within each bin (i.e., $E[r_{A,l}] = E[r_{A,d}]$ for*
*all pairs l in degree d). In contrast, the X chromosome genetic correlation ($r_{X,l}$) varies within*
*each bin depending on the specific type of relationship (e.g., mother-daughter has $r_{X,l} = 0.5$ vs*
*father-daughter has $r_{X,l} = 1/\sqrt{2}$), even among relatives of the same degree.*

To evaluate the robustness of this approach across different shared environmental patterns, we
conducted comprehensive simulations with four distinct shared environmental scenarios: (1) a
reference scenario with gradual shared environmental decay (SC1), (2) a nuclear-family-specific
scenario where shared environment only affects first-degree relatives (SC2), (3) a maternal-
effect scenario where mother-offspring pairs share stronger environmental effects (SC3), and (4)
a scenario where shared environmental effects are strongest in second-degree relatives (SC4).

These simulations demonstrated that BIGFAM maintains robust performance in most scenarios.
However, we observed a slight overestimation in the maternal-effect scenario (SC3), likely due
to the alignment of shared environmental patterns with X chromosome inheritance patterns.
This occurs because maternal relationships naturally have both higher X chromosome
correlations (as mothers pass their X chromosome to both sons and daughters) and

simultaneously tend to share stronger environmental effects with their children. When these two
patterns align, it becomes challenging to fully separate the X chromosome effects from the
enhanced maternal environmental sharing. We described on this in the **Results** section as
follows.

*To further evaluate BIGFAM under different shared environmental scenarios, we tested the*
*performance in estimating V_X on four shared environmental scenarios. These scenarios*
*included: a reference scenario with gradual environmental decay (SC1), a nuclear-family-*
*specific scenario where shared environment only affects first-degree relatives (SC2), a*
*maternal-effect scenario (SC3), and a scenario where shared environmental effects are*
*strongest in second-degree relatives (SC4). For all scenarios, we set autosome, X*
*chromosome, and shared environmental variance components as $(V_A, V_X, V_S) = (0.4, 0.02, 0.2)$*
*(see Methods). BIGFAM demonstrated robust performance across most scenarios, accurately*
*estimating the true V_X value (Fig.4a). In the maternal-effect scenario (SC3), we observed a*
*slight overestimation (0.030 with 95% CIs: [0.013, 0.045]) compared to other scenarios. This*
*overestimation likely occurs because maternal relationships tend to have both stronger shared*
*environmental effects and higher X chromosome correlation compared to other relationships.*
*For example, the X chromosome correlation of mother-daughter relationships ($\frac{1}{2}$) is higher*
*than father-daughter relationships ($\frac{1}{\sqrt{2}}$). Since our model assumes these effects are*
*independent, the correlation between them in maternal relationships can lead to slightly*
*inflated V_X estimates.*

Through this comprehensive validation, we found that BIGFAM provides reliable X chromosome
heritability estimates across most scenarios, with a specific limitation in cases where shared
environmental patterns correlate with X chromosome inheritance patterns (as in the maternal-
effect scenario). We are grateful for this question as it has helped us better articulate both the
capabilities and limitations of BIGFAM's X chromosome heritability estimation approach.

[Major 4.2] Figure 1: the caption needs a lot more details. It is actually unclear what each
 of the letter represents in panel b, and the resolution is not high enough for the labels to
 be legible.

(Response 4.2) We are grateful for your important feedback about Figure 1's clarity and
 legibility concerns. The Figure.1 is crucial as it illustrates BIGFAM's key concepts, particularly in
 panel b, which demonstrates how relationship-specific variations in X chromosome genetic
 correlations enable heritability estimation without genotype data.

To address these concerns, we have thoroughly redesigned the figure with the following
 enhancements: 1. Restructured panel b into a clear network diagram format 2. Added explicit
 labels for each node to identify family relationships 3. Significantly improved the resolution for
 better legibility 4. Enhanced the visual representation of genetic correlation strengths through
 varying line weights

Additionally, the revised caption now provides detailed explanations as follows.

*BIGFAM has two primary objectives: (a) partitioning variance components by genetic and*
 *shared environmental effects, and (b) inference of variance component by X chromosome. a.*
 *Schematic representation of the central idea for the first objective. It illustrates how genetic*
 *correlation (r_G) and shared environmental correlation (r_S) between relatives exhibit different*
 *rates of decay as degree-of-relatedness (d) increases. The genetic correlation (r_G , green*
 *dashed line) decreases with a factor of 2, while the shared environmental correlation (r_S , pink*
 *dashed line) follows a distinct decay pattern with a factor of w_S as d increases. b.*
 *Visualization of the central idea for the second objective. It highlights that genetic correlations*
 *on the X chromosome ($r_{X,l}$) between relatives differ across various familial relationships (l)*
 *even within the same degree of relatedness (first-degree in here). Each node represents a*
 *family member (e.g., father, mother, daughter, son). The thickness of each line reflects the*
 *strength of genetic correlation on the X chromosome ($r_{X,l}$), and dashed lines denote*
 *relationships with $r_{X,l} = 0$.*

We believe these comprehensive modifications will significantly improve the figure's clarity and
 help readers better understand the fundamental concepts underlying BIGFAM's methodology.
 We appreciate your careful attention to detail, which has helped us enhance the figure's clarity
 and legibility.

[Major 4.3] In line 126-127, the author state that “we can assume $r_{s,l}$ is a r.v. identically
distributed for all l with the same d .” Is there any evidence in the literature supporting
this statement? Does this assumption have any biological support or foundation?

(Response 4.3) We are grateful for this critical question regarding the biological foundation of
our model’s assumptions about shared environmental correlations. This question touches upon
a fundamental aspect of our methodology that warrants careful examination. While we assume
that shared environmental effects are identically distributed within each degree of relatedness,
the biological support or foundation of this assumption requires careful consideration.

Our comprehensive literature review revealed several key points: multiple studies have
examined shared environmental effects across relationships^{[8][9][10]}, however, these studies
focus primarily on variance components rather than correlations. This distinction is crucial
because while genetic correlations can be estimated using measurable factors like SNPs,
shared environmental correlations are more challenging to quantify due to the complex and
often unobservable nature of environmental factors. Consequently, most existing methods,
particularly REML-based approaches, simplify their models by using binary indicators (1 for
related pairs, 0 for unrelated pairs) to construct shared environmental matrices^[7].

BIGFAM takes a different approach by decomposing shared environmental effects into two
components: shared environmental correlation ($r_{s,l}$) and degree-specific variance components
(V_s). Their product ($r_{s,l} * V_s$) represents the shared environmental variance component for each
relationship type.

To rigorously evaluate the robustness of our assumption, we conducted extensive simulation
studies examining four distinct scenarios for $r_{s,l}$ distribution: (1) normal distribution (baseline
scenario), (2) left-skewed distribution (modeling cases where most relatives have strong and
some relatives have weak shared environmental effects), (3) right-skewed distribution (modeling
cases where most relatives have weak and some relatives have strong shared environmental
effects), and (4) uniform distribution (modeling equal probability of all sharing levels). The
results show that BIGFAM maintains consistent performance across all distributions, accurately
estimating the true value. While we observed slight variations in median estimates and 95%
confidence intervals across different distributions, all estimates remained close to the true value.
We presented this simulation results in **Fig.S9** as follows.

However, we identified an important limitation: when shared environmental correlation ($r_{s,l}$)
correlate with genetic inheritance patterns, estimates can become biased. For example, in the
maternal-effect scenario where mother-offspring pairs have both higher X chromosome
correlations and stronger shared environmental effects, we observed overestimation of V_x . We
detail this limitation in our **Results** section:

*To thoroughly evaluate this limitation, we tested BIGFAM across four distinct shared*
*environmental scenarios ... Our results demonstrated that BIGFAM maintains robust*
*performance in most scenarios, but showed a slight overestimation in the maternal-effect*
*scenario (SC3). This overestimation occurs due to a unique characteristic of maternal*
*relationships: they naturally have higher X chromosome correlations (as mothers pass their X*
*chromosome to both sons and daughters) and simultaneously tend to share stronger*
*environmental effects with their children. When these two patterns align, it becomes*
*challenging to fully separate the X chromosome effects from the enhanced maternal*
*environmental sharing.*

While direct biological evidence for our assumption remains limited in the literature, our
comprehensive validation across diverse scenarios demonstrates BIGFAM's robust
performance even when this assumption is violated. Nevertheless, we acknowledge specific
scenarios where bias can occur, particularly when shared environmental patterns correlate with
genetic inheritance patterns. We are grateful for this question as it has prompted us to more
thoroughly validate and clearly articulate both the capabilities and limitations of our approach.

**[Major 4.4] The first goal of BIGFAM is to infer whether the shared environmental**
**component can be distinguished from the genetic component, so the null is $w_s = w_g$. I**
**wonder why the null hypothesis is not whether or not there exists a shared**
**environmental component (ie.. null is $w_s = 1$) In fact, the later is more intuitive for me. I**
**couldn't get my head around why the decay in the shared environmetnal component by**
**default is the same as the genetic component. (unless the pairs are twins perhaps)?**

**(Response 4.4)** We are grateful for this insightful question about our choice of null hypothesis.
Your suggestion of testing for the existence of shared environmental effects raises an important
methodological consideration that deserves careful examination. While testing for the existence
of shared environmental effects is indeed an intuitive approach, our choice of null hypothesis
($w_s = w_g$) stems from BIGFAM's fundamental approach: across multiple degrees of
relatedness, genetic (V_G) and shared environmental (V_S) components from pairwise regression
(FR-reg) coefficients can be statistically partitioned.

The rationale behind our choice of null hypothesis ($w_s = w_g = 2$) lies in how BIGFAM utilizes
FR-reg coefficients from relative pairs. Under the ACE model, FR-reg coefficients contain both
genetic and shared environmental part, and we assume that each part decays differently as
degrees-of-relatedness increases. When these components decay at the same rate ($w_s = w_g =$
2), we face the most challenging scenario for component partitioning. For example, consider a
second-degree relative pair: their FR-reg coefficient would be half that of first-degree relatives,
and third-degree relatives would show one-fourth of the first-degree coefficient. In this case, we
cannot determine whether these patterns arise from genetic effects, shared environmental
effects, or a combination of both.

To illustrate this mathematically, consider a scenario where variance component by genetic
effects (V_G) is 0.4 and by shared environment (V_S) is 0.1 with shared environmental decaying
factor is identical to the genetic decaying factor ($w_s = 2$). The FR-reg coefficients would be 0.3
for first-degree relatives ($V_G/2 + V_S = 0.2 + 0.1$), 0.15 for second-degree ($V_G/4 + V_S/2 = 0.1 +$
0.05), and 0.075 for third-degree relatives ($V_G/8 + V_S/4 = 0.05 + 0.025$). However, these same
coefficients could arise from numerous other combinations, such as $V_G = 0.6$ with $V_S = 0$, or $V_G =$
0 with $V_S = 0.3$. Only when the decay rates differ can we uniquely identify these components.

We have added this explanation to the **Results** section to clarify this important point as follows:

*Based on these decompositions, we can express the expected FR-reg coefficient as follows.*

$$E[\lambda_d] = 2^{-d} \cdot V_G + w_s^{-d+1} \cdot V_S.$$

*When the decaying rate is identical in both genetic and shared environmental parts ($w_s = 2$),*
*the equation simplifies to $E[\lambda_d] = 2^{-d} \cdot (V_G + 2V_S)$. In this case, it is statistically impossible*
*to distinguish between V_G and V_S as they become a single combined term. However, if the*
*decaying rate differs ($w_s \neq 2$), we can decompose the two components using FR-reg*
*coefficient by analyzing how FR-reg coefficients change across different degrees of*
*relatedness (see Methods).*

Additionally, we explored your suggested approach of testing for the existence of shared
environmental effects by estimating separate shared environmental components for each
degree of relatedness. However, this approach led to convergence issues. These issues could
be resolved in two ways: incorporating genotype-based relatedness information or merging
shared environmental components across degrees. The first solution using genotype data
contradicts our goal of estimating variance components without genotype information. The
second approach of merging shared environmental components requires a shared

environmental decay parameter (w_S), which is what BIGFAM estimates. We describe this
additional analysis in the **Results** section as follows.

*These convergence issues arise from three key factors. First, the incorporation of additional*
*information matrices increases the number of parameters to be estimated, leading to*
*numerical instability. Second, multicollinearity issues between the pedigree-based GRM and*
*additional shared environmental matrices further complicate the estimation process.*
*Specifically, the values in these matrices are structurally dependent: when a pair shows a*
*specific degree of relationship in the GRM, it deterministically defines the values in all shared*
*environmental matrices used in the analysis. Third, the inherent sparsity of the pedigree-based*
*GRM contributes to this instability, as only a small fraction of pairs (those who are related)*
*have non-zero values among all possible pairs of individuals. These factors collectively*
*prevent the estimates from converging in both REML and HE regression implementations, as*
*detailed in the Supplementary Text.*

In conclusion, our choice of $w_S = w_G$ as the null hypothesis enables us to statistically partition V_G
and V_S without genotype data when $w_S \neq w_G$. Through this analysis and your thoughtful
suggestion, we have demonstrated that testing for shared environmental effects at each degree
of relatedness requires additional information beyond what's available in genotype-free settings.
We appreciate this comment as it helped us clarify these important aspects of our method.

[Major 4.5] For the real trait analyses, it'd be quite interesting to analyze the shared
environmental components of behavioral traits that have been established to have a
strong genetic component, such as educational attainment and physical activity.
Moreover, there has been some quite prominent studies showing that the interplay
between genetics and environment can play a major role in the inheritance of trait as well
(e.g. Kong et al. 2018 Science – “genetic nurture” effect.) Did you find any significant
variance components there?

(Response 4.5) Thank you for this important question about the interplay between genetics and
environment in trait inheritance. We agree that this interaction plays a crucial role in
understanding various variance components of traits, as demonstrated by Kong et al.'s work
showing significant nurture effects in educational attainment, nutrition, and health-related traits.

Our analysis also revealed interesting patterns of shared environmental components in nutrition
and health-related traits. For example, standing height showed the highest shared
environmental effect (0.245, 95% CIs = [0.164, 0.398]), which aligns with its known sensitivity to
nutritional status. Additionally, we found that *urate* and *whole body impedance* showed nuclear-
family-specific shared environmental effects ($w_S \approx 100$), consistent with their strong association
with dietary habits and body fat composition. Interestingly, forced vital capacity (FVC)
demonstrated similar genetic components across cohorts ($V_G \approx 0.33$) but different shared
environmental effects ($V_S = 0.096$ in GS:SFHS vs $V_S = 0.025$ in UKB), suggesting population-
specific environmental influences. We describe these findings in detail in the **Results** section:

*The results on shared environmental effects revealed several notable findings. First, we*
*identified phenotypes which have particularly high shared environmental variance component*
*(V_S). The top three phenotypes were standing height (0.245, 95% CIs = [0.164, 0.398]),*
*immature reticulocyte fraction (0.158, 95% CIs = [0.022, 0.164]), and neutrophil percentage*
*(0.156, 95% CIs = [0.040, 0.161]). Notably, these traits are known to be significantly*
*influenced by nutritional status [11][12][13], which may explain their high shared*
*environmental effect. Second, we identified phenotypes exhibiting nuclear-family-specific*
*shared environmental effects. ... we found that urate and whole body impedance showed*
*notably high shared environmental decay values ($w_S \approx 100$). This finding is particularly*
*interesting as both traits are known to be strongly influenced by dietary and lifestyle habits*
*within nuclear families [14][15]. ... Third, we observed an interesting pattern in forced vital*
*capacity (FVC), which showed significant (slow) decay of shared environmental effects in both*
*cohorts. Despite having consistent slow decay patterns of shared environmental effects across*
*both cohorts, FVC demonstrated similar genetic variance components ($V_G = 0.331$, 95% CIs =*
*[0.327, 0.334] in GS:SFHS; $V_G = 0.333$, 95% CIs = [0.332, 0.334] in UKB) but notably*
*different shared environmental effects ($V_S = 0.096$, 95% CIs = [0.0945, 0.098] in GS:SFHS;*
*$V_S = 0.025$, 95% CIs = [0.025, 0.026] in UKB). This suggests that while genetic architecture*
*might be preserved across populations, shared environmental effects can be cohort-specific,*
*even when the pattern of environmental decay remains consistent.*

However, while our analysis reveals important patterns in shared environmental effects, we
acknowledge that our current model cannot estimate the interaction between genetic and
shared environmental effects (GxE). This limitation is particularly relevant given recent studies
demonstrating the importance of gene-environment interactions in trait inheritance, such as
genetic nurture effects. To acknowledge this important aspect of trait inheritance, we have
added this limitation to our **Discussion** section:

*Importantly, our current model cannot estimate the interaction between genetic and shared*
*environmental effects. Recent studies have demonstrated that such interactions, particularly*

*through genetic nurture effects, can significantly influence trait inheritance[16]. Future*
*development of more sophisticated models could enable the estimation of these gene-*
*environment interactions, along with genetic and shared environmental correlations across*
*multiple traits. We anticipate that further insights into shared environmental factors or the*
*development of more sophisticated models could enable the estimation of genetic and shared*
*environmental correlations across multiple traits.*

In summary, our analysis has identified several phenotypes with interesting shared
environmental patterns, including traits with high shared environmental components (e.g.,
standing height), nuclear-family-specific effects (e.g., urate and whole body impedance). While
our current model cannot estimate the interaction between genetic and shared environmental
effects (GxE), these findings provide valuable insights into the role of shared environment in trait
inheritance. We appreciate this comment as it highlights both the strengths of our current
findings in identifying shared environmental components and the potential directions for future
methodological development.

**[Major 4.6] In line 397, the authors presented two traits that have large shared**
**environmental decaying values. It'd be good if the authors can follow up on this finding,**
**and provide more intuition or evidence on why or why not this result is expected. This**
**will strengthen the results section by a lot.**

**(Response 4.6)** We appreciate this important comment regarding the interpretation of traits with
large shared environmental decaying values. We agree that this concept might not be
immediately intuitive and requires careful explanation to understand its biological implications.

To provide better intuition, we first conducted simulation studies with nuclear-family-specific
shared environmental effects scenario. In this scenario, shared environmental effects only exist
within first-degree relatives, and not between more distant relatives. Our results showed that in
this scenario, w_S takes extremely large values ($w_S \approx 100$), as described in the **Results** section
as follows.

*... we simulated additional four relationship-specific shared environmental scenarios: a*
*reference scenario where shared environmental effects gradually decay with degree (SC1), a*
*nuclear-family-specific scenario where shared environment only affects first-degree relatives*
*(SC2), ... In the nuclear-family-specific scenario (SC2), the w_S showed the extremely large*
*value ($w_S \approx 100$), indicating a sharp decline in shared environmental effects between first-*
*and second-degree relatives, effectively capturing the nuclear-family-specific nature of*
*environmental sharing.*

Building on these simulation results, we examined our real data analysis specifically looking for
traits that might exhibit similar nuclear-family-specific patterns. We found that *urate* and *whole*
*body impedance* showed similarly large w_S values, consistent with our simulation results where
nuclear-family-specific shared environmental effects resulted in extremely large w_S values ($w_S \approx$
100). This similarity suggests these traits exhibit nuclear-family-specific environmental effects.
Notably, both traits are known to be strongly influenced by multiple shared environmental
factors, including dietary habits, lifestyle patterns, and body fat composition, as demonstrated by
numerous studies. The complex interplay of these various factors within nuclear families likely
explains why their shared environmental correlations are predominantly family-specific. We
expanded on this finding in the **Results** section as follows.

*... we identified phenotypes exhibiting nuclear-family-specific shared environmental effects. In*
*simulation study, we highlighted that w_S was notably large ($w_S \approx 100$) in nuclear-family-*
*specific shared environmental scenarios. Applying this insight to our real data analysis, we*

*found that urate and whole body impedance showed notably high shared environmental decay*
*values ($w_S \approx 100$). Previous studies have shown that both traits are influenced by various*
*environmental factors including dietary habits [14] and lifestyle patterns [15], and are closely*
*associated with body fat composition [17][18]. While direct evidence for nuclear-family-*
*specific environmental effects is limited, the complex nature of these multiple environmental*
*factors and their tendency to be shared within nuclear families might explain our observation*
*of nuclear-family-specific shared environmental patterns in these traits.*

To further validate these findings, we conducted additional analyses separating first-degree
relatives into parent-offspring and sibling pairs. Notably, *impedance of whole body* maintained
its large w_S values in both relationship types:

*Additionally, among phenotypes showing consistent patterns in both PO and SIB pairs,*
*impedance of whole body showed remarkably large shared environmental decaying values*
*($w_S \approx 100$) in both relationship types. This phenotype was highlighted in the previous section*
*for having notably large shared environmental decaying value. The consistently high w_S*
*values reinforce our previous observation of nuclear-family-specific shared environmental*
*effects in this phenotype.*

Through these comprehensive analyses, including both simulation studies and real data
validation across different relationship types, we demonstrate that large w_S values consistently
indicate traits with environmental effects primarily shared within nuclear families, rather than
extending to more distant relatives. We appreciate this comment as it prompted us to provide a
more comprehensive explanation of these findings.

**[Major 4.7] The section on heritability from BIGFAM and other methods, starting from line**
**404, is an important method validation section that can perhaps be brought more upfront**
**in the results section. I had the question of how does the heritability estimated from**
**BIGFAM differs from GCTA and LDSC as I was reading through the text before this**
**section.**

**(Response 4.7)** Thank you for this important suggestion regarding the placement and clarity of
our method validation section. We agree that comparing BIGFAM with other heritability
estimation methods is crucial for validating our variance component estimates, and its current
placement might underemphasize its importance. In response, we have moved this comparison
from Figure 6 to Figure 3.

To provide a clear comparison, we classified heritability estimation methods into three
categories based on their data requirements: 1. Summary-based methods (LDSC, LDpred): Use
GWAS summary statistics 2. Genotype-based methods (RDR, GCTA, HE regression): Use
individual-level genotype data 3. Pedigree-based methods (BIGFAM, SEM): Use only
phenotype and relationship information We provide detailed descriptions of each method and
their respective definitions of heritability in the **Results** section as follows:

*To validate the variance components estimated by BIGFAM, we compared our genetic effect*
*estimates (V_G) with those from 8 other methods. We classified these methods into three distinct*
*categories based on their input data requirements. The first category, summary-based*
*methods, estimates genetic variance components (heritability) using GWAS summary statistics.*
*This category includes LDSC[19] and LDpred[20]. ... These methods capture different aspects*
*of heritability. First, summary-based methods utilize LD scores and generally provide the*
*lowest estimates of SNP-heritability[5][6]. Second, genotype-based methods use genetic*
*relationship matrices (GRM) to estimate SNP-heritability. Third, pedigree-based methods*
*employ pedigree-based relatedness to estimate pedigree-heritability, typically providing the*
*highest estimates among these three categories. However, if BIGFAM effectively partitions*
*genetic effects from shared environmental effects, its V_G estimates should show high*
*correlation with other methods despite these systematic differences in magnitude. To test this*
*hypothesis, we analyzed 40 phenotypes from UK Biobank (UKB) that showed significantly*
*non-zero V_G across all methods. The correlations (r^2) between methods were estimated*
*through 1,000 bootstrap resampling.*

To validate the variance components estimated by BIGFAM, we compared our genetic effect
estimates (V_G) with those from 8 other methods. We classified these methods into three distinct
categories based on their input data requirements. The first category, summary-based methods,
estimates genetic variance components (heritability) using GWAS summary statistics. This
category includes LDSC[19] and LDpred[20]. The second category, genotype-based methods,
leverages individual-level genotype data from relatives to estimate variance components. This
category comprises Relatedness Disequilibrium Regression (RDR)[21], GCTA, and Haseman-
Elston (HE) regression. For GCTA analysis, we utilized the big-K small-K method[22] to address
potential bias from shared environmental effects in relatives[3]. This method provides two
distinct estimates (GCTA-snp and GCTA-ped) (**Supplementary Text**). For HE regression[3], we
employed regression using the square difference of the phenotypes for pairwise individuals (HE-
SD) and using the cross product of the phenotypes for pairwise individuals (HE-CP). The third
category, pedigree-based methods, estimates variance components using only phenotype and
relationship information without genotype data. This category includes BIGFAM and Structural
Equation Model (SEM)[23]. Detailed descriptions of each method are provided in the
**Supplementary Text**. These methods capture different aspects of heritability: summary-based

methods provide lower bounds, genotype-based methods capture SNP heritability, and
 pedigree-based methods yield the highest estimates.

Given these methodological differences in how each category defines and estimates heritability,
 comparing absolute magnitudes could be misleading as it might suggest that certain methods
 capture more heritability than others. However, since all methods aim to capture variance
 components by genetic effects, we focused our analysis on correlations (r^2) between methods.
 We reasoned that if BIGFAM successfully partitions genetic and shared environmental effects,
 its estimates should correlate strongly with other methods despite differences in absolute
 values. The results showed that methods within the same category demonstrated high
 concordance, supporting the notion that each category captures distinct aspects of heritability.
 Notably, BIGFAM showed stronger correlations with methods from other categories (summary
 and genotype-based methods; r^2 ranges from 0.729 to 0.791) than with SEM (pedigree-based
 method; $r^2 = 0.628$). This distinction was particularly evident in comparisons with summary-
 based methods, where BIGFAM achieved correlation levels comparable to genotype-based
 methods, while SEM showed notably lower correlations. We describe these results in detail in
 the **Results** section:

*The results showed that methods within the same category demonstrated high concordance in*
 *their estimates (Fig.3). Interestingly, BIGFAM showed stronger correlations with methods*
 *from other categories (summary and genotype-based methods; r^2 ranges from 0.729 to 0.791)*
 *than with SEM (pedigree-based method; $r^2 = 0.628$). When examining correlations with*
 *summary-based methods, BIGFAM achieved comparable levels to those observed with*
 *genotype-based methods (Fig.3). Specifically, among the pairwise comparison, LDpred2's*
 *correlation with BIGFAM was the second highest ($r^2 = 0.804$, 95% CIs: [0.620, 0.891]),*
 *while its correlation with SEM was the lowest ($r^2 = 0.587$, 95% CI: [0.339, 0.761]). These*
 *results demonstrate that BIGFAM shows higher concordance with other methods compared to*
 *existing genotype-free methods. It suggests that BIGFAM successfully partitions genetic and*
 *shared environmental effects, even without using genotype data.*

As a result, we revised Figure 3 to show pairwise comparisons of heritability estimates across
 different methods, moving it to a more prominent position earlier in the manuscript as follows.

We appreciate this suggestion as it prompted us to provide a more comprehensive validation
 framework for BIGFAM. The suggestion led us to clarify the distinct definitions of heritability
 across different methods, while showing that BIGFAM's estimates consistently correlate well

with established methods across all categories. This comprehensive validation approach has
significantly strengthened our manuscript by providing clear evidence of BIGFAM's ability to
accurately estimate variance components, even without genotype data.

**[Major 4.8] How robust are the results to different or alternative relationship definition?**

**(Response 4.8)** Thank you for raising this important question about the robustness of BIGFAM
to different relationship definitions. Since BIGFAM uses family-relationship regression (FR-reg)
coefficients at each degree level to partition variance components (V_G , V_S), understanding how
different relationship types within each degree affect our estimates is crucial for method
validation. While testing all possible relationship combinations becomes increasingly complex
for distant relatives, we focused on first-degree relationships where we can clearly distinguish
between parent-offspring (PO) and sibling (SIB) pairs. To thoroughly evaluate this, we
conducted additional analyses by stratifying first-degree relatives into these two distinct
relationship types.

Our analysis revealed consistent patterns across most phenotypes between PO and SIB pairs,
with about one-third of phenotypes (18/51) showing significantly different decay patterns
between genetic and shared environmental effects in both relationship types. We describe these
findings in detail in the **Results** section as follows:

*Among these 51 phenotypes, 35% (18/51) showed significantly different decay patterns*
*between genetic ($w_G = 2$) and shared environmental effects (w_S) in both PO and SIB pairs.*
*We classified these phenotypes into four distinct decay pattern groups based on the 95%*
*confidence intervals (CIs) of the slope estimates in the slope test. Three groups showed*
*consistent patterns between PO and SIB pairs: the slow group of 17 phenotypes (5 from*
*GS:SFHS, 12 from UKB) with upper CI limit below 1, the fast group of 1 phenotype from UKB*
*with lower CI limit above 1, and the similar group of 16 phenotypes (1 from GS:SFHS, 15*
*from UKB) with CIs containing 1. The remaining different group comprised 17 phenotypes (2*
*from GS:SFHS, 15 from UKB) showing distinct patterns between PO and SIB pairs.*

While these decay pattern analyses provide important insights into relationship-specific effects,
we further investigated the robustness of our method by directly comparing the estimated
variance components between PO and SIB pairs. Further analysis of the predicted variance
components demonstrated strong consistency across different relationship definitions. The
genetic variance component (V_G) showed particularly robust correlation between PO and SIB
estimates ($r = 0.883$, 95% CI: [0.814, 0.936]). While shared environmental components (V_S)
showed moderate correlation initially ($r = 0.440$, 95% CI: [0.159, 0.682]). These results suggest
two key insights: first, genetic effects remain relatively stable across different familial
relationships, and second, shared environmental effects, while more variable, show positive
consistency between PO and SIB pairs. We detail these findings in the **Results** section:

*Analysis of the predicted variance components revealed strong correlations between PO and*
*SIB pairs (Fig.S5). The genetic variance component (V_G) demonstrated a particularly robust*
*correlation ($r = 0.883$, 95% CI: [0.814, 0.936]), while the shared environmental variance*
*component (V_S) showed a moderate but significant positive correlation ($r = 0.440$, 95% CI:*
*[0.159, 0.682]). Notably, these correlations strengthened substantially when we restricted our*
*analysis to phenotypes exhibiting consistent decay patterns between PO and SIB pairs (slow,*
*fast, and similar, excluding different). In these cases, the V_S correlation increased markedly*
*from $r = 0.440$ to $r = 0.713$ (95% CI: [0.402, 0.911]), while the V_G correlation showed a*
*modest improvement from $r = 0.883$ to $r = 0.916$ (95% CI: [0.839, 0.966]). These findings*
*suggest two key insights: first, genetic effects remain relatively stable across different familial*
*relationships, and second, shared environmental effects, while more variable, show strong*
*consistency between PO and SIB pairs when their decay patterns align.*

Through these comprehensive analyses of different relationship types, we demonstrate that
BIGFAM's estimates are robust across different relationship definitions, particularly for genetic
components. While shared environmental effects show more variability, they maintain strong
consistency when decay patterns align between relationship types. This validation supports the
reliability of BIGFAM's variance component estimation across different familial relationship
definitions.

**[Major 4.9] The last bit of the results can be strengthened a lot. In particular, it would be**
**really helpful to include an application of BIGFAM on EHR data, for shared environmental**
**component analyses or X chromosome analyses. Currently, being genotype-free is the**
**most important highlight or benefit of this method, but the evidence that can directly**
**support this statement is lacking. Any results that are interpretable and can demonstrate**
**the utility of BIGFAM in a real trait analyses would be helpful.**

**(Response 4.9)** We appreciate your suggestion about demonstrating BIGFAM's utility with EHR
data, as this would indeed highlight the method's key advantage of being genotype-free. Recent
paper^[4] have demonstrated the possibility of estimating heritability from large EHR data without
genotype information. While we initially attempted to use phenotype-only EHR data from various
hospitals in Korea, access proved challenging due to policy, privacy concerns and consent
issues.

However, using large datasets with genotype information (UKB, GS:SFHS) is necessary for
validating BIGFAM against existing methods. We reasoned that if BIGFAM effectively controls
for shared environmental effects in relatives while estimating genetic components, its results
should correlate well with estimates from unrelated individuals. To test this, we compared
BIGFAM's estimates with those from three different methodological categories: summary-based
methods, genotype-based methods, and other pedigree-based methods. While these
approaches estimate slightly different aspects of heritability, high correlation between them
would suggest successful partitioning of genetic and environmental components. The results,
described in **Results** section:

*The results showed that methods within the same category demonstrated high concordance in*
*their estimates (Fig.3). Interestingly, BIGFAM showed stronger correlations with methods*
*from other categories (summary and genotype-based methods; r^2 ranges from 0.729 to 0.791)*
*than with SEM (pedigree-based method; $r^2 = 0.628$). When examining correlations with*
*summary-based methods, BIGFAM achieved comparable levels to those observed with*
*genotype-based methods (Fig.3). Specifically, among the pairwise comparison, LDpred2's*
*correlation with BIGFAM was the second highest ($r^2 = 0.804$, 95% CIs: [0.620, 0.891]),*
*while its correlation with SEM was the lowest ($r^2 = 0.587$, 95% CI: [0.339, 0.761]). These*
*results demonstrate that BIGFAM shows higher concordance with other methods compared to*
*existing genotype-free methods. It suggests that BIGFAM successfully partitions genetic and*
*shared environmental effects, even without using genotype data.*

Furthermore, for X chromosome effects, where existing methods are particularly limited
(especially with related individuals and without genotype data), BIGFAM offers unique
advantages. While BIGFAM enables X chromosome heritability estimation without genotype
data, it remains crucial to validate these estimates against established methods. To this end, we
compared BIGFAM's X chromosome heritability with genotype-based estimates from unrelated
individuals in UKB. Across 21 phenotypes, we found high correlations for both heritability
estimates ($r = 0.658$, 95% CIs: [0.397, 0.987]) and DCR (dosage compensation ratio, the male-
to-female ratio of X chromosome heritability; $r = 0.637$, 95% CIs: [0.422, 0.885]). These results,
presented in **Fig.5**, suggest that BIGFAM effectively controls for shared environmental effects in
X chromosome heritability estimation.

Additionally, while GS:SFHS lacks X chromosome genotype data, BIGFAM successfully
 estimated X chromosome effects in this cohort. Despite limited phenotype overlap between the
 two cohorts, cholesterol showed significant V_X in both datasets with remarkably consistent
 estimates: 0.005 (95% CIs: [0.001,0.009]) in UKB and 0.006 (95% CIs: [0.002,0.010]) in
 GS:SFHS. These results are presented in **Fig.4**:

These results validate that BIGFAM can effectively estimate variance components even without
 genotype data. While current access to EHR datasets remains limited due to various policy,
 privacy concerns, and consent issues, we anticipate that as these datasets become more
 accessible to researchers, BIGFAM will serve as a cost-efficient tool for estimating heritability
 across diverse cohorts and populations without the need for genotype data.

[Major 4.10] Line 884: Are the results robust to these different inference criteria for the
degree of relatedness?

(Response 4.10) Thank you for raising this important question about the robustness of our
results to different inference criteria for the degree of relatedness. Unlike GS:SFHS, which
provides accurate familial relationship information through pedigree data, UKB only provides
kinship coefficients between relatives, requiring us to infer the degree of relatedness.

To address this, we carefully examined the distribution of kinship coefficients in the UKB
dataset. We observed distinct peaks around the expected kinship values for each degree of
relatedness (0.5 for first-degree, 0.25 for second-degree, and 0.125 for third-degree relatives).
Based on these distributions, we established appropriate intervals for relationship inference:
first-degree (0.4, 0.6), second-degree (0.2, 0.3), and third-degree (0.10, 0.15) relatives. We
have added this distribution and our cutoff criteria to **Fig.S13**:

Notably, while first and second-degree relatives show clear peaks at their expected kinship
values, the peak for third-degree relatives is slightly shifted from the expected kinship value of
0.125. This shift likely reflects the greater complexity of distant relationships, where some
fourth-degree relatives with higher-than-expected kinship coefficients might be included. We
acknowledge this limitation in our **Discussion** section:

*BIGFAM, while innovative, has several limitations and raises questions that require further*
*research. First, relationship information inferred from UKB can be inaccurate. Despite using*
*marker-based kinship coefficients, age, sex, and genetic correlation on the X chromosome,*
*discrepancies between inferred and true relationships can arise.*

We are grateful for this comment as it prompted us to thoroughly evaluate and document our
relationship inference criteria and their potential impact on our results, particularly for more
distant relationships.

[Major 4.11] Figure 2: What does the strong linear relationship between the slope and the
intercept indicate? It's not clear to me why this is expected to be the case.

(Response 4.11) Thank you for this insightful question about the apparent linear relationship in
Figure 2. Your observation helped us identify a potential source of misinterpretation in our
previous simulation design and led us to improve both our analysis and presentation.

In our previous manuscript, Figure 2 showed slope test results from simulations where we
varied w_S while keeping V_G and V_S fixed. We found that in this setting, the slope test results
could misleadingly suggest a linear relationship between slope and intercept, while
demonstrating how slope test results vary with w_S under fixed variance components,

To address this limitation and provide a more comprehensive evaluation of our slope test, we
conducted new simulations with randomly sampled parameters across all components (V_G , V_S ,
and w_S). We describe this improved simulation approach in the **Results** section as follows.

*To evaluate how effectively our slope test can identify different patterns of environmental*
*decay, we simulated four scenarios where shared environmental effects exhibit distinct decay*
*patterns: fast decay ($w_S > 2$), similar decay ($w_S \approx 2$), slow decay ($1 < w_S < 2$), and increase*
*($w_S < 1$). For each scenario, we generated 1,000 sets of simulated FR-reg coefficients (see*
*Methods) by randomly sampling genetic variance components (V_G) from 0.1 to 0.8 and shared*
*environmental variance components (V_S) from 0 to 0.2.*

Additionally, we expanded our simulation scenarios to include cases where shared
environmental effects increase with degree of relatedness (*increase* decay pattern, $w_S < 1$).
This new scenario revealed distinct patterns in the slope test results, with negative slopes
clearly differentiating it from other decay patterns. We present these comprehensive results in
the revised **Fig.2a**:

We appreciate this comment as it led us to improve our simulation framework and presentation.
The revised analysis now more clearly demonstrates the effectiveness of our slope test in
capturing various shared environmental patterns while avoiding potential misinterpretations from
the previous visualization.

**[Major 4.12] Figure 3: Notably, the strong linear relationship between the slope and the**
**intercept in simulations is not visible in real trait analyses. What's the intuition behind**
**this?**

**(Response 4.12)** Thank you for the question about the linear relationship difference between
simulation results and real trait analyses. As you mentioned, the linear relationship in simulation
results is not visible in real trait analyses. This difference arises because our previous simulation
setting was conducted under fixed (V_G , V_S) and only varying w_S , while real traits have unique
combinations of (V_G , V_S , w_S).

In our previous simulation with fixed V_G and V_S , varying only w_S with limited parameter
combinations created an artificial linear pattern that induces an unintended linear relationship.
To prevent this misinterpretation and remove the artificial linear pattern, we revised simulation
approach: randomly samples all parameters (V_G , V_S , w_S), better reflects the natural variation
observed in real traits. This comprehensive sampling strategy not only eliminates artificial
patterns but also validates our method's ability to detect true environmental decay patterns
across the diverse parameter combinations we observe in nature.

*To evaluate how effectively our slope test can identify different patterns of environmental*
*decay, we simulated four scenarios where shared environmental effects exhibit distinct decay*
*patterns: fast decay ($w_S > 2$), similar decay ($w_S \approx 2$), slow decay ($1 < w_S < 2$), and increase*
*($w_S < 1$). For each scenario, we generated 1,000 sets of simulated FR-reg coefficients (see*
*Methods) by randomly sampling genetic variance components (V_G) from 0.1 to 0.8 and shared*
*environmental variance components (V_S) from 0 to 0.2.*

These simulation results help explain why real traits do not show a linear relationship in the
slope test - each trait has its own unique combination of (V_G , V_S , w_S), leading to diverse patterns
rather than a simple linear relationship.

We appreciate this comment as it helped us recognize that our previous simulation visualization
could be misleading. We have revised our simulation plots to better reflect how slope test
results can vary widely depending on the underlying parameters of each phenotype, avoiding
any unintended implications of inherent linear relationships.

**[Major 4.13] Figure 6: For method comparison, direct comparison of the estimates**
**between/across methods is preferred, since it's more straightforward. Two set of results**
**can be highly correlated but have very low concordance (e.g. one is always twice as big**
**as the other).**

**(Response 4.13)** Thank you for this valuable suggestion about the importance of direct
comparisons between methods. We agree that high correlation does not necessarily imply high
concordance, and direct comparisons between methods can provide more straightforward
insights into their relationships.

To address this, we first classified the heritability estimation methods into three distinct
categories based on their data requirements: 1. Summary-based methods (LDSC, LDpred): Use
GWAS summary statistics 2. Genotype-based methods (RDR, GCTA, HE regression): Use
individual-level genotype data 3. Pedigree-based methods (BIGFAM, SEM): Use only
phenotype and relationship information We provide detailed descriptions of each method and
their respective definitions of heritability in the **Results** section as follows:

*To validate the variance components estimated by BIGFAM, we compared our genetic effect*
*estimates (V_G) with those from 8 other methods. We classified these methods into three distinct*
*categories based on their input data requirements. The first category, summary-based*
*methods, estimates genetic variance components (heritability) using GWAS summary statistics.*
*This category includes LDSC[19] and LDpred[20]. The second category, genotype-based*
*methods, leverages individual-level genotype data from relatives to estimate variance*
*components. This category comprises Relatedness Disequilibrium Regression (RDR)[21],*
*GCTA, and Haseman-Elston (HE) regression. For GCTA analysis, we utilized the big-K small-*
*K method[22] to address potential bias from shared environmental effects in relatives[3]. This*
*method provides two distinct estimates (GCTA-snp and GCTA-ped) (Supplementary Text). For*
*HE regression[3], we employed regression using the square difference of the phenotypes for*
*pairwise individuals (HE-SD) and using the cross product of the phenotypes for pairwise*
*individuals (HE-CP). The third category, pedigree-based methods, estimates variance*
*components using only phenotype and relationship information without genotype data. This*
*category includes BIGFAM and Structural Equation Model (SEM)[23].*

Given these methodological differences in how each category defines and estimates heritability,
comparing absolute magnitudes could be misleading as it might suggest that certain methods
capture more heritability than others. However, since all methods aim to capture variance
components by genetic effects, we focused our analysis on correlations (r^2) between methods.
We reasoned that if BIGFAM successfully partitions genetic and shared environmental effects,
its estimates should correlate strongly with other methods despite differences in absolute
values.

To examine both correlation and concordance, we created pairwise comparison plots between
all methods. The results showed that methods within the same category demonstrated high
concordance in their estimates. These results are presented in **Fig.S6** as follows.

Notably, BIGFAM showed stronger correlations with methods from other categories (summary
 and genotype-based methods; r^2 ranges from 0.729 to 0.791) than with SEM (pedigree-based
 method; $r^2 = 0.628$). This distinction was particularly evident in comparisons with summary-
 based methods, where BIGFAM achieved correlation levels comparable to genotype-based
 methods, while SEM showed notably lower correlations. These results are presented in **Fig.3** as
 follows.

We appreciate this suggestion as it led us to provide a more comprehensive comparison of
 methods through both direct comparisons and correlation analyses, helping to demonstrate
 BIGFAM's effectiveness in partitioning genetic and shared environmental effects even without
 genotype data.

**Minor Comments**

**[Minor 4.1] The equation in line 720: I found this way of presenting the model a bit**
**confusing, though I understand it might come from the previous reviewer’s comment that**
**the choice of X and Y may matter. Is the current model fitting every pair of related**
**individuals twice (or as two data points)? More explanations or clarifications might be**
**necessary.**

**(Response)** Thank you for this important question about the model presentation. The equation
represents our systematic approach to handling the directionality in relative pair relationships,
where the choice of predictor and response variables could potentially affect the results.

To clarify the treatment of predictor-response directions, we have explicitly stated that the
regression analysis includes “all possible predictor-response combinations within each
relationship type” to emphasize that both directions (e.g., parent-to-offspring and offspring-to-
parent) are systematically considered. This approach ensures that the analysis captures the
variance components from both perspectives, rather than treating one direction as canonical.

To ensure better understanding, we provided an example with parent-offspring pairs: both
parent-to-offspring and offspring-to-parent directions are “simultaneously” considered in
estimating λ_{rel} . This explicitly shows how the model incorporates both directions to fully capture
shared variance components, addressing potential concerns about treating relationships as
separate data points. In the revised manuscript, we present a refined equation and explanation
on **Methods** section.

*To address this uncertainty and better capture the variance components between relatives, we*
*include all possible predictor-response combinations within each relationship type in our*
*regression analysis. Specifically, for a given relationship type rel (which can be either a*
*degree-of-relatedness (d) or a specific familial relationship (l)), we model:*

$$\begin{bmatrix} y_{m_1} \\ y_{m_2} \end{bmatrix} = \beta_0 + \lambda_{rel} \cdot \begin{bmatrix} y_{m_2} \\ y_{m_1} \end{bmatrix}$$

*where y_{m_1} and y_{m_2} represent the standardized phenotypes of relative pair (m_1, m_2) and β_0 is*
*the intercept. For instance, in parent-offspring pairs, both parent-to-offspring and offspring-*
*to-parent directions are simultaneously considered in estimating λ_{rel} , ensuring our estimates*
*capture the full extent of shared variance components between relatives.*

These changes aim to make it clear that the model systematically integrates both directions for
each pair without unintended duplication or redundancy.

**[Minor 4.2] The equation in line 760: the second term is just a constant, so not knowing it**
**does not affect the test on the slope?**

**(Response)**

Thank you for this insightful question about the role of the intercept term in our slope test. While
you are correct that the intercept term ($V_G + 2V_S$) is a constant combination of variance
components, our slope test specifically focuses on examining whether the slope of the slope
test is greater or less than 1, rather than determining the exact composition of the intercept.

The theoretical basis for this approach lies in the properties of our model. The closed-form
equation with intercept ($V_G + 2V_S$) is only valid when the shared environmental decaying factor
equals the genetic decaying factor ($w_S = w_G = 2$). When $w_S \neq w_G$, this relationship no longer

holds, and both the slope and intercept take different forms. Through our simulations, we
demonstrated that the slope value systematically varies with w_S : when $w_S < w_G$, the slope is
less than 1, and when $w_S > w_G$, the slope is greater than 1.

This relationship between w_S and the slope value is what we leverage in our method pipeline.
The slope test serves to identify the pattern of shared environmental decay, while the actual
estimation of variance components (V_G and V_S) occurs in the subsequent prediction step, where
we utilize the identified decay pattern. Therefore, while the intercept term does contain
information about the variance components, its specific value does not affect our ability to
determine the shared environmental decay pattern through the slope test.

We appreciate this question as it helped us clarify the distinct roles of the slope and intercept in
our method, and how they contribute to the overall analysis pipeline.

**[Minor 4.3] In line 808, maybe don't say "common variance components", as it's a bit**
**unclear what "common" refers to.**

**(Response)**

Thank you for this helpful suggestion about terminology. We agree that the phrase "common
variance components" could be unclear and potentially misleading, especially as it might be
confused with common (shared) environmental variance.

To address this, we have revised the text to be more precise. The sentence: *In this step, we*
*simplify the model parameters for subsequent analyses by regressing out common variance*
*components from FR-reg coefficients* has been changed to: *In this step, we simplify the model*
*parameters for subsequent analyses by removing the mean effects within each degree-of-*
*relatedness from FR-reg coefficients.*

This revised wording more accurately describes the specific operation being performed in this
step.

**[Minor 4.4] Line 820-821: I again have a question about this assumption. Why is the**
**vairance by autosome canceled out but not the X chromosome?**

**(Response)** Thank you for this important question about the differential treatment of autosomal
and X-chromosomal variance components. The key distinction lies in how genetic correlations
are defined for autosomes versus the X chromosome across different familial relationships.

For autosomal genetic correlations ($r_{A,l}$), the values are constant within each degree of
relatedness (d). For example, all first-degree relationships (parent-child and siblings) share
$r_{A,l} = 0.5$. Therefore, when we subtract the mean effect within each degree ($E[\lambda_d]$) from
individual relationship coefficients ($E[\hat{\lambda}_l]$), the autosomal variance component (V_A) cancels out.

In contrast, X-chromosomal genetic correlations ($r_{X,l}$) vary across relationships within the same
degree due to sex-specific inheritance patterns. For example, mother-daughter: $r_{X,l} = 0.5$ and
father-son: $r_{X,l} = 0$.

Consequently, when we subtract the mean effect within each degree ($E[\lambda_d]$) from individual
relationship coefficients ($\hat{\lambda}_l$), the X-chromosomal variance component (V_X) remains in the
residual, while the autosomal variance component (V_A) cancels out.

This relationship-specific variation in X-chromosomal correlations allows us to estimate V_X
through residual analysis, while the constant autosomal correlations within each degree lead to
the cancellation of V_A .

**[Minor 4.5] Equations in line 818 and other similar equation forms: why is the V_x in the**
**last line squared?**

**(Response)** Thank you for catching this error. The squared term (V_X^2) in the equation was a
typo. We have corrected the equation from $t_l \cdot V_X^2 + \epsilon_S$ to $t_l \cdot V_X + \epsilon_S$. We appreciate your careful
review which helped us identify and fix this mistake.

**[Minor 4.6] The w in the second term of equation in 867 should have a subscript of s**

**(Response)** Thank you for catching this notational oversight. We have corrected the equation
by adding the missing subscript to w . The equation has been revised from:

$$\hat{\lambda}_d = 2^{-d} \cdot V_G + w^{-d+1} \cdot V_S + \epsilon$$

to:

$$\hat{\lambda}_d = 2^{-d} \cdot V_G + w_S^{-d+1} \cdot V_S + \epsilon$$

We appreciate your attention to detail in helping us maintain consistent notation throughout the
manuscript.

**[Minor 4.7] In figure 3: one of the box is white, but it may not have been intended to be**
**this way.**

**(Response)**

Thank you for noticing this visual inconsistency in Figure 3. The white box was indeed
unintentional and resulted from a figure processing issue. We have corrected this technical error
and resubmitted the figure with the proper coloring scheme.

**Reference**

- 1. Yang, J., Lee, S. H., Goddard, M. E. & Visscher, P. M. GCTA: A Tool for Genome-wide
Complex Trait Analysis. *The American Journal of Human Genetics* **88**, 76–82 (2011).
- 2. Butler, D. G., Cullis, B. R., Gilmour, A. R., Gogel, B. J. & Thompson, R. ASReml
estimates variance components under a general linear.
- 3. Yang, J., Zeng, J., Goddard, M. E., Wray, N. R. & Visscher, P. M. Concepts, estimation
and interpretation of SNP-based heritability. *Nat Genet* **49**, 1304–1310 (2017).
- 4. Polubriaginof, F. C. G. *et al.* Disease Heritability Inferred from Familial Relationships
Reported in Medical Records. *Cell* **173**, 1692–1704.e11 (2018).
- 5. Srivastava, A. K., Williams, S. M. & Zhang, G. Heritability Estimation Approaches
Utilizing Genome-Wide Data. *Current Protocols* **3**, e734 (2023).
- 6. Barry, C.-J. S. *et al.* How to estimate heritability: a guide for genetic epidemiologists.
*International Journal of Epidemiology* **52**, 624–632 (2023).
- 7. Hill, W. D. *et al.* Genomic analysis of family data reveals additional genetic effects on
intelligence and personality. *Mol Psychiatry* **23**, 2347–2362 (2018).
- 8. Matteson, L. K., McGue, M. & Iacono, W. G. Shared Environmental Influences on
Personality: A Combined Twin and Adoption Approach. *Behav Genet* **43**, 491–504 (2013).
- 9. Fernández-Rhodes, L. *et al.* Characterization of the contribution of shared environmental
and genetic factors to metabolic syndrome methylation heritability and familial correlations. *BMC*
*Genetics* **19**, 69 (2018).
- 10. Cleveland, H. H., Jacobson, K. C., Lipinski, J. J. & Rowe, D. C. Genetic and shared
environmental contributions to the relationship between the HOME environment and child and
adolescent achievement. *Intelligence* **28**, 69–86 (2000).
- 11. Inzaghi, E., Pampanini, V., Deodati, A. & Cianfarani, S. The Effects of Nutrition on Linear
Growth. *Nutrients* **14**, 1752 (2022).
- 12. Torino, A. B. B., Gilberti, M. de F. P., da Costa, E., de Lima, G. A. F. & Grotto, H. Z. W.
Evaluation of red cell and reticulocyte parameters as indicative of iron deficiency in patients with
anemia of chronic disease. *Rev Bras Hematol Hemoter* **36**, 424–429 (2014).
- 13. Kaya, T. *et al.* Association between neutrophil-to-lymphocyte ratio and nutritional status
in geriatric patients. *J Clin Lab Anal* **33**, e22636 (2018).
- 14. Major, T. J., Topless, R. K., Dalbeth, N. & Merriman, T. R. Evaluation of the diet wide
contribution to serum urate levels: meta-analysis of population based cohorts. *BMJ* **363**, k3951
(2018).
- 15. Brunani, A. *et al.* Body composition assessment using bioelectrical impedance analysis
(BIA) in a wide cohort of patients affected with mild to severe obesity. *Clinical Nutrition* **40**,
3973–3981 (2021).
- 16. Kong, A. *et al.* The nature of nurture: Effects of parental genotypes. *Science* **359**, 424–
428 (2018).

- 17. Kyle, U. G. *et al.* Bioelectrical impedance analysis—part I: review of principles and
methods. *Clinical Nutrition* **23**, 1226–1243 (2004).
- 18. de Oliveira, E. P., Moreto, F., Silveira, L. V. de A. & Burini, R. C. Dietary, anthropometric,
and biochemical determinants of uric acid in free-living adults. *Nutr J* **12**, 11 (2013).
- 19. Bulik-Sullivan, B. K. *et al.* LD Score regression distinguishes confounding from
polygenicity in genome-wide association studies. *Nat Genet* **47**, 291–295 (2015).
- 20. Privé, F., Albiñana, C., Arbel, J., Pasaniuc, B. & Vilhjálmsson, B. J. Inferring disease
architecture and predictive ability with LDpred2-auto. *The American Journal of Human Genetics*
**110**, 2042–2055 (2023).
- 21. Young, A. I. *et al.* Relatedness disequilibrium regression estimates heritability without
environmental bias. *Nat Genet* **50**, 1304–1310 (2018).
- 22. Zaitlen, N. *et al.* Using Extended Genealogy to Estimate Components of Heritability for
23 Quantitative and Dichotomous Traits. *PLOS Genetics* **9**, e1003520 (2013).
- 23. Muñoz, M. *et al.* Evaluating the contribution of genetics and familial shared environment
to common disease using the UK Biobank. *Nat Genet* **48**, 980–983 (2016).

RESPONSE TO REVIEWERS' COMMENTS

We are deeply grateful for the thorough and insightful comments provided by the anonymous reviewers. Their feedback has been instrumental in improving both the technical accuracy and clarity of our manuscript. In response to their valuable insights, we have carefully addressed each comment and made several key revisions to enhance the clarity, accuracy, and robustness of our study:

1. **Comparison with Existing Studies:** We compared our findings with results from other studies to demonstrate the consistency of the variance components estimated by BIGFAM. This comparison validated the assumption regarding the decay of shared environmental effects with increasing degrees of relationship.
2. **Clarification of Heritability Estimates:** We clarified the differences between SNP-based heritability and family-based heritability estimates. This includes emphasizing that SNP-based heritability is a lower bound of narrow-sense heritability due to ungenotyped markers and imperfect linkage disequilibrium.
3. **Type 1 Error and Power Analysis of Slope Test:** We performed simulations to evaluate the type 1 error and power of the slope test used in our analysis. These simulations confirmed that the test is well-calibrated and has sufficient power under various conditions.

These revisions have strengthened our manuscript by addressing the reviewers' concerns and providing a clearer and more comprehensive understanding of our methods and findings. We are grateful for the reviewers' valuable feedback, which has significantly contributed to the improvement of our work.

Reviewer 5

Major Comments

[Major 5.1] I agree with Review 1 that the decay of the shared environment with increasing degree of relationship is a strong assumption. Can the authors validate this assumption by comparing the estimated shared environmental effects of the same trait from different studies and show the consistency?

(Response 5.1) Thank you for raising this important concern regarding the decay of shared environmental effects with increasing degree of relationship. We acknowledge that this is indeed a significant assumption in our model, as also highlighted by Reviewer 1. If shared environmental effects were to increase rather than decrease with increasing degree of relationship, our BIGFAM method could potentially produce inappropriate estimates of variance components. To address this concern, we have conducted additional analyses to validate this assumption by comparing our estimated shared environmental effects with findings from other studies.

First, we confirmed that the shared environmental variance component for height estimated by BIGFAM (0.245, 95% CIs = [0.164, 0.398]) closely aligns with values previously reported in twin studies (0.2–0.4). With this research, we revised our manuscript as follows:

... we identified phenotypes which have particularly high shared environmental variance component (V_S). The top three phenotypes were standing height (0.245, 95% CIs = [0.164, 0.398]), immature reticulocyte fraction (0.158, 95% CIs = [0.022, 0.164]), and neutrophil percentage (0.156, 95% CIs = [0.040, 0.161]). ... In particular, the shared environmental effect for standing height (0.245) is consistent with previous twin studies, which report estimates around 0.3 and show age-dependent variation^{1,2} ranging from 0.2 to 0.4.

Second, regarding forced vital capacity (FVC), we estimated variance components using BIGFAM and found that while genetic variance components (V_G) showed no significant differences between cohorts, shared environmental effects (V_S) did vary. This pattern is consistent with findings from twin studies on conduct disorder (CD), where V_G remained stable across cohorts, but V_S varied. We have added the following to our revised manuscript:

Third, we observed an interesting contrast in forced vital capacity (FVC) between cohorts: while genetic variance components remained consistent, shared environmental effects differed significantly. FVC showed significant (slow) decay of shared environmental effects in both cohorts, with remarkably similar genetic variance components ($V_G = 0.331$, 95% CIs = [0.327, 0.334] in GS:SFHS; $V_G = 0.333$, 95% CIs = [0.332, 0.334] in UKB), which align well with twin studies reporting heritability of 0.26 (0.03–0.49)³. However, despite this genetic consistency, shared environmental effects varied substantially between cohorts ($V_S = 0.096$, 95% CIs = [0.0945, 0.098] in GS:SFHS vs. $V_S = 0.025$, 95% CIs = [0.025, 0.026] in UKB). This differential pattern has been observed in other traits as well, such as conduct disorder (CD), where studies have shown that while V_G remains relatively stable across populations, V_S tends to increase in recent cohorts⁴. These findings suggest that while genetic architecture might be preserved across populations, shared environmental effects can be cohort-specific, potentially due to demographic differences even when the pattern of environmental decay remains consistent.

We are grateful for your insightful comment, which has prompted us to validate the assumption of shared environmental decay by comparing our findings with existing studies. This detailed examination has improved the clarity and robustness of our model. If there are any further

suggestions or inquiries, we are eager to address them. Thank you once again for your valuable insights.

[Major 5.2] SNP-based heritability is not directly comparable to family-based heritability estimates, as the former is a lower bound of the narrow-sense heritability due to ungenotyped markers and imperfect LD. The developed method and SNP-based method aim at estimating different quantities. This should be clarified.

(Response 5.2) Thank you for your insightful comments regarding the comparison between SNP-based heritability and family-based heritability estimates. Because SNP-based heritability and pedigree-based heritability estimate different values, comparing these estimates without clear differentiation can cause misunderstandings. We have carefully considered your feedback and made the following revisions to clarify these differences in our manuscript:

1. **Clarification of SNP-heritability:** We have revised a section to explicitly state that SNP-based heritability is often considered a lower bound of narrow-sense heritability due to the presence of ungenotyped markers and imperfect linkage disequilibrium (LD).
2. **Different Quantities Estimated:** We have emphasized that SNP-based methods and family-based methods, such as BIGFAM, aim to estimate different quantities. While SNP-based methods focus on the heritability captured by genotyped variants, family-based methods can capture broader genetic influences, including those not tagged by SNPs.

To minimize misunderstandings, we have clarified these points and subsequently calculated the correlation between BIGFAM and other methods. Incorporating these points, we have revised our manuscript as follows:

It is important to note that SNP-based heritability is not directly comparable to family-based heritability estimates. SNP-based methods focus on the heritability captured by genotyped variants, which is considered a lower bound of narrow-sense heritability due to ungenotyped markers and imperfect linkage disequilibrium (LD)^{5,6}. In contrast, family-based methods, like BIGFAM, can capture broader genetic influences, including those not tagged by SNPs. Therefore, while comparing these methods, it is crucial to understand that they provide insights into different aspects of genetic variance. However, if BIGFAM effectively partitions genetic effects from shared environmental effects, its V_G estimates should show high correlation with other methods despite these systematic differences in magnitude...

These revisions aim to provide a clearer understanding of the scope and limitations of each method, as well as the rationale behind our comparative analysis. We hope these changes address your concerns and enhance the clarity of our manuscript. Thank you once again for your valuable feedback.

[Major 5.3] The identification of shared environmental variance components depends on the power of slope test in step 1. Is the slope test in step 1 calibrated? Can the authors show the type-I error and power for step 1 under different assumptions of shared environmental effects?

(Response 5.3) Thank you for highlighting the importance of the calibration and power of the slope test, which is crucial for accurately identifying shared environmental variance components. If the slope test is not well-calibrated or lacks sufficient power, our predictions could indeed be unreliable. To address these concerns, we conducted additional simulations to evaluate both the type 1 error rate and power of the slope test under various conditions:

1. **Type 1 Error Evaluation:** We assessed whether the slope test is well-calibrated under the null hypothesis (where shared environmental effects decay at the same rate as genetic effects, i.e., $w_S = 2$). We performed 1,000 simulations across a range of error values (ϵ from 0.001 to 0.01). For these simulations, we generated FR-reg coefficients for different degrees of relatedness ($d \in [1,2,3]$) using the following model (**Methods** in revised manuscript):

we generate the FR-reg coefficients for d -degree relatives ($\hat{\lambda}_d$) based on three true parameters (V_G , V_S , and w_S).

$$\hat{\lambda}_d = 2^{-d} \cdot V_G + w_S^{-d+1} \cdot V_S + \epsilon$$

where V_G is the variance component by genetic effects, V_S is the variance component by share environmental effects within first-degree relatives, w_S is the shared environmental decaying factors, and ϵ is the error term following mean zero normal distribution.

For each simulation, V_G was randomly selected from 0.05 to 0.95, and V_S from 0.01 to 0.55 with w_S fixed at 2 for null hypothesis testing.

Our results indicate that the slope test maintains a type 1 error rate of approximately 5.9% across the simulated error values, which is slightly higher than but reasonably close to the expected level of 5%. This suggests that the test is relatively well-calibrated under the null hypothesis. We presented this type 1 error evaluation in **Supplementary Figure.1a**.

2. **Power Analysis:** We evaluated the power of the slope test by simulating scenarios where shared environmental effects decay at rates different from genetic effects ($w_S \neq 2$). Specifically, we tested 20 slow decaying scenarios ($w_S \in (1,2)$) and 30 fast decaying

scenarios ($w_S \in (2,8)$). For each scenario, we conducted 1,000 resampling iterations at three different error levels ($\epsilon = [0.005, 0.01, 0.05]$) and assessed how frequently the slope test correctly identified the true decay pattern.

The results demonstrate that the power of the slope test increases as the true w_S value deviates further from 2 (indicating greater difference between genetic and environmental decay rates) and as the error (ϵ) decreases. We present these results in **Supplementary Figure.1b**.

While our slope test primarily focuses on categorizing FR-reg coefficients into *slow*, *similar*, or *fast* decaying patterns, BIGFAM demonstrates robust performance even in cases where shared environmental effects increase with degree of relationship (*increase*, i.e., $w_S < 1$). Although the slope test's power is 0 in these *increase* scenarios (as they fall outside our slope test classification framework), BIGFAM's subsequent prediction step effectively handles such cases with large confidence intervals that typically include zero (**Fig.2b**). This ensures reliable interpretation of results even when the true environmental effects deviate from the slope test's results.

We also mentioned the results of type 1 error and power analysis of slope test in the **revised manuscript** as follows:

The slope test classifies these patterns based on whether the decay rate of shared environmental effects significantly differs from that of genetic effects. Specifically, using the estimated slope and its 95% confidence intervals (CIs), the test first determines if the decay pattern is similar (interval includes 1) or different (interval excludes 1). If the pattern is different, it is further classified as fast decay if the lower bound exceeds 1, or slow decay if the upper bound is below 1. The reliability of this classification is confirmed by type 1 error and power analysis (Fig.S1).

We are grateful for your thoughtful question, which has prompted us to thoroughly validate this critical aspect of slope test. These additional analyses strengthen our confidence in the reliability of BIGFAM's variance component estimates and provide important context for interpreting our results.

Reference

1. Jelenkovic, A. *et al.* Genetic and environmental influences on height from infancy to early adulthood: An individual-based pooled analysis of 45 twin cohorts. *Sci Rep* **6**, 28496 (2016).
2. Dewau, R. *et al.* Meta-Analysis of the Heritability of Childhood Height From 560 000 Pairs of Relatives Born Between 1929 and 2004. *American Journal of Human Biology* **37**, e24188 (2025).
3. Ingebrigtsen, T. S. *et al.* Genetic Influences on Pulmonary Function: A Large Sample Twin Study. *Lung* **189**, 323–330 (2011).
4. Jacobson, K. C., Prescott, C. A., Neale, M. C. & Kendler, K. S. Cohort differences in genetic and environmental influences on retrospective reports of conduct disorder among adult male twins. *Psychological Medicine* **30**, 775–787 (2000).
5. Srivastava, A. K., Williams, S. M. & Zhang, G. Heritability Estimation Approaches Utilizing Genome-Wide Data. *Current Protocols* **3**, e734 (2023).
6. Barry, C.-J. S. *et al.* How to estimate heritability: a guide for genetic epidemiologists. *International Journal of Epidemiology* **52**, 624–632 (2023).